# Multi-Objective Markov Games: Theoretic Foundations and Learning Algorithms

## Abstract

In practical multi-agent systems, agents often have diverse objectives, which makes the system more complex, as each agent's performance across multiple criteria depends on the joint actions of all agents, creating intricate strategic trade-offs. To address this, we introduce the Multi-Objective Markov Game (MOMG), a framework for multi-agent reinforcement learning with multiple objectives. We propose the Pareto-Nash Equilibrium (PNE) as the primary solution concept, where no agent can unilaterally improve one objective without sacrificing performance on another. We prove existence of PNE, and establish an equivalence between the PNE and the set of Nash Equilibria of MOMG's corresponding linearly scalarized games, enabling solutions of MOMG by transferring to a standard single-objective Markov game. However, we note that computing a PNE is theoretically and computationally challenging, thus we propose and study weaker but more tractable solution concepts. Building on these foundations, we develop online learning algorithm that identify a single solution to MOMGs. Furthermore, we propose a novel two-phase, preference-free algorithm that decouples exploration from planning. Our algorithm enables computation of a PNE for any given preference profile without collecting new samples, providing an efficient methodological characterization of the entire Pareto-Nash front.

## 1 Introduction

Multi-agent systems (MAS) (Weiss, 1999; Shoham & Leyton-Brown, 2008; Wooldridge, 2009; Olfati-Saber et al., 2007; Dorri et al., 2018) are becoming integral to complex, real-world domains, from the management of robotic warehouses (Lowe et al., 2017; Matignon et al., 2012), automated trading in financial markets (Wellman, 2006; Mi et al., 2023; Huang et al., 2024; Brusatin et al., 2024), and network management (Zhang et al., 2018; Abrol et al., 2024; Shabestary & Abdulhai, 2022). Multi-Agent Reinforcement Learning (MARL) (Zhang et al., 2021; Hernandez-Leal et al., 2019; Yang et al., 2020; Lowe et al., 2017; Zhu et al., 2024; Ning & Xie, 2024; Wellman et al., 2025) has emerged as the principal paradigm for engineering intelligent behavior in such systems, achieving remarkable empirical and theoretical progress, e.g., (Zhang et al., 2021; Hernandez-Leal et al., 2019; Yang et al., 2020; Lowe et al., 2017; Zhu et al., 2024; Ning & Xie, 2024; Wellman et al., 2025; Huh & Mohapatra, 2024; Yu et al., 2021a; Hu et al., 2021; Hsu et al., 2025; Wai et al., 2018; Doan et al., 2019; Macua et al., 2014; Stanković & Stanković, 2016; Zhang et al., 2018). Yet, the predominant formulation in MARL research rests upon a simplifying assumption that is misaligned with modern autonomous systems: that the goal of each agent can be adequately captured by a single, scalar reward function. This single-reward paradigm represents a bottleneck, hindering the development of autonomous agents that must operate in complex socio-technical environments.

In most practical settings, agents are seldom driven by a single, monolithic objective. Instead, they may navigate a landscape of diverse and often conflicting goals (Eriksson & Weber, 2008; Osika et al., 2024; Chapman et al., 2023). For instance, a financial trading algorithm must weigh the imperative of profit maximization against the critical need for risk mitigation (Addy et al., 2024; Olanrewaju, 2025; Bhardwaj et al., 2024). This multi-objective reality is a defining feature of modern autonomy, and while the field of Multi-Objective Reinforcement Learning (MORL) (Van Moffaert et al., 2013b; Van Moffaert & Nowé, 2014; Yang et al., 2019; Hayes et al., 2021; Liu et al., 2014) provides a robust framework for single-agent decision-making with such vectorial rewards, it assumes a stationary, non-strategic environment. This assumption breaks down in multi-agent settings.

The central challenge is not merely that each agent has multiple objectives, but that the optimal way to balance these objectives is inextricably linked to the actions of all other agents (Wong et al., 2023; Albrecht et al., 2024). An agent's set of achievable, optimal trade-offs—its Pareto front—is not a static object to be discovered; it is a dynamic outcome of strategic interaction. This strategic coupling problem is the core difficulty: an agent's optimal strategy for balancing its internal objectives is critically dependent on the strategies chosen by all other agents, who are simultaneously solving their own multi-objective trade-off problems. A naive approach where each agent independently solves its own MORL problem is fundamentally flawed. The agents' decision spaces are not independent but are inextricably linked, creating a complex web of strategic interdependencies.

This coupling necessitates a new class of models and solution concepts that formally unify the principles of game theory (Fudenberg & Tirole, 1991), which governs strategic stability, with the principles of multi-objective optimization (Gunantara, 2018; Deb et al., 2016; Konak et al., 2006; Caramia et al., 2020; Tamaki et al., 1996; Gagné et al., 2020), which defines rational choice under vectorial outcomes. Standard Markov games (Littman, 1994) are built upon the restrictive assumption of scalar rewards. This forces practitioners to pre-specify a fixed utility function, collapsing the rich, multi-dimensional preference space of an agent into a single number before the analysis begins, failing to capture the true nature of decision-making under competing objectives.

In this paper, we address this gap by introducing a comprehensive theoretical and algorithmic framework for multi-objective multi-agent learning. Our contributions are summarized as follows:

**A Formal Framework and Principled Solution Concept.** We introduce the Multi-Objective Markov Game (MOMG), a formal framework for modeling multi-agent reinforcement learning with vectorial rewards. We propose the Pareto-Nash Equilibrium (PNE) as its primary solution concept. A PNE is a policy profile where no agent can unilaterally improve one objective without sacrificing performance on another, thereby merging the strategic stability of a Nash Equilibrium with the rational choice criteria of Pareto optimality. We establish the foundational properties of this concept by proving that a PNE is guaranteed to exist in any finite MOMG.

**A Bridge to Tractability via Linear Scalarization**. We further establish a formal equivalence between the set of PNEs in an MOMG and the union of all Nash Equilibria of its corresponding linearly scalarized games. This result serves as the conceptual linchpin for our work, providing a constructive bridge from the novel, complex MOMG framework to the well-understood domain of standard single-objective Markov Games. This result transforms the research challenge from inventing entirely new multi-objective equilibrium-finding methods to leveraging the vast body of existing single-objective MG solvers.

**Provably and Efficient Algorithm Design.** The theoretical connections we derived enable us to reduce a MOMG to a standard MG through some pre-set preference, based on which we propose a Nash iteration algorithm to find a single PNE. Identifying the intractability of computing Nash-type equilibria, we further analyze weaker but more computationally efficient solution concepts, including the Weak Pareto-Nash Equilibrium (WPNE) and the Pareto Correlated Equilibrium (PCE). Our result further enables us to apply decentralized MARL algorithm to find a single PCE, enhancing efficiency in both sample and computational complexity. We further introduce a novel two-phase, preference-free methodology that decouples exploration from planning, representing a paradigm shift from identifying a single PNE to the Pareto-Nash front. Instead of recollecting samples and retraining from scratch for every new preference profile, our method invests in a single, preference-agnostic exploration phase to build a robust world model. This model can then be queried to compute the PNE for any given preference profile without collecting new samples. To ensure the model is accurate under any preference, we reformulate the exploration phase as a cooperative game, where all agents aim to reduce the estimation uncertainty, which efficiently ensures accuracy of any learned PNE from the model. Our algorithm thus enables applications where agent preferences may be unknown a priori or may change over time, allowing for recovering the entire Pareto-Nash front.

## 2 BACKGROUND AND PRELIMINARIES

### 2.1 FINITE-HORIZON MULTI-OBJECTIVE MARKOV DECISION PROCESSES

An MOMDP (Chatterjee et al., 2006; Wakuta & Togawa, 1998; Wiering & De Jong, 2007) is formally defined as a tuple $M = (\mathcal{S}, \mathcal{A}, P, \mathbf{r}, H)$, where $\mathcal{S}, \mathcal{A}$ are the finite state and action spaces, and

$P : \mathcal{S} \times \mathcal{A} \times [H] \to \Delta(\mathcal{S})$ is the transition kernel, where $P_h(s'|s, a)$ is the probability of transitioning from state $s$ to $s'$ after taking action $a$. The reward vector is $\mathbf{r} : \mathcal{S} \times \mathcal{A} \times [H] \to \mathbb{R}^M$, where $\mathbf{r}_h(s, a)$ is a $M$-dimensional vector that the agent received at step $h$ after taking action $a$ at state $s$.

A policy is a sequence of decision rules $\pi = (\pi_1, \ldots, \pi_H)$, where each $\pi_t : \mathcal{S} \to \Delta(\mathcal{A})$ specifies the action to take at step $h \in [H]$ under different states. The performance of a policy $\pi$ at time step $h$, denoted $\boldsymbol{V}_h^\pi(s)$, is a $m$-dimensional vector of the expected cumulative reward following $\pi$:

$$\boldsymbol{V}_h^\pi(s) = \mathbb{E}_{\pi, P} \left[ \sum_{k=h}^{H-1} \boldsymbol{r}_k(s_k, a_k) \mid s_h = s \right],$$

where the expectation is taken w.r.t. the policy and the transition kernels. As there is no single policy optimizing $\boldsymbol{V}$ for all objectives, MOMDP instead learns **Pareto optimal policies** (Wiering & De Jong, 2007; Roijers et al., 2021; Cai et al., 2023; Pirotta et al., 2015; Liu et al.; Lu et al., 2023; Qiu et al., 2024), defined as follows.

**Definition 1** (Pareto optimal policy). *Let $\boldsymbol{u} = (u_1, \ldots, u_M)$ and $\boldsymbol{v} = (v_1, \ldots, v_M)$ be two vectors in $\mathbb{R}^M$. The vector $\boldsymbol{u}$ Pareto dominates the vector $\boldsymbol{v}$ if it holds that: $u_i \geq v_i$ for all components $i \in [M]$; And there exists some $j \in [M]$ such that $u_j > v_j$. A policy $\pi^*$ is Pareto optimal if no other policy $\pi$ Pareto dominates it, i.e., there is no $\pi$ such that $\boldsymbol{V}_1^\pi(s_1)$ Pareto dominates $\boldsymbol{V}_1^{\pi^*}(s_1)$.*

## 2.2 Finite-Horizon Single-Objective Markov Games

A Finite-Horizon Single-Objective Markov Game (SMG) (Leonardos & Piliouras, 2022; Fan et al., 2019; Zhu & Zhao, 2020; Jin et al., 2022; Deng et al., 2023; Zhang et al., 2024; Fink, 1964), or Stochastic Game, models multi-agent interactions. Each agent seeks to maximize its own scalar reward. An $N$-player Markov Game is defined by the tuple $G = (\mathcal{N}, \mathcal{S}, \{\mathcal{A}_i\}_{i \in \mathcal{N}}, P, \{r_i\}_{i \in \mathcal{N}}, H)$. Here, $\mathcal{N} = \{1, 2, \ldots, N\}$ is the finite set of agents, and $P_h : \mathcal{S} \times \mathcal{A}(\triangleq \times_i \mathcal{A}_i) \to \Delta(\mathcal{S})$ is the transition function, dependent on a joint action $\boldsymbol{a} = (a_1, \ldots, a_N)$. Each agent $i$ then receives a scalar reward $r_h^i(s, \boldsymbol{a}) \in [0, 1]$ after all agents taking the joint action $\boldsymbol{a}$ at state $s$ and step $h$.

All agents' strategies is captured by a joint policy $\boldsymbol{\pi} : \mathcal{S} \to \Delta(\mathcal{A})$, a profile of policies for all agents. The value function for agent $i$ under $\boldsymbol{\pi}$ at time $t$ is:

$$V_{i,h}^{\boldsymbol{\pi}}(s) = \mathbb{E}_{\boldsymbol{\pi}, P} \left[ \sum_{k=h}^{H-1} r_k^i(s_k, \boldsymbol{a}_k) \mid s_h = s \right].$$

The elemental solution concept is the **Nash Equilibrium (NE)** (Shapley, 1953; Littman, 1994; Fink, 1964). A product policy $\boldsymbol{\pi}^*$ is an NE if no agent $i$ can improve its expected return by unilaterally changing its policy $\pi_{i,t}$, given that all other agents stick to $\boldsymbol{\pi}_{-i,t}^*$, i.e., for all $i \in \mathcal{N}$ and any alternative policy $\pi_i$:

$$V_{i,1}^{(\pi_i^*, \boldsymbol{\pi}_{-i}^*)}(s_1) \geq V_{i,1}^{(\pi_i, \boldsymbol{\pi}_{-i}^*)}(s_1).$$

## 3 Multi-Objective Markov Games and Solutions

In this section, we propose our formulation of multi-objective multi-agent MAS, the multi-objective Markov games (MOMGs) as follows.

**Definition 2** ($N$-Player General-Sum MOMG). *A finite-horizon $N$-player MOMG is specified as $(\mathcal{N}, \mathcal{S}, \{\mathcal{A}_j\}_{j=1}^N, H, P, \{\boldsymbol{r}_j\}_{j=1}^N)$ similarly to a Markov game, with each agent $j$ receiving a vectorial reward $\boldsymbol{r}_{j,h}(s, \boldsymbol{a}) \in [0, 1]^M$ at step $h$ under the joint action $\boldsymbol{a}$ and state $s$.*

Similarly, the agents' strategy is measured by a joint policy $\boldsymbol{\pi}$. Given a joint policy $\boldsymbol{\pi}$, the vector-valued state-value function for player $j$ is $\boldsymbol{V}_{j,h}^{\boldsymbol{\pi}}(s) = (V_{j,h}^{\boldsymbol{\pi}, r_j^1}(s), \ldots, V_{j,h}^{\boldsymbol{\pi}, r_j^M}(s))$, where each entry is the standard value function w.r.t. the reward entry $r_j^{\cdot}$.

Our formulation is an extension of MOMDP and SMG, yet neither of their solutions can solve our MOMG formulation, hence we then study the solutions to our MOMGs. We begin by defining the ideal solution concept, the Pareto-Nash Equilibrium, and establishing its fundamental properties. We

then show that it poses challenges for algorithmic verification, which motivates our introduction of a weaker notion, a computationally grounded solution that forms the basis for our practical algorithms.

We note that a similar formulation is also considered in (Yu et al., 2021b; Chang et al., 2014), yet with different learning goals.

### 3.1 SOLUTIONS TO MULTI-OBJECTIVE MARKOV GAMES

We first propose the following Pareto-Nash Equilibrium notion.

**Definition 3** (Pareto-Nash Equilibrium (PNE)). *A product policy $\boldsymbol{\pi}^* = (\pi_1^*, \ldots, \pi_N^*)$ is a PNE if for any agent $k \in \mathcal{N}$, $\pi_k^*$ is not Pareto dominated by any policy $\pi_k$ while other agents sticking to $\boldsymbol{\pi}_{-k}^*$, i.e., there is no policy $\pi_k$ such that $V_{k,1}^{(\pi_k, \boldsymbol{\pi}_{-k}^*)}(s_1)$ Pareto dominates $V_{k,1}^{\boldsymbol{\pi}^*}(s_1)$.*

A PNE is a policy profile where no single agent can find such a trade-off-free improvement while others deploying the PNE policy. Note that PNE generalizes both NE in SMGs, and Pareto optimum in MOMDPs. Specifically, when $N = 1$, a PNE reduces to a Pareto optimal policy; And when $M = 1$, a PNE becomes a standard NE. PNE hence merges both strategic stability from game theory with Pareto optimality from multi-objective optimization. A deviation is only unambiguously profitable if it results in a Pareto improvement—improving at least one objective without worsening any other. Any other change involves a trade-off an agent may be unwilling to make.

We then show that a stochastic PNE alwasys exists.

**Theorem 1** (Existence of PNE). *For any finite MOMG, there always exists a stochastic PNE.*

Note that even in general normal-form games or Markov games, NE are generally stochastic or mixed strategies (Fudenberg & Tirole, 1991; Osborne & Rubinstein, 1994), hence we mainly focus on stochastic policies in this paper.

Given the existence of PNE, a natural question is how to compute a PNE, or even find all of the PNE. We then provide a methodological solution through the following result, which directly connects MOMGs and standard MGs. Specifically, given a MOMG $G = (\mathcal{N}, \mathcal{S}, \{\mathcal{A}^k\}, H, P, \{\boldsymbol{r}^k\})$, and a preference vector $\boldsymbol{\Lambda} = \{\boldsymbol{\lambda}^1, \boldsymbol{\lambda}^2, \ldots, \boldsymbol{\lambda}^N\}$, where each $\boldsymbol{\lambda}^k \in \Delta_M^o$[1] being a distribution over $[M]$ with positive entries, its corresponding linear scalarized SMG is $G_{\boldsymbol{\Lambda}} = (\mathcal{N}, \mathcal{S}, \{\mathcal{A}_i\}_{i \in \mathcal{N}}, P, \{r_i = (\boldsymbol{\lambda}^i)^\top \boldsymbol{r}_i\}_{i \in \mathcal{N}}, H)$, i.e., the agent receives a scalar reward as $r_i = (\boldsymbol{\lambda}^i)^\top \boldsymbol{r}_i$.

We then show the following equivalence.

**Theorem 2** (Equivalence between MOMGs and Linear Scalarized SMGs). *The set of PNE of a MOMG is equivalent to the union of all NE of linear scalarizations with positive preferences:*

$$\boldsymbol{PNE}(G) = \cup_{\boldsymbol{\Lambda} \in (\Delta_M^o)^N} \boldsymbol{NE}(G_{\boldsymbol{\Lambda}}). \tag{1}$$

The above results imply that finding the NE of linear scalarization games with any positive preference $\boldsymbol{\lambda}$ can specify a PNE for the multi-objective Markov game; More importantly, all of the PNE can be specified in this way, by solving the linear scalarized SMGs. Thus we are able to reduce the complicated MOMG to widely studied SMGs, and adapt effective solutions therein to find PNE.

**Remark 1.** *We note that the notion of PNE and results of similar forms are studied in multi-objective normal-form games (Lozovanu et al., 2005; Zhao, 1991; Qu et al., 2015; Wang, 1993). However, our studies are not direct extensions. In normal-form games, the set of achievable payoffs for a player using mixed strategies is inherently a convex combination of the pure strategy payoffs, or is guaranteed through additional quasi-convexity on the utility functions. However, in a Markov Game, an agent's value vector is an expectation over entire trajectories of states and actions, which is generally not (quasi-)convex (Chakravorty & Mahajan, 2018), thus the previous approaches cannot be applied. Instead, we study the occupancy measure of the underlying Markov games to ensure the convexity and derive our results (see Proposition 4 and Lemma 1).*

### 3.2 WEAK PARETO-NASH EQUILIBRIA

In some cases, it is generally sufficient to find some approximated solutions. Hence we now consider the inverse problem of finding a PNE: given a policy, we want to measure the closeness of it against

---

[1]$\Delta_M$ is the probability simplex over $[M] \triangleq \{1, ..., M\}$, and $\Delta_M^o$ is its interior: $\Delta_M^o \triangleq \{\mathbf{q} \in \Delta_M : \mathbf{q} > 0\}$.

some PNE. Since every PNE is a NE of some linear scalarized game, we utilize the Nash Gap (Jin et al., 2022; Ma et al., 2023) defined in single-objective games with an adaptive preference distribution, to define a Pareto-Nash Gap as

$$\mathrm{PNG}(\boldsymbol{\pi}) := \max_{k \in [N]} \sup_{\pi'_k} \inf_{\boldsymbol{\lambda} \in \Delta_M^o} \left\{ (\boldsymbol{\lambda})^\top (\boldsymbol{V}_{k,1}^{(\pi'_k, \boldsymbol{\pi}_{-k})}(s_1) - \boldsymbol{V}_{k,1}^{\boldsymbol{\pi}}(s_1)) \right\}. \tag{2}$$

Notably, this definition generalizes the notion of Nash gap in single-agent Markov games, and the notion of Pareto gap in single-agent multi-objective RL (Qiu et al., 2024; Jiang et al., 2023b; Lu et al., 2019; Turgay et al., 2018; Drugan & Nowe, 2013). However, in the following result, we show that, having a zero PNG does not imply the policy being a PNE.

**Proposition 1.** *If a policy profile $\boldsymbol{\pi}$ is a Pareto-Nash Equilibrium, then its PNG is zero. However, zero PNG does not imply $\boldsymbol{\pi}$ is a PNE.*

**Remark 2.** *The result seems to be a contraction with Theorem 2, which states that any linear scalarized game NE (with some $\boldsymbol{\Lambda}$) is a PNE, as the a zero-PNG seems to imply the policy is a NE for some $\boldsymbol{\Lambda}$ (which is the solution to $\inf_{\boldsymbol{\lambda}}(\cdot)$-scalarized game). However, this is not true. The reason is that $\Delta_M^o$ is not a compact set (since it is open), hence the $\inf$ may not be obtained within it but at the boundary of it, i.e., $\Delta_M / \Delta_M^o$.*

This result implies that, PNG of a policy does not imply whether it is a PNE, thus there is no way to capture the 'optimism' of it. To address this issue, we introduce a weaker solution as follows.

**Definition 4** (Weakly Pareto-Nash Equilibrium (WPNE)). *A vector $u \in \mathbb{R}^M$ **strictly Pareto dominates** a vector $v \in \mathbb{R}^M$ if $u_i > v_i$ for all $i \in [M]$. A product policy $\boldsymbol{\pi}^*$ is a Weak Pareto-Nash Equilibrium (WPNE) if for any agent $k$, there does not exist any unilateral deviation policy $\pi'_k$ such that the resulting value vector $\boldsymbol{V}_{k,1}^{(\pi'_k, \boldsymbol{\pi}_{-k}^*)}(s_1)$ strictly dominates the original value vector $\boldsymbol{V}_{k,1}^{\boldsymbol{\pi}^*}(s_1)$.*

A WPNE is a product policy that no unilateral deviation can improve performance along **all** objectives. Hence, all PNE is also WPNE, and WPNE always exists.

We further derive the following theoretical characterizations of WPNE, which enable us to develop concrete learning algorithms.

**Theorem 3.** *(a). The set of WPNE is equivalent to the union of all NE of linear scalarizations with non-negative preferences:*

$$\boldsymbol{WPNE}(G) = \cup_{\boldsymbol{\Lambda} \in (\Delta_M)^N} \boldsymbol{NE}(G_{\boldsymbol{\Lambda}}). \tag{3}$$

*(b). Define the Weak Pareto-Nash gap (WPNG) for a given policy $\pi = (\pi_1, \ldots, \pi_N)$ as:*

$$WPNG(\pi) := \max_{k \in [N]} \sup_{\pi'_k} \inf_{\boldsymbol{\lambda} \in \Delta_M} \left\{ (\boldsymbol{\lambda})^\top (V_{k,1}^{(\pi'_k, \pi_{-k})}(s_1) - V_{k,1}^{\pi}(s_1)) \right\},$$

*where the supremum is taken over all possible deviating policies $\pi'_k$ for agent $k$. Then, a policy profile $\pi$ is a Weak Pareto-Nash Equilibrium if and only if its Multi-Agent Weak PNG is zero: $WPNG(\pi) = 0 \iff \pi$ is a WPNE.*

These results hence imply the practical solvability of WPNE through solving the linear scalarized game. Namely, given an preference distribution $\boldsymbol{\Lambda} \in (\Delta_M)^N$, an $\epsilon$-NE of $G_{\boldsymbol{\Lambda}}$ is also approximately a WPNE (with an $\epsilon$-WPNG). This enhanced solvability is the major advantage of the notions of WPNE compared to PNE. Hence in the following sections, we will mianly aim to develop algorithms to find WPNE of the game. We also refer to the set of all WPNG as a Pareto-Nash front.

### 3.3 UTILITY-BASED EQUILIBRIA FOR MULTI-OBJECTIVE MARKOV GAMES

Besides the Pareto-Nash Equilibrium we proposed, another class of solutions to multi-objective problem is based on utility functions. A utility function $u : \mathbb{R}^M \to \mathbb{R}$ measures some prior preference of an agent, transferring an $M$-dimensional vector to a scalar, and hence reduce the multi-objective problems to single-objective ones. It has been extensively studied in multi-objective MDPs (Lozovanu et al., 2005) and multi-objective normal-form games (Rodríguez Soto et al., 2024; Röpke et al., 2022; Rădulescu et al., 2020; Borm et al., 1988; Lozovanu et al., 2005; Röpke et al., 2023). We generalize these studies and propose two utility-based solutions to MOMGs, as follows.

**Definition 5** (ESR). *A product policy $\boldsymbol{\pi}^* = (\pi_1^*, \ldots, \pi_N^*)$ is an **Expected Scalarized Return Nash Equilibrium (ESR-NE)** if it is a NE for the SMG with Scalarized reward $\bar{r}_k = u_k(\boldsymbol{r}_k)$. Namely, for any agent $k$ and any policy $\pi_k$, it holds that*

$$V_{\bar{r}_k,1}^{(\pi_k,\boldsymbol{\pi}_{-k}^*)}(s_1) \leq V_{\bar{r}_k,1}^{(\pi_k^*,\boldsymbol{\pi}_{-k}^*)}(s_1). \tag{4}$$

ESR can be viewed as an extension of the linear scalarization discussed above with non-linear utility functions. However, as we shall discuss later, it can result in a less consistent solution notion.

**Definition 6** (SER). *A product policy $\boldsymbol{\pi}^* = (\pi_1^*, \ldots, \pi_N^*)$ is a **Scalarized Expected Return Nash Equilibrium (SER-NE)** if for any agent $k$ and any policy $\pi_k$, it holds that*

$$u_k(\boldsymbol{V}_{k,1}^{(\pi_k,\boldsymbol{\pi}_{-k}^*)}(s_1)) \leq u_k(\boldsymbol{V}_{k,1}^{(\pi_k^*,\boldsymbol{\pi}_{-k}^*)}(s_1)). \tag{5}$$

SER assigns some preference to the value functions under different rewards, transferring the MOMG to a single-objective normal-form game, i.e., a single-step decision game with a payoff function $u_k$. Note that when $u$ is linear, SER is equivalent to SER.

Despite these two notions are extensively studied in multi-objective normal-form game and multi-objective RL, we argue that they may be less suitable than our PNE notion for POMGs, through the following two aspects: existence and inconsistency.

**Proposition 2.** *(Existence) An ESR-NE always exists. An SER-NE may not always exist; If the utility functions are continuous and quasi-concave, then there always exists an SER-NE.*

As proved, SER-NE may not always exist expect with additional conditions on the utility functions. Such a result aligns with the previous studies of ESR in multi-objective normal-form games (Röpke et al., 2022; 2023) or multi-objective RL (Agarwal et al., 2022; Guidobene et al., 2025). However, our PNE is guaranteed to exist, thus has a better applicability.

On the other hand, ESR can be viewed as a strict extension of the linear sclarization and ESR-NE always exists. One potential hope is to consider ESR-NE to gain some additional benefits compared to linear scalarization. However, we use the following result to justify that, considering ESR-NE with non-linear utility function can results in inconsistency with special cases of MOMG.

**Proposition 3.** *(Inconsistency) There exists a single-agent multi-objective MDP and a concave utility function, whose ESR-NE (i.e., an ESR-optimal policy) is not weakly Pareto optimal.*

This result implies that, although ESR-NE exists, it may not align with the nature of multi-objective learning. Using it in single-agent setting can result in solutions that is not (weakly) Pareto optimal. Same issue also exists for SER-NE. The utility functions are additionally assumed to be element-wise strictly monotonically increasing or concave for single-agent multi-objective RL, to ensure the solution to the Pareto optimal equation is Pareto optimal (Agarwal et al., 2022; Guidobene et al., 2025). Yet our PNE is consistent with all of MOMGs' special cases.

Given these understandings, we focus on designing efficient algorithms to find weakly PNE.

## 4 ONLINE LEARNING FOR SINGLE WPNE IDENTIFICATION

Based on our previous results, solving for a single (W)PNE can be connected to a standard Markov game through linear scalarization. We then consider the online learning setting, where all agents need to explore the unknown environment to solve the MOMG. We consider the games settings with random, unknown reward functions, where agents can only observe a random realization of rewards (note that in standard MGs, the rewards are assumed to be known). We first show that the convergence criteria used in MG, the Nash regret, can also be used in our MOMGs to quantify the closeness of the learned policy to an arbitrary WPNE, i.e., if an algorithm learns an approximated equilibrium in the scalarized game, it is also an approximated WPNE, and thus we can utilize any MG algorithms to find a single WPNE.

**Definition 7.** *The **Nash Regret** is the Nash Gaps of the sequence of policies, $\boldsymbol{\pi}^1, \ldots, \boldsymbol{\pi}^T$, executed by the algorithm over $T$ episodes: $Regret(T) := \sum_{t=1}^T \max_k \left( \max_{\pi'^k} U^{k,(\pi'^k,\boldsymbol{\pi}^{-k,t})}(s_1) - U^{k,\boldsymbol{\pi}^t}(s_1) \right)$, where $U^{k,\boldsymbol{\pi}}(s_1) = \boldsymbol{\lambda}_k^\top \boldsymbol{V}_{k,1}^{\boldsymbol{\pi}}(s_1)$ is the player $k$'s linear scalarized value function.*

This metric quantifies the total cumulative loss from not playing a Nash Equilibrium of the linearly scalarized reward in every episode. We can show that, for any linear scalarized game, its corresponding Nash regret upper bounds the WPNG (the major goal in online MOMGs). Namely, if $\max_k \left( \max_{\pi'^k} U^{k,(\pi'^k, \boldsymbol{\pi}^{-k})}(s_1) - U^{k,\boldsymbol{\pi}}(s_1) \right) \leq \epsilon$, then the WPNG will be also smaller than $\epsilon$:

$$\text{WPNG}(\boldsymbol{\pi}) = \max_k \max_{\pi'^k} \inf_{\boldsymbol{\lambda}_k} \boldsymbol{\lambda}_k^\top (V_{k,1}^{(\pi'_k, \pi_{-k})}(s_1) - V_{k,1}^\pi(s_1))$$

$$\leq \max_k \max_{\pi'^k} \boldsymbol{\lambda}_k^\top (V_{k,1}^{(\pi'_k, \pi_{-k})}(s_1) - V_{k,1}^\pi(s_1)) \leq \epsilon, \tag{6}$$

implying an approximate WPNE. Hence solving for a single WPNE essentially reduces to solve a standard MG, and we can use the Nash regret in Definition 7 to quantify the accuracy of the solution. We then extend the standard Nash iteration algorithm for standard MGs (Liu et al., 2021b) to the random, unknown reward setting, and develop our **Optimistic Nash Value Iteration for Multi-Objective Games (ONVI-MG)** for single WPNE identification.

## 4.1 OPTIMISTIC NASH VALUE ITERATION

The core principle is to develop the optimism in the face of uncertainty framework for our multi-agent setting to tackle the unknown rewards and dynamics. Notably, in our setting, the reward is an $M$-dim vector, which might have higher sample complexity to estimate. However, since we have a pre-set preference, we can estimate the scalarized reward only, bypassing the inefficient vector estimation. Toward this, our algorithm maintains empirical counts $N_h^t(s, \boldsymbol{a})$ for each state-joint-action triplet $(s, \boldsymbol{a})$ at step $h$ up to episode $t$. From these counts, we construct an empirical model consisting of the transition probabilities $\hat{P}_h^t$ and mean reward vectors $\hat{\boldsymbol{r}}_h^{k,t}$ for each agent. To encourage exploration to reduce both uncertainties, we define shared bonus terms inversely related to the number of times a joint action has been taken: $\Psi_h^t(s, \boldsymbol{a}) := \sqrt{\frac{c_1 \log(SAHT/\delta)}{N_h^{t-1}(s,\boldsymbol{a}) \vee 1}}$ (Reward), $\quad \Phi_h^t(s, \boldsymbol{a}) :=$ $H\sqrt{\frac{c_2 S \log(SAHT/\delta)}{N_h^{t-1}(s,\boldsymbol{a}) \vee 1}}$ (Transition), where $c_1, c_2$ are suitable constants and $x \vee y = \max(x, y)$.

The algorithm then proceeds via backward induction in each episode to compute an optimistic joint policy. For each state $s$ and step $h$, agents compute and execute a NE for a one-shot 'stage game' where the payoffs are optimistic Q-values: $Q_h^{k,t}(s, \boldsymbol{a}) := \min \left\{ H, (\boldsymbol{\lambda}^k)^\top \hat{\boldsymbol{r}}_h^{k,t}(s, \boldsymbol{a}) + \Psi_h^t(s, \boldsymbol{a}) + \sum_{s'} \hat{P}_h^t(s'|s, \boldsymbol{a}) U_{h+1}^{k,t}(s') + \Phi_h^t(s, \boldsymbol{a}) \right\}$, where $U_{h+1}^{k,t}(s')$ is the value function for agent $k$ at the next stage, computed from the NE of that stage's game. We defer our algorithm (Algorithm 2) to Appendix.

We then derive the theoretical guarantee on the regret of our algorithm.

**Theorem 4.** *With probability at least $1 - \delta$, $\text{Regret}(T) \leq \mathcal{O}\left(H^2 S \sqrt{AT \log(SAHT/\delta)}\right)$. To find an $\epsilon$-WPNE, it requires a sample complexity of $\tilde{\mathcal{O}}\left(\frac{H^5 S^2 A}{\epsilon^2}\right)$.*

The result implies that, our algorithm can find a single WPNE in a sample efficient fashion, which hence presents the first concrete and provable solution to our MOMGs.

## 4.2 PARETO CORRELATED EQUILIBRIUM

Despite our ONVI algorithm can identify a single WPNE, the sample complexity depends on the joint action space, suffering from the multi-agency curse (Jin et al., 2021). Moreover, a computation of a matrix-formed Nash equilibrium, which is PPAD-hard (Deng et al., 2023; Jin et al., 2022), is required in each iteration. Although assuming NE's solvability and mainly considering statistical complexity is a standard in Markov game learning, e.g., (Liu et al., 2021b; Hu & Wellman, 2003; Li, 2003), it is also motivated to study weaker equilibria with better computational complexity. Toward these, in this section, we will propose a further relaxation of the PNE, Pareto Correlated Equilibrium, and design algorithms to identify it with more efficient sample and computational complexity.

**Definition 8** (Pareto Correlated Equilibrium). *A joint policy $\pi$ is called a **Pareto Correlated Equilibrium** (PCE) if for any player $j$ and any stochastic modification $\phi^{(j)}$ (Osborne & Rubinstein,*

*1994), it holds that the value vector for player $j$ under the modified policy, $\boldsymbol{V}_{j,1}^{\phi^{(j)}\circ\boldsymbol{\pi}}(s_1)$, does not Pareto dominate the value vector under the original policy, $\boldsymbol{V}_{j,1}^{\boldsymbol{\pi}}(s_1)$.*

We further show that, PCE also enjoys a similar equivalence as PNE.

**Theorem 5.** *Any finite Multi-Objective Markov Game has a Pareto Correlated Equilibrium. Moreover, the set of PCEs is equivalent to the union of CEs of positive scalarization MGs:*

$$\boldsymbol{PCE}(G) = \cup_{\boldsymbol{\Lambda}\in(\Delta_M^o)^N}\boldsymbol{CE}(G_{\boldsymbol{\Lambda}}). \tag{7}$$

Thus a (weakly) PCE can similarly be identified through finding a CE of the linearly scalarized SMG. We thus propose our multi-objective V-learning (MO-V-Learning), which takes a fixed preference profile $\boldsymbol{\Lambda} = \{\boldsymbol{\lambda}^1, \ldots, \boldsymbol{\lambda}^N\} \in (\Delta_M)^N$ as input and finds an $\epsilon$-CE for $G_{\boldsymbol{\Lambda}}$, which is also an $\epsilon$-WPCE for $G$. We defer our algorithm to Algorithm 3 in Appendix.

We then show that MO-V-Learning finds an $\epsilon$-WPCE for the given $\Lambda$ with a sample complexity that scales polynomially in $\max_j A_j$ instead of $\prod_j A_j$, and hence breaks the curse of multi-agency in MOMGs. Moreover, in our MO-V-Learning algorithm, only computations of CE in matrix-formed games are required, which can be done within polynomial time (Farina & Sandholm, 2020; Papadimitriou & Roughgarden, 2008), hence also enjoys better computational complexity.

**Theorem 6.** *with probability at least $1 - \delta$, MO-V-Learning (see Algorithm 3 in Appendix) finds an $\epsilon$-WPCE, with the sample complexity $\tilde{\mathcal{O}}\left(\frac{H^6 S \max_j A_j^2}{\epsilon^2}\right)$.*

## 5 TWO-PHASE PREFERENCE-FREE ALGORITHMS FOR PN FRONT

The algorithms we developed provide a crucial first step towards solving MOMGs, offering provably efficient methods for identifying a single WPNE or WPCE. However, their direct application is predicated on a significant assumption: that a fixed, known preference profile $\Lambda$ for all agents is provided as input. While effective in scenarios with pre-defined and static agent objectives, this preference-conditioned paradigm reveals a substantial practical and computational bottleneck when agent preferences are unknown or subject to change over time, or we aim to understand the full landscape of strategic trade-offs—the entire Pareto-Nash front. Characterizing the Pareto-Nash front by naively applying a preference-conditioned algorithm entails executing the entire online learning process repeatedly for a multitude of different preference profiles, which is computationally prohibitive and profoundly data-inefficient.

A more principled methodology can be developed from the fundamental observation that the underlying dynamics of the MOMG are invariant to the agents' preferences. The costly process of learning these dynamics through environmental interaction can be decoupled from the less expensive, purely computational process of planning an equilibrium for a specific preference profile.

This motivates a shift in paradigm towards a two-phase framework. The first phase involves a significant, one-time investment in a comprehensive, preference-agnostic exploration strategy designed to build a high-fidelity empirical model of the game. The second phase leverages this reusable model to compute, on-demand, the PNE for any given preference profile without the need for any additional environmental samples. The objective thus shifts from learning a single PNE to learning an accurate model of the entire MOMG.

The major challenge of such an algorithm is to efficiently explore the environment and collect data, to ensure the learned model is accurate enough to learn all the Pareto-Nash front (instead of a single one), while maintaining sample efficient. In standard MGs with a single reward/objective, explorations are only encouraged for those high-valued state-action pairs to ensure low regret; Whereas in our setting, the preference is unknown and we have to strategically explore the whole environment to ensure accuracy of planning under any preferences. To achieve this, our algorithm temporarily recasts the general-sum MOMG into a fully cooperative, common-payoff game, by defining a single reward as as the maximum of all uncertainties (see Line 9 in Algorithm 1). The NE of the resulting cooperative game will naturally incentivize the agents to choose joint actions that visit state-action pairs where at least one component of the system model (a transition probability, or a reward for any agent's objective) is highly uncertain, thus ensuring comprehensive exploration. We further show

that such a strategy utilizes the underlying multi-agent structure, and the collected data can ensure the accuracy of the learned policy from Phase-2. Our algorithm is presented in Algorithm 1.

---

**Algorithm 1** Two-Phase Multi-Player Learning

---

1: Phase 1: Preference-Free Exploration
2: **Initialize:** Dataset $\mathcal{D} \leftarrow \emptyset$, counts $N_h(s, \boldsymbol{a}) \leftarrow 0$. Let $C_r, C_p$ be logging constants.
3: **for** $t = 1, 2, \ldots, T$ **do**
4:      **for** $h = H, \ldots, 1$ **do**
5:          For all $(s, \boldsymbol{a})$, calculate bonuses based on counts $N_h^{t-1}(s, \boldsymbol{a})$:
6:          $\Psi_{j,i,h}^t(s, \boldsymbol{a}) \leftarrow \sqrt{\frac{C_r}{N_h^{t-1}(s,\boldsymbol{a}) \vee 1}} \wedge 1 \quad \forall j \in [N], \forall i \in [M]$
7:          $\Phi_h^t(s, \boldsymbol{a}) \leftarrow \sqrt{\frac{C_p S H^2}{N_h^{t-1}(s,\boldsymbol{a}) \vee 1}} \wedge H$
8:          Define a single, shared uncertainty reward for the exploration game:
9:          $\bar{r}_h^t(s, \boldsymbol{a}) \leftarrow \max\{\Phi_h^t(s, \boldsymbol{a})/H, \{\Psi_{j,i,h}^t(s, \boldsymbol{a})\}_{j,i}\}$
10:          Construct optimistic Q-function for the exploration game:
11:          $\bar{Q}_h^t(s, \boldsymbol{a}) \leftarrow \{\bar{r}_h^t(s, \boldsymbol{a}) + \sum_{s'} \hat{P}_h^{t-1}(s'|s, \boldsymbol{a}) \bar{V}_{h+1}^t(s') + \Phi_h^t(s, \boldsymbol{a})\}_{[0, H-h+1]}$
12:          where $\bar{V}_{h+1}^t(s')$ is the value of a Nash Equilibrium of the game at step $h + 1$.
13:          Compute policies $\bar{\boldsymbol{\pi}}_h^t = (\bar{\pi}_{1,h}^t, \ldots, \bar{\pi}_{N,h}^t)$ as an NE of normal-form game $\bar{Q}_h^t(s, \cdot)$.
14:      **end for**
15:      Execute $\bar{\boldsymbol{\pi}}^t$ for one episode, collecting trajectory $\{s_h^t, \boldsymbol{a}_h^t, \{\boldsymbol{r}_{j,h}^t\}_{j=1}^N\}_{h=1}^H$.
16:      Add trajectory to $\mathcal{D}$ and update counts $N_h(s, \boldsymbol{a})$.
17: **end for**
18: Phase 2: Planning with Preferences
19: **Input:** Preference profile $\boldsymbol{\Lambda} = (\boldsymbol{\lambda}_1, \ldots, \boldsymbol{\lambda}_N)$
20: Estimate empirical model $(\{\hat{\boldsymbol{r}}_{j,h}\}_{j=1}^N, \hat{P}_h)$ from the collected dataset $\mathcal{D}$.
21: For each player $j \in [N]$, compute scalarized reward: $\hat{r}_{j,\boldsymbol{\lambda}_j,h}(s, \boldsymbol{a}) \leftarrow \sum_{i=1}^M \lambda_{j,i} \hat{r}_{j,i,h}(s, \boldsymbol{a})$.
22: Solve the estimated $N$-player game $(\mathcal{S}, \{\mathcal{A}_j\}, H, \hat{P}, \{\hat{r}_{j,\boldsymbol{\lambda}_j}\})$ to find an NE: $\hat{\boldsymbol{\pi}}_{\boldsymbol{\Lambda}}^*$.
23: **Return:** Policies $\hat{\boldsymbol{\pi}}_{\boldsymbol{\Lambda}}^*$.

---

We then develop our theoretical results, showing the efficiency of our two-phase approach.

**Theorem 7.** *With probability at least $1 - \delta$, the policy $\hat{\boldsymbol{\pi}}_{\boldsymbol{\Lambda}}^*$ returned by Algorithm 1 for any input preference profile $\boldsymbol{\Lambda}$ is an $\epsilon$-WPNE, as long as $T = \tilde{\mathcal{O}}\left(\frac{H^8 S^2 N^2 M^2 A}{\epsilon^2}\right)$.*

**Remark 3.** *We note our sample complexity depends on the joint action space size A. However, we note that the goal of Phase-1 is to learn the model, instead of merely learning a PCE, thus it is expected that the model under all joint actions should be explored. We hence conjecture that there exists a efficiency-generalizbility trade off, and leave it as a future research problem that whether such complexity can be improved.*

Our result hence implies that, our two-phase algorithm can recover the whole WPN front and solve MOMGs efficiently. We further include a numerical verification in Section B.

## 6 CONCLUSION

In this paper, we proposed multi-agent Markov Games as a fundamental framework for multi-agent RL with multiple and diverse objectives. We then studied the solvability of our MOMG framework, proposing the Pareto-Nash Equilibrium and its weaker variants as the primal solutions of MOMGs. We then developed an essential equivalence between our MOMGs and the single-objective Markov games. Based on it, we then proposed sample-efficient online algorithms to identify a single equilibrium of the MOMG, providing the first concrete and provably convergent algorithm for MOMGs. We further developed a two-phase algorithm that is able to recover the Pareto-Nash front through an one-time data collection. Our studies hence provided both theoretical foundations and algorithmic solutions to multi-agent multi-objective RL, enjoying a wide applicability for practical multi-agent systems with diverse or multi-modal rewards or objectives.

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

## A   RELATED WORKS

**Markov Games.** Markov games (MGs), or stochastic games, introduced by (Shapley, 1953; Fink, 1964), form the standard foundation for multi-agent reinforcement learning (MARL), particularly in equilibrium learning. Comprehensive surveys such as (Busoniu et al., 2008; Oroojlooy & Hajinezhad, 2023; Zhang et al., 2021) offer thorough coverage of the field's evolution. Early work in MARL focused on asymptotic convergence guarantees (Littman et al., 2001; Littman & Szepesvári, 1996), whereas recent research emphasizes finite-sample analyses to establish non-asymptotic guarantees, especially for learning Nash equilibria (NE)—a central solution concept. The existence of NE in general-sum MGs was shown by (Fink, 1964), and the algorithmic foundation was laid by the seminal work of (Littman, 1994). Classical algorithms such as Nash-Q (Hu & Wellman, 2003) and FF-Q (Littman et al., 2001) were proposed to compute NE and its variants. However, computing NE in general-sum multi-player settings remains PPAD-complete (Daskalakis, 2013), and no polynomial-time algorithms exist for this case (Jin et al., 2022; Deng et al., 2023). In contrast, the two-player zero-sum setting admits tractable solutions, with the first polynomial-time algorithm developed by (Hansen et al., 2013). To address the computational intractability in general-sum MGs, attention has shifted to weaker notions like CE and CCE, with polynomial-time algorithms such as V-learning (Jin et al., 2021; Mao & Başar, 2023; Song et al., 2021) and Nash value iteration (Liu et al., 2021b) enabling efficient computation. Furthermore, significant progress in finite-sample analysis—spanning both model-based and model-free algorithms—has been achieved in the two-player zero-sum setting, as evidenced by (Bai & Jin, 2020; Xie et al., 2020; Cui et al., 2023; Chen & Li, 2022; Liu et al., 2021b; Feng et al., 2023; Li et al., 2024), advancing the theoretical understanding of equilibrium learning in standard MARL without robustness considerations.

**Multi-Objective Reinforcement Learning.** Multi-objective reinforcement learning (MORL) is based on multi-objective optimization multi-objective optimization, e.g., (Choo & Atkins, 1983; Steuer, 1986; Geoffrion, 1968; Ehrgott, 2005; Bowman Jr, 1976; Miettinen, 1999; Caramia et al., 2020; Gunantara, 2018; Deb et al., 2016; Giagkiozis & Fleming, 2015; Riquelme et al., 2015; Das & Dennis, 1997; Liu et al., 2021c;a; Chen et al., 2023; Mahapatra et al., 2023; Sener & Koltun, 2018; Klamroth & Jørgen, 2007; Kasimbeyli et al., 2019; Fernando et al., 2022; Hu et al., 2024; Chen et al., 2024; Mahapatra & Rajan, 2020; Xiao et al., 2024; Lin et al., 2024), which have explored various scalarization methods to find a Pareto optimal solution.

Multi-objective optimization is then later extended to the online learning setting, including online convex optimization and bandit problems (Drugan & Nowe, 2013; Yahyaa et al., 2014b; Turgay et al., 2018; Lu et al., 2019; Tekin & Turğay, 2018; Busa-Fekete et al., 2017; Yahyaa et al., 2014a; Jiang et al., 2023a). Extensive studies on MORL are also developed (Roijers et al., 2013; Ehrgott & Wiecek, 2005; Puterman, 1990; Agarwal et al., 2022; Van Moffaert et al., 2013a; Natarajan & Tadepalli, 2005; Wang & Sebag, 2013; Barrett & Narayanan, 2008; Pirotta et al., 2015; Van Moffaert et al., 2013b; Xu et al., 2020; Hayes et al., 2022; Van Moffaert et al., 2013b; Van Moffaert & Nowé, 2014; Chen et al., 2019; Yang et al., 2019; Wiering et al., 2014; Zhu et al., 2023; Wu et al., 2021b; Yu et al., 2021b; Wu et al., 2021a; Zhou et al., 2022; Li et al., 2020; Lu et al., 2022; Qiu et al., 2024), which have studied different scalarization methods. However, we consider multi-agent RL, and results of these studies cannot be directly extended.

**Normal-Form Games with Vectorial Payoffs.** Game theory has developed extensive studies on normal-form games with vector or multi-criteria payoffs, where the notion of PNE is proposed and studied (Blackwell, 1956; Shapley & Rigby, 1959; Krieger, 2003; Voorneveld et al., 1999; Park, 2019; Salonen, 1992; Zapata et al., 2019; Yu & Yuan, 1998; Lozovanu et al., 2005; Sridhar & Mandyam, 2012). However, all of these works are focusing on normal-form games, which do not extend to Markov games with sequential decisions.

## B   NUMERICAL EXPERIMENTS

In this section, we use a simplified MOMG ($N = 2, H = 1$) to numerically validate our theoretical results.

## B.1 Environment Setup

We consider a two-player, two-action game extended from the matching pennies game (Gibbons, 1992). Player 1 can take action $A/B$ and Player 2 can take $C/D$. For each joint action, the vectorial payoff matrix is specified as below:

Table 1: Payoff Bi-Matrix for Multi-Objective Matching Pennies

|              | P2: Action C        | P2: Action D        |
|--------------|---------------------|---------------------|
| **P1: Action A** | $([5,1],[0,0])$ | $([0,0],[5,1])$ |
| **P1: Action B** | $([0,0],[1,5])$ | $([1,5],[0,0])$ |

## B.2 PNE Front Derivation

We then derive the PNE front of this game through linear scalarization. Let $p$ be the probability P1 plays Action A, and $q$ be the probability P2 plays Action C, thus the joint policy is specified by $(p,q)$.

Given any preference $\Lambda = \{\lambda_1 = (\alpha, 1-\alpha), \lambda_2 = (\beta, 1-\beta)\}$, Player 1's scalarized payoff is $S_1 = \alpha r_1 + (1-\alpha)r_2$; And Player 2's scalarized payoff is $S_2 = \beta r_1 + (1-\beta)r_2$. Then for each joint action, the scalarized payoffs are:

- **(A, C):** $S_1 = \alpha(5) + (1-\alpha)(1) = 4\alpha + 1$, $S_2 = \beta(0) + (1-\beta)(0) = 0$;
- **(A, D):** $S_1 = \alpha(0) + (1-\alpha)(0) = 0$, $S_2 = \beta(5) + (1-\beta)(1) = 4\beta + 1$;
- **(B, C):** $S_1 = \alpha(0) + (1-\alpha)(0) = 0$, $S_2 = \beta(1) + (1-\beta)(5) = 5 - 4\beta$;
- **(B, D):** $S_1 = \alpha(1) + (1-\alpha)(5) = 5 - 4\alpha$, $S_2 = \beta(0) + (1-\beta)(0) = 0$.

This yields the scalarized game matrix:

|                        | P2: Action C $(q)$   | P2: Action D $(1-q)$ |
|------------------------|----------------------|----------------------|
| P1: Action A $(p)$     | $(4\alpha + 1, 0)$   | $(0, 4\beta + 1)$    |
| P1: Action B $(1-p)$   | $(0, 5 - 4\beta)$    | $(5 - 4\alpha, 0)$   |

The equilibrium $(p^*, q^*)$ is found when both players are indifferent between their actions, and we can show that the mixed strategy Nash Equilibrium is:

- Player 1's Strategy: $p^* = \frac{5-4\beta}{6}$,
- Player 2's Strategy: $q^* = \frac{5-4\alpha}{6}$,

where $\alpha, \beta \in (0,1)$ are the players' respective preference weights.

By Theorem 2, the whole PNE set of the game is then

$$\left\{ \left( \frac{5-4\beta}{6}, \frac{5-4\alpha}{6} \right) : (\alpha, \beta) \in (0,1)^2 \right\}, \tag{8}$$

and the Pareto-Nash front (for Player 1) is

$$\left\{ \left[ \frac{5(5-4\alpha)(5-4\beta)}{36} + \frac{(1+4\alpha)(1+4\beta)}{36}, \frac{(5-4\alpha)(5-4\beta)}{36} + \frac{5(1+4\alpha)(1+4\beta)}{36} \right] : \alpha, \beta \in (0,1) \right\}. \tag{9}$$

If we set $\alpha, \beta \in [0,1]$, then we can obtain the set of WPNE and weakly Pareto-Nash front, per Theorem 3.

We numerically plot the PN front in Figure 1.

**Remark 4.** *As a quick sanity check, we check the point $(1,1)$ lies in the Pareto-Nash front of the MOMG. We can verify that when the preference is $\lambda_1 = \lambda_2 = (1/3, 2/3)$, the scalarized game has a NE of $\left( \frac{3-\sqrt{3}}{6}, \frac{3+\sqrt{3}}{6} \right)$, at which both players' payoff vectors are $[1,1]$. Hence it suffices to verify*

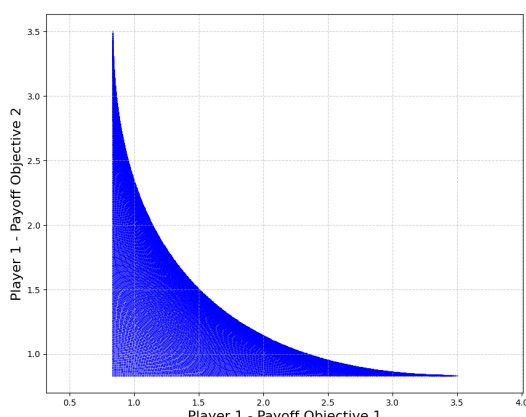

Figure 1: Pareto-Nash Front of Player 1

*the policy $\left(\frac{3-\sqrt{3}}{6}, \frac{3+\sqrt{3}}{6}\right)$ is a PNE. Specifically, when Player 2 fixes its policy, while Player 1 take a policy $p$ (i.e., takes $A$ with probability $p$ and takes $B$ with probability $1-p$), then the payoff vector of Player 1 is*

$$\mathbf{r}_1\left(p, \frac{3+\sqrt{3}}{6}\right) = \left[\frac{3-\sqrt{3}}{6} + (2+\sqrt{3})p, \frac{15-5\sqrt{3}}{6} + (\sqrt{3}-2)p\right]. \tag{10}$$

*Clearly, there does not exists any $p \in [0,1]$, such that $\mathbf{r}_1\left(p, \frac{3+\sqrt{3}}{6}\right) \geq [1,1]$, i.e., Player 1 cannot simultaneously improve both objectives.*

*Similarly, for Player 2,*

$$\mathbf{r}_2\left(\frac{3-\sqrt{3}}{6}, q\right) = \left[\frac{15-5\sqrt{3}}{6} + (\sqrt{3}-2)q, \frac{3-3\sqrt{3}}{6} + (\sqrt{3}+2)q\right], \tag{11}$$

*and there is no policy $y$ to improve both objectives. Thus, $\left(\frac{3-\sqrt{3}}{6}, \frac{3+\sqrt{3}}{6}\right)$ is a PNE.*

### B.3 EXPERIMENTAL VALIDATION: TWO-PHASE LEARNING RECOVERS THE (W)PNE FRONT

We empirically validate that our two-phase procedure in Algorithm 1 converges to the entire Pareto–Nash front. The experiment isolates the two roles of our method: (i) preference-free model estimation from data and (ii) preference-conditioned planning by solving linearly scalarized games on the learned model.

Phase 1: Model estimation. We implement the first stage of Algorithm 1 to estimate the reward model from rollouts. At each time step, every observed reward component is perturbed with independent Gaussian noise $\mathcal{N}(0, 0.1)$. For a growing data budget $N$, we re-estimate the reward model from the first $N$ samples, with $N = 10n$ for $n = 1, ..., 500$.

Phase 2: Planning via linear scalarization. Given an estimated model and a preference profile $\mathbf{\Lambda}$, we solve the linearly scalarized game to a Nash equilibrium and record the induced vector payoff. Specifically, for each $N$, under the reward model estimated with these $N$ samples, we compute the NE $(p_N^*, q_N^*)$ of the scalarized game on the learned model and log Player 1's two-objective payoff.

Visualization. We plot all of Player 1's payoff in Figure 2. The figures present (1). single convergence path produced by running a standard equilibrium solver (e.g., fictitious play or multiplicative-weights) at a fixed preference profile $\mathbf{\Lambda}$ on the learned model, under increasing number of collected samples; And (2). the cloud of payoffs obtained by solving scalarized games for 200 preferences under the final reward model (estimated with 5000 samples).

As $N$ increases, the scatter produced by solving scalarized games under all preferences collapses toward the Pareto-Nash front. These convergent trajectories at a fixed $\mathbf{\Lambda}$ are consistent with our theory

that NE of linearly scalarized games coincide with PNE (strictly positive weights) and with WPNE (non-negative weights); Moreover, With 5000 samples and 200 preferences, the learned model yields a cloud that closely traces the full front, indicating that looping over preferences recovers the entire (W)PNE set from a single learned model. These observations empirically corroborate our theoretical bridge from MOMGs to linearly scalarized games and the effectiveness of our two-phase pipeline.

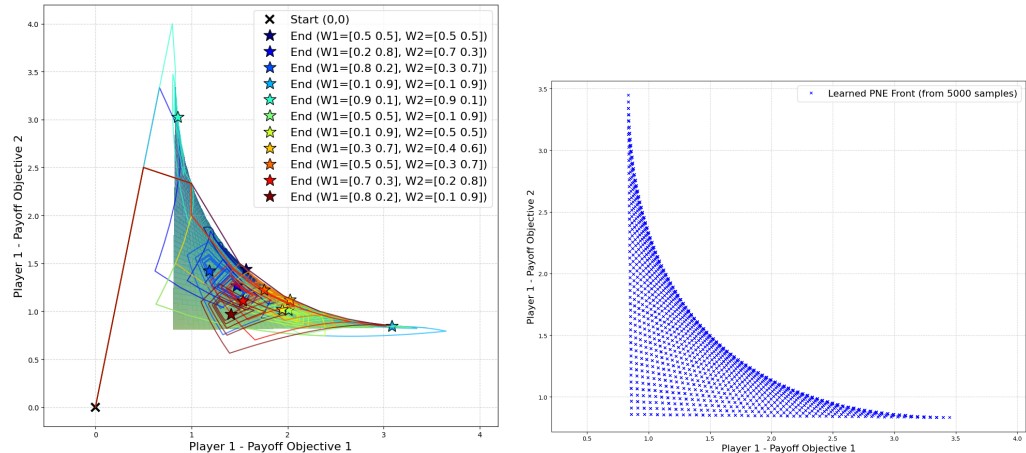

Figure 2: Learning of PNE

## C  PROOFS FOR SECTION 3

### C.1  PNE

**Theorem 8** (Existence of PNE). *For any Multi-Objective Markov Game with a finite number of agents $N$, a finite state space $\mathcal{S}$, and a finite joint action space $\mathcal{A}$, there exists a PNE in the space of stationary stochastic policies.*

*Proof.* We will derive the proof through scalarized approaches.

Let the Multi-Objective Markov Gamebe $G = (\mathcal{N}, \mathcal{S}, \{\mathcal{A}^k\}, H, \{M\}, P, \{r^k\})$, and given a preference vector $\boldsymbol{\Lambda} = \{\boldsymbol{\lambda}^1, \boldsymbol{\lambda}^2, \ldots, \boldsymbol{\lambda}^N\}$, where each $\boldsymbol{\lambda}^k \in \Delta_M^o$, we then construct a corresponding single-objective Markov Game, denoted $G_{\boldsymbol{\Lambda}}$ as follows.

For each agent $k$, define a scalar reward function $U^k : \mathcal{S} \times \mathcal{A} \to [0, 1]$ as the linear combination of its vector rewards:

$$U^k(s, \boldsymbol{a}) \triangleq (\boldsymbol{\lambda}^k)^\top \boldsymbol{r}^k(s, \boldsymbol{a}) = \sum_{i=1}^M \lambda_i^k r_i^k(s, \boldsymbol{a}).$$

We then define Scalarized Value functions as the expected value functions for agent $k$ under a joint policy $\boldsymbol{\pi}$ as $U^{k,\boldsymbol{\pi}}$ as

$$U^{k,\boldsymbol{\pi}}(s_1) \triangleq \mathbb{E}\left[\sum_{h=1}^H U^k(s_h, \boldsymbol{a}_h)\Big|s_1, \boldsymbol{\pi}\right] = (\boldsymbol{\lambda}^k)^\top \boldsymbol{V}_1^{k,\boldsymbol{\pi}}(s_1),$$

where the last equation is due to the linearity of expectation.

Using these scalarized value functions, the resulting game $G_{\boldsymbol{\Lambda}} = (\mathcal{N}, \mathcal{S}, \{\mathcal{A}^k\}, H, P, \{U^k\})$ is a standard $N$-player, general-sum, finite Markov Game. In this game, each agent $k$ aims to find a policy $\pi^k$ that maximizes its individual scalar utility $U^{k,\boldsymbol{\pi}}$. It is shown in (Fink, 1964) that $G_{\boldsymbol{\Lambda}}$ has a NE, which we denote as $\boldsymbol{\pi}^* = (\pi^{1,*}, \ldots, \pi^{N,*})$. We then show that, $\boldsymbol{\pi}^*$ is a PNE.

Since $\boldsymbol{\pi}^*$ is a NE of $G_{\boldsymbol{\Lambda}}$, no agent $k$ can unilaterally deviate to another policy $\hat{\pi}^k$ to improve its scalar utility, i.e.,

$$U^{k,(\pi^{k,*},\boldsymbol{\pi}^{-k,*})}(s_1) \geq U^{k,(\hat{\pi}^k,\boldsymbol{\pi}^{-k,*})}(s_1).$$

Substituting the definition of the scalar utility, this is:

$$(\boldsymbol{\lambda}^k)^\top \boldsymbol{V}_1^{k,(\pi^{k,*},\boldsymbol{\pi}^{-k,*})}(s_1) \geq (\boldsymbol{\lambda}^k)^\top \boldsymbol{V}_1^{k,(\hat{\pi}^k,\boldsymbol{\pi}^{-k,*})}(s_1). \tag{12}$$

Assume that $\boldsymbol{\pi}^*$ is not a PNE of the original MOMG $G$, then there exists at least one agent $j$ who has a unilateral deviation policy $\hat{\pi}^j$ such that its new value vector Pareto dominates its value vector under $\boldsymbol{\pi}^*$:

- $V_{i,1}^{j,(\hat{\pi}^j,\boldsymbol{\pi}^{-j,*})}(s_1) \geq V_{i,1}^{j,(\pi^{j,*},\boldsymbol{\pi}^{-j,*})}(s_1)$ for all objectives $i \in \{1, \ldots, M\}$.

- $V_{i',1}^{j,(\hat{\pi}^j,\boldsymbol{\pi}^{-j,*})}(s_1) > V_{i',1}^{j,(\pi^{j,*},\boldsymbol{\pi}^{-j,*})}(s_1)$ for at least one objective $i' \in \{1, \ldots, M\}$.

Multiply the inequalities above by the corresponding components $\lambda_i^j$ of agent $j$'s preference vector $\boldsymbol{\lambda}^j$ implies that $\lambda_i^j V_{i,1}^{j,(\hat{\pi}^j,\boldsymbol{\pi}^{-j,*})}(s_1) \geq \lambda_i^j V_{i,1}^{j,(\pi^{j,*},\boldsymbol{\pi}^{-j,*})}(s_1)$ for all $i$, and $\lambda_{i'}^j V_{i',1}^{j,(\hat{\pi}^j,\boldsymbol{\pi}^{-j,*})}(s_1) > \lambda_{i'}^j V_{i',1}^{j,(\pi^{j,*},\boldsymbol{\pi}^{-j,*})}(s_1)$, due to the positive values of $\boldsymbol{\Lambda}$.

Summing these inequalities over all objectives $i \in \{1, \ldots, M\}$, the presence of at least one strict inequality means the total sum must be strictly greater:

$$\sum_{i=1}^M \lambda_i^j V_{i,1}^{j,(\hat{\pi}^j,\boldsymbol{\pi}^{-j,*})}(s_1) > \sum_{i=1}^M \lambda_i^j V_{i,1}^{j,(\pi^{j,*},\boldsymbol{\pi}^{-j,*})}(s_1),$$

i.e.,

$$(\boldsymbol{\lambda}^j)^\top \boldsymbol{V}_1^{j,(\hat{\pi}^j,\boldsymbol{\pi}^{-j,*})}(s_1) > (\boldsymbol{\lambda}^j)^\top \boldsymbol{V}_1^{j,(\pi^{j,*},\boldsymbol{\pi}^{-j,*})}(s_1).$$

This is contradict to equation 12, hence it completes the proof. $\qquad\square$

**Theorem 9** (Linear Scalarization and PNE). *Let the policy space for all agents consist of stationary stochastic policies.*

1. *Any NE of the linearly scalarized game with a set of preferences $\{\boldsymbol{\lambda}^k\}_{k\in\mathcal{N}}$, where each preference vector $\boldsymbol{\lambda}^k$ is in the relative interior of the probability simplex (i.e., $\boldsymbol{\lambda}^k \in \Delta_M^o$), is a PNE of the original Multi-Objective Markov Game.*

2. *Conversely, for any PNE $\boldsymbol{\pi}^*$ of the Multi-Objective Markov Game, there exists a set of preferences $\{\boldsymbol{\lambda}^k\}_{k\in\mathcal{N}}$ with $\boldsymbol{\lambda}^k \in \Delta_M^o$ such that $\boldsymbol{\pi}^*$ is a NE of the corresponding linearly scalarized game.*

*Proof.* First, we prove that a Nash Equilibrium of the scalarized game implies a Pareto-Nash Equilibrium (1). We proceed by contradiction. Let $\boldsymbol{\pi}^*$ be a NE of the linearly scalarized game with preferences $\boldsymbol{\Lambda} = \{\boldsymbol{\lambda}^k\}_{k\in\mathcal{N}}$ where each $\boldsymbol{\lambda}^k \in \Delta_M^o$. The NE condition states that for any agent $k$ and any alternative policy $\hat{\pi}^k$, the scalar utility cannot be improved:

$$(\boldsymbol{\lambda}^k)^\top \boldsymbol{V}_1^{k,(\pi^{k,*},\boldsymbol{\pi}^{-k,*})}(s_1) \geq (\boldsymbol{\lambda}^k)^\top \boldsymbol{V}_1^{k,(\hat{\pi}^k,\boldsymbol{\pi}^{-k,*})}(s_1).$$

Now, assume that $\boldsymbol{\pi}^*$ is *not* a PNE. By definition, this means there must exist an agent $j$ and a deviating policy $\hat{\pi}^j$ such that its value vector $\boldsymbol{V}_1^{j,(\hat{\pi}^j,\boldsymbol{\pi}^{-j,*})}(s_1)$ Pareto dominates $\boldsymbol{V}_1^{j,(\pi^{j,*},\boldsymbol{\pi}^{-j,*})}(s_1)$. This dominance implies that $V_{i,1}^{j,(\hat{\pi}^j,\boldsymbol{\pi}^{-j,*})}(s_1) \geq V_{i,1}^{j,(\pi^{j,*},\boldsymbol{\pi}^{-j,*})}(s_1)$ for all objectives $i$, with the inequality being strict for at least one objective $i'$. Since agent $j$'s preference vector $\boldsymbol{\lambda}^j$ has strictly positive components ($\lambda_i^j > 0$ for all $i$), multiplying these component-wise inequalities by $\lambda_i^j$ and summing them results in a strict inequality for the total scalar utility:

$$\sum_{i=1}^M \lambda_i^j V_{i,1}^{j,(\hat{\pi}^j,\boldsymbol{\pi}^{-j,*})}(s_1) > \sum_{i=1}^M \lambda_i^j V_{i,1}^{j,(\pi^{j,*},\boldsymbol{\pi}^{-j,*})}(s_1).$$

This is equivalent to $(\boldsymbol{\lambda}^j)^\top \boldsymbol{V}_1^{j,(\hat{\pi}^j,\boldsymbol{\pi}^{-j,*})}(s_1) > (\boldsymbol{\lambda}^j)^\top \boldsymbol{V}_1^{j,(\pi^{j,*},\boldsymbol{\pi}^{-j,*})}(s_1)$, which directly contradicts the initial NE condition. Therefore, the assumption must be false, and $\boldsymbol{\pi}^*$ must be a PNE.

Next, we prove that a Pareto-Nash Equilibrium implies a Nash Equilibrium under some scalarization (2). Let $\boldsymbol{\pi}^* = (\pi^{1,*}, \ldots, \pi^{N,*})$ be a PNE. We must construct a set of preferences $\boldsymbol{\Lambda}$ for which $\boldsymbol{\pi}^*$ is a NE. The PNE condition implies that for any agent $k$, no unilateral deviation can lead to a Pareto-dominant outcome. Consider the set of all achievable value vectors for an arbitrary agent $k$, given that other agents play their fixed policies $\boldsymbol{\pi}^{-k,*}$:

$$\mathbb{V}^k(\boldsymbol{\pi}^{-k,*}) := \{\boldsymbol{V}_1^{k,(\pi^k,\boldsymbol{\pi}^{-k,*})}(s_1)|\pi^k \in \Pi^k\}.$$

By Lemma 1, this set $\mathbb{V}^k(\boldsymbol{\pi}^{-k,*})$ is a convex polytope. The PNE condition means that the value vector $\boldsymbol{V}_1^{k,\boldsymbol{\pi}^*}(s_1)$ lies on the Pareto front of this convex polytope.

From Proposition 4, any point on the Pareto front of a convex set can be supported by a hyperplane with a normal vector that has strictly positive components. Thus, there exists a vector $\boldsymbol{\lambda}^k \in \Delta_M^o$ such that for all achievable value vectors $\boldsymbol{v} \in \mathbb{V}^k(\boldsymbol{\pi}^{-k,*})$, the following holds:

$$(\boldsymbol{\lambda}^k)^\top \boldsymbol{V}_1^{k,\boldsymbol{\pi}^*}(s_1) \geq (\boldsymbol{\lambda}^k)^\top \boldsymbol{v}.$$

This is precisely the NE condition for agent $k$: given the other players' strategies, its policy $\pi^{k,*}$ maximizes its own scalar utility defined by $\boldsymbol{\lambda}^k$. Since this construction is possible for every agent $k \in \mathcal{N}$, we can find a set of preferences $\boldsymbol{\Lambda} = \{\boldsymbol{\lambda}^1, \ldots, \boldsymbol{\lambda}^N\}$ for which the PNE $\boldsymbol{\pi}^*$ is a Nash Equilibrium. $\qquad\square$

**Lemma 1.** *For any agent $k$ in a Multi-Objective Markov Game, and for any fixed joint policy $\boldsymbol{\pi}^{-k}$ of the other agents, the set of achievable value vectors for agent $k$, denoted $\mathbb{V}^k(\boldsymbol{\pi}^{-k})$, is a convex polytope in $\mathbb{R}^M$, assuming the policy space $\Pi^k$ is the set of stationary stochastic policies.*

*Proof.* Fix an agent $k$ and the stationary stochastic policies of all other agents, $\boldsymbol{\pi}^{-k}$. From agent $k$'s perspective, the other agents become part of a stationary environment. We can define an "effective" single-agent Multi-Objective MDP that agent $k$ is facing. This effective MDP has the same state space $\mathcal{S}$ and agent $k$'s action space $\mathcal{A}^k$. The effective transition probability for agent $k$ taking action $a^k \in \mathcal{A}^k$ in state $s \in \mathcal{S}$ is the expectation over the other agents' actions:

$$P'(s'|s,a^k) := \sum_{\boldsymbol{a}^{-k} \in \mathcal{A}^{-k}} \left( \prod_{j \neq k} \pi^j(a^j|s) \right) P(s'|s,(a^k,\boldsymbol{a}^{-k})).$$

Similarly, the effective vector-valued reward for agent $k$ is:

$$\boldsymbol{r}'^k(s,a^k) := \sum_{\boldsymbol{a}^{-k} \in \mathcal{A}^{-k}} \left( \prod_{j \neq k} \pi^j(a^j|s) \right) \boldsymbol{r}^k(s,(a^k,\boldsymbol{a}^{-k})).$$

Now, we consider the set of all valid state-action occupancy measures $\theta^k$ for agent $k$ in this effective MDP. An occupancy measure $\theta^k = (\theta_h^k(s,a^k))_{s \in \mathcal{S}, a^k \in \mathcal{A}^k, h \in [H]}$ is a point in $\mathbb{R}^{S \times |\mathcal{A}^k| \times H}$. As shown in (Qiu et al., 2024), the set of all valid occupancy measures, which we denote $\Theta^k$, is defined by a set of linear equality and inequality constraints:

$$\theta_h^k(s,a^k) \geq 0, \quad \forall h, s, a^k,$$

$$\sum_{s,a^k} \theta_1^k(s,a^k) = 1,$$

$$\sum_{a^k \in \mathcal{A}^k} \theta_{h+1}^k(s',a^k) = \sum_{s \in \mathcal{S}} \sum_{a^k \in \mathcal{A}^k} \theta_h^k(s,a^k) P'(s'|s,a^k), \quad \forall h, s'.$$

This set of linear constraints defines $\Theta^k$ as a convex polytope. The value vector for agent $k$ corresponding to a policy $\pi^k$ (and thus a specific occupancy measure $\theta^k \in \Theta^k$) is given by the linear transformation:

$$\boldsymbol{V}_1^{k,(\pi^k,\boldsymbol{\pi}^{-k})}(s_1) = \sum_{h=1}^{H} \sum_{s \in \mathcal{S}} \sum_{a^k \in \mathcal{A}^k} \theta_h^k(s,a^k) \boldsymbol{r}_h'^k(s,a^k).$$

The set of all achievable value vectors, $\mathbb{V}^k(\boldsymbol{\pi}^{-k})$, is the image of the convex polytope $\Theta^k$ under this linear transformation. A fundamental property of convex geometry states that the linear image of a convex polytope is also a convex polytope. Therefore, $\mathbb{V}^k(\boldsymbol{\pi}^{-k})$ is a convex polytope. $\qquad\square$

**Proposition 4.** *Let $\mathbb{V}^k(\boldsymbol{\pi}^{-k})$ be the convex polytope of achievable value vectors for agent $k$ against fixed opponent policies $\boldsymbol{\pi}^{-k}$. If a value vector $\boldsymbol{v}^* \in \mathbb{V}^k(\boldsymbol{\pi}^{-k})$ is on the Pareto front of this set, then there exists a preference vector $\boldsymbol{\lambda}^k \in \Delta_M^o$ such that $\boldsymbol{v}^*$ maximizes the linear scalarization $(\boldsymbol{\lambda}^k)^\top \boldsymbol{v}$ over all $\boldsymbol{v} \in \mathbb{V}^k(\boldsymbol{\pi}^{-k})$.*

*Proof.* Let $\boldsymbol{v}^*$ be a point on the Pareto front of the convex set $\mathbb{V}^k(\boldsymbol{\pi}^{-k})$. By definition, no other point $\boldsymbol{v} \in \mathbb{V}^k(\boldsymbol{\pi}^{-k})$ Pareto dominates $\boldsymbol{v}^*$.

Consider the set $P(\boldsymbol{v}^*) = \{\boldsymbol{v} \in \mathbb{R}^M \mid v_i > v_i^* \text{ for all } i \in \{1, \ldots, M\}\}$. This is the convex set of points that strictly Pareto dominate $\boldsymbol{v}^*$. The Pareto optimality of $\boldsymbol{v}^*$ implies that the interior of $\mathbb{V}^k(\boldsymbol{\pi}^{-k})$ and the set $P(\boldsymbol{v}^*)$ are disjoint. By the Separating Hyperplane Theorem, there exists a non-zero vector $\boldsymbol{\lambda}^k \in \mathbb{R}^M$ and a scalar $c$ that define a hyperplane separating them, such that $(\boldsymbol{\lambda}^k)^\top \boldsymbol{v} \leq c$ for all $\boldsymbol{v} \in \mathbb{V}^k(\boldsymbol{\pi}^{-k})$ and $(\boldsymbol{\lambda}^k)^\top \boldsymbol{u} \geq c$ for all $\boldsymbol{u}$ in the closure of $P(\boldsymbol{v}^*)$. Since $\boldsymbol{v}^*$ is on the boundary of both sets, we have $(\boldsymbol{\lambda}^k)^\top \boldsymbol{v}^* = c$. This shows the hyperplane supports the set $\mathbb{V}^k(\boldsymbol{\pi}^{-k})$ at point $\boldsymbol{v}^*$.

Now, we show that the components of $\boldsymbol{\lambda}^k$ are all strictly positive. First, all components must be non-negative ($\lambda_i^k \geq 0$). If any component $\lambda_j^k$ were negative, we could make the $j$-th component of a point $\boldsymbol{u} \in P(\boldsymbol{v}^*)$ arbitrarily large, causing $(\boldsymbol{\lambda}^k)^\top \boldsymbol{u} \to -\infty$, which violates the separating condition. Furthermore, all components must be strictly positive ($\lambda_i^k > 0$). Assume for contradiction that some component $\lambda_j^k = 0$. Because $\boldsymbol{v}^*$ is on the Pareto front of a convex body, it cannot be dominated. However, if $\lambda_j^k = 0$, the supporting hyperplane $(\boldsymbol{\lambda}^k)^\top \boldsymbol{v} = c$ is parallel to the $v_j$ axis. This would allow for another point $\boldsymbol{v}' \in \mathbb{V}^k(\boldsymbol{\pi}^{-k})$ to exist with $v_j' > v_j^*$ and $v_i' \geq v_i^*$ for $i \neq j$ while still satisfying the hyperplane condition, contradicting that $\boldsymbol{v}^*$ is on the Pareto front. Therefore, all components of $\boldsymbol{\lambda}^k$ must be strictly positive.

Since all $\lambda_i^k > 0$, we can normalize the vector by dividing by its L1-norm to ensure its components sum to 1, placing it in the relative interior of the simplex, $\Delta_M^o$. The supporting hyperplane condition $(\boldsymbol{\lambda}^k)^\top \boldsymbol{v}^* \geq (\boldsymbol{\lambda}^k)^\top \boldsymbol{v}$ for all $\boldsymbol{v} \in \mathbb{V}^k(\boldsymbol{\pi}^{-k})$ is precisely the statement that $\boldsymbol{v}^*$ maximizes the linear scalarization. $\square$

**Proposition 5.** *A policy profile $\pi$ is a Pareto-Nash Equilibrium if and only if its Multi-Agent Strict PNG is zero.*

$$PNG(\pi) = 0 \Leftarrow \pi \text{ is a PNE}$$

*Proof.* The proof consists of two parts, showing both directions of the equivalence.

**Part 1: If $\pi$ is a PNE, then PNG$(\pi) = 0$ ($\Leftarrow$).**

By the definition of a PNE, for any agent $k$ and any of their unilateral deviation policies $\pi_k'$, the resulting value vector $V_{k,1}^{(\pi_k', \pi_{-k})}(s_1)$ does not Pareto dominate $V_{k,1}^\pi(s_1)$. We then denote the difference vector for a deviation by $d = V_{k,1}^{(\pi_k', \pi_{-k})}(s_1) - V_{k,1}^\pi(s_1)$. The no-dominance condition means that it is *not* the case that ($d_i \geq 0$ for all objectives $i$ AND $d_j > 0$ for at least one objective $j$). This leaves two possibilities for the vector $d$:

(a) There is at least one component $j$ for which $d_j < 0$.

(b) All components are zero, $d = 0$. This occurs for the trivial deviation where $\pi_k' = \pi_k$.

We now evaluate $\inf_{\lambda^k \in \Delta_M^o} (\lambda^k)^\top d$ for both cases.

- In Case (a), since there is a negative component $d_j < 0$ and $d$ is bounded (by $H$). Thus there exists $\lambda \in \Delta_M^o$ so that $\lambda d \leq 0$;

- In Case (b), the dot product is clearly zero.

In both possible cases, the infimum is non-positive.

Since for any possible deviation $\pi'_k$, the inner infimum term is non-positive, the supremum over all such deviations must also be non-positive:

$$\sup_{\pi'_k} \inf_{\lambda^k \in \Delta^o_M} (\lambda^k)^\top (V^{(\pi'_k, \pi_{-k})}_{k,1}(s_1) - V^\pi_{k,1}(s_1)) \leq 0.$$

However, the $\sup = 0$ can be attained at $\pi'_k = \pi_k$, hence it completes the first part.

**Part 2: If PNG$(\pi) = 0$, then $\pi$ is not necessarily a PNE ($\nRightarrow$).**

We assume that PNG$(\pi) = 0$, which implies that for every agent $k$:

$$\sup_{\pi'_k} \inf_{\lambda^k \in \Delta^o_M} (\lambda^k)^\top (V^{(\pi'_k, \pi_{-k})}_{k,1}(s_1) - V^\pi_{k,1}(s_1)) \leq 0.$$

This means that for any possible unilateral deviation $\pi'_k$, the inner infimum must be non-positive:

$$\inf_{\lambda^k \in \Delta^o_M} (\lambda^k)^\top (V^{(\pi'_k, \pi_{-k})}_{k,1}(s_1) - V^\pi_{k,1}(s_1)) \leq 0.$$

We additional assume, for the sake of contradiction, that $\pi$ is **not** a PNE. Then by definition, there must exist at least one agent $k$ and a deviating policy $\pi'_k$ such that $V^{(\pi'_k, \pi_{-k})}_{k,1}(s_1)$ Pareto dominates $V^\pi_{k,1}(s_1)$. Let $d = V^{(\pi'_k, \pi_{-k})}_{k,1}(s_1) - V^\pi_{k,1}(s_1)$. By the definition of Pareto dominance, we have $d_i \geq 0$ for all $i$ and $d_j > 0$ for at least one $j$.

However, since $\Delta^o_M$ is an open set, the $\inf$ over it can be 0, hence the proof cannot be established.

$\square$

### C.2 WPNE

**Theorem 10.** *The set of all Weak Pareto-Nash Equilibria in a Multi-Objective Markov Game $G$ is equivalent to the union of all Nash Equilibria of linear scalarizations with non-negative preferences:*

$$\textbf{WPNE}(G) = \bigcup_{\Lambda \in (\Delta_M)^N} \textbf{NE}(G_\Lambda).$$

*Proof.* The proof requires showing inclusion in both directions.

We first show that any NE of a non-negatively scalarized game is a WPNE: $\bigcup_{\Lambda \in (\Delta_M)^N} \textbf{NE}(G_\Lambda) \subseteq \textbf{WPNE}(G)$.

Let $\pi^*$ be a Nash Equilibrium of a scalarized game $G_\Lambda$ for some preference profile $\Lambda = \{\lambda^1, \ldots, \lambda^N\}$ where each $\lambda^k \in \Delta_M$. By the definition of a Nash Equilibrium, for any agent $k$ and any unilateral deviation $\pi'_k$, agent $k$ cannot improve their scalarized payoff. That is:

$$(\lambda^k)^\top V^{\pi^*}_{k,1}(s_1) \geq (\lambda^k)^\top V^{(\pi'_k, \pi^*_{-k})}_{k,1}(s_1).$$

We then assume that $\pi^*$ is **not** a WPNE. By Definition 2, this means there must exist at least one agent $k$ and a deviation policy $\pi'_k$ that achieves a *strictly dominating* value vector. This implies:

$$V^{(\pi'_k, \pi^*_{-k})}_{k,i}(s_1) > V^{\pi^*}_{k,i}(s_1) \quad \text{for all objectives } i \in [M].$$

Let $d = V^{(\pi'_k, \pi^*_{-k})}_{k,1} - V^{\pi^*}_{k,1}$. From the step above, this vector $d$ has all components strictly positive ($d_i > 0$ for all $i$). Since the preference vector $\lambda^k \in \Delta_M$ is non-zero (it sums to 1) and all its components are non-negative ($\lambda^k_i \geq 0$). This means at least one component $\lambda^k_j$ must be strictly positive. Consider the dot product $(\lambda^k)^\top d = \sum_{i=1}^M \lambda^k_i d_i$. Since all $d_i > 0$ and all $\lambda^k_i \geq 0$, and at least one $\lambda^k_j > 0$, the sum must be strictly positive:

$$(\lambda^k)^\top d = \sum_{i=1}^M \underbrace{\lambda^k_i}_{\geq 0} \underbrace{d_i}_{> 0} > 0.$$

(The sum is strictly positive because the term $\lambda_j^k d_j$ is strictly positive, and all other terms are non-negative).

This implies:

$$(\lambda^k)^\top (V_{k,1}^{(\pi_k', \pi_{-k}^*)} - V_{k,1}^{\pi^*}) > 0 \implies (\lambda^k)^\top V_{k,1}^{(\pi_k', \pi_{-k}^*)} > (\lambda^k)^\top V_{k,1}^{\pi^*},$$

which contradicts the NE condition.

We then show the other direction, that any WPNE is an NE for at least one non-negative scalarization: $\textbf{WPNE}(G) \subseteq \bigcup_{\Lambda \in (\Delta_M)^N} \textbf{NE}(G_\Lambda)$.

Consider $\pi^*$ to be a Weak Pareto-Nash Equilibrium, and we aim to construct a preference profile $\Lambda \in (\Delta_M)^N$ such that $\pi^*$ is a Nash Equilibrium for the scalarized game $G_\Lambda$.

We first consider any agent $k$ and fix the policies of all other agents to $\pi_{-k}^*$. Let $\mathbb{V}^k(\pi_{-k}^*) = \{V_{k,1}^{(\pi_k', \pi_{-k}^*)}(s_1) \mid \pi_k' \in \Pi_k\}$ be the set of all value vectors agent $k$ can achieve by unilaterally deviating. As shown in Lemma 1, this set is a convex polytope.

We then note that the fact that $\pi^*$ is a WPNE means that the value vector $v^* = V_{k,1}^{\pi^*}(s_1)$ is a weakly Pareto optimal point of this convex set $\mathbb{V}^k(\pi_{-k}^*)$, as no other point $v' \in \mathbb{V}^k(\pi_{-k}^*)$ (achieved by some deviation $\pi_k'$) exists such that $v' > v^*$ (strictly dominates). Then from Prop. 4.2 in Qiu et al. (2024), any weakly Pareto optimal point of $\mathbb{V}^k$ can be supported by a non-trivial, non-negative hyperplane. This implies that there exists a non-zero vector $\lambda^k \geq 0$ (which can be normalized such that $\lambda^k \in \Delta_M$) that defines a supporting hyperplane at $v^*$, such that:

$$(\lambda^k)^\top v^* \geq (\lambda^k)^\top v \quad \text{for all } v \in \mathbb{V}^k(\pi_{-k}^*).$$

Substituting the definitions of $v^*$ and $v$ back into this inequality, we get:

$$(\lambda^k)^\top V_{k,1}^{\pi^*}(s_1) \geq (\lambda^k)^\top V_{k,1}^{(\pi_k', \pi_{-k}^*)}(s_1) \quad \text{for all deviations } \pi_k'.$$

This implies that $\pi_k^*$ is a best response to $\pi_{-k}^*$ in the single-objective game scalarized by $\lambda^k$.

Since we can perform this construction for *every* agent $k \in [N]$, we have found a set of preference vectors $\Lambda = \{\lambda^1, \ldots, \lambda^N\}$, with each $\lambda^k \in \Delta_M$, such that $\pi^*$ is a Nash Equilibrium of the scalarized game $G_\Lambda$. This shows that $\pi^* \in \textbf{NE}(G_\Lambda)$ for a constructed $\Lambda \in (\Delta_M)^N$. Therefore, $\textbf{WPNE}(G) \subseteq \bigcup_{\Lambda \in (\Delta_M)^N} \textbf{NE}(G_\Lambda)$.

Combining Part 1 and Part 2, we proved the equivalence. $\square$

**Theorem 11.** *A policy profile $\pi$ is a Weak Pareto-Nash Equilibrium if and only if its Multi-Agent Weak PNG is zero.*

$$WPNG(\pi) = 0 \iff \pi \text{ is a WPNE}.$$

*Proof.* We prove this equivalence in two parts.

**Part 1: If $\pi$ is a WPNE, then WPNG$(\pi) = 0$ ($\Leftarrow$).**

Let $\pi$ be a Weak Pareto-Nash Equilibrium (WPNE), then for any agent $k$ and any unilateral deviation policy $\pi_k'$, it is **not** the case that the resulting value vector $V_{k,1}' = V_{k,1}^{(\pi_k', \pi_{-k})}(s_1)$ strictly dominates the original value $V_{k,1}^\pi(s_1)$. Hence for any deviation $\pi_k'$, the difference vector $d = V_{k,1}' - V_{k,1}^\pi$ must have at least one non-positive component. That is, there must exist at least one objective $j \in [M]$ such that $d_j \leq 0$. (If all components were strictly positive, $d_i > 0$ for all $i$, this would be strict dominance, which is ruled out by the WPNE definition).

We now evaluate the term $\inf_{\lambda \in \Delta_M} \lambda^\top d$. Since the vector $d$ has at least one non-positive component $d_j \leq 0$, we can choose a specific vector $\lambda^* \in \Delta_M$ that places all of its weight on that single component (i.e., set $\lambda_j^* = 1$ and $\lambda_i^* = 0$ for all $i \neq j$). This $\lambda^*$ is a valid point in the closed simplex $\Delta_M$.

This specific choice of $\lambda^*$ yields the dot product:

$$(\lambda^*)^\top d = 1 \cdot d_j + \sum_{i \neq j} 0 \cdot d_i = d_j \leq 0.$$

Since we found a valid $\lambda \in \Delta_M$ that results in a non-positive dot product, the infimum over all possible $\lambda \in \Delta_M$ must also be non-positive.

$$\inf_{\lambda \in \Delta_M} \lambda^\top d \leq 0.$$

This inequality holds for *every* possible deviation $\pi'_k$ for agent $k$. Therefore, the supremum over all deviations is also non-positive:

$$\sup_{\pi'_k} \left( \inf_{\lambda \in \Delta_M} (\lambda)^\top (V'_{k,1} - V^\pi_{k,1}) \right) \leq 0.$$

This holds for all agents $k$. The WPNG is the maximum of these non-positive values, so $\mathrm{WPNG}(\pi) \leq 0$. Since the gap can never be negative (a player can always choose the trivial deviation $\pi'_k = \pi_k$, which yields $d = 0$ and a gap of 0), we must have $\mathrm{WPNG}(\pi) = 0$.

**Part 2: If WPNG$(\pi) = 0$, then $\pi$ is a WPNE** $(\Rightarrow)$.

We then prove this direction by contradiction. Assume $\mathrm{WPNG}(\pi) = 0$, but the policy $\pi$ is **not** a WPNE. If $\pi$ is not a WPNE, then there must exist at least one agent $k$ and a specific deviating policy $\pi'_k$ that achieves a *strictly dominating* value vector. That is:

$$\exists k, \pi'_k \quad \text{s.t.} \quad V^{(\pi'_k, \pi_{-k})}_{k,i}(s_1) > V^\pi_{k,i}(s_1) \quad \text{for all } i \in [M].$$

Let $d = V^{(\pi'_k, \pi_{-k})}_{k,1}(s_1) - V^\pi_{k,1}(s_1)$. From the step above, this vector $d$ has all components strictly positive: $d_i > 0$ for all $i \in [M]$.

We now evaluate $\inf_{\lambda \in \Delta_M} \lambda^\top d$ for this specific vector $d$. The term $\lambda^\top d = \sum_i \lambda_i d_i$ is a convex combination of the components of $d$. Since every $d_i$ is strictly positive, any convex combination of them will also be strictly positive. The infimum (which is a minimum, as $\Delta_M$ is a compact set) is achieved by placing all weight on the smallest component of $d$:

$$\inf_{\lambda \in \Delta_M} \lambda^\top d = \min_{i \in [M]} \{d_i\}.$$

Since all $d_i > 0$, their minimum must also be a strictly positive constant. Let this minimum be $\delta = \min_i \{d_i\} > 0$.

We have found a specific agent $k$ and a specific deviation $\pi'_k$ that yields a gap value of $\delta > 0$. The supremum over *all* deviations for this agent $k$ must be at least this large:

$$\sup_{\pi^*_k} \left( \inf_{\lambda \in \Delta_M} (\lambda)^\top (V^{(\pi^*_k, \pi_{-k})}_{k,1} - V^\pi_{k,1}) \right) \geq \delta > 0.$$

The WPNG is the maximum of this gap value over all agents. Since one agent's gap is strictly positive, the maximum must also be strictly positive:

$$\mathrm{WPNG}(\pi) \geq \delta > 0.$$

It hence completes the proof. $\qquad\square$

### C.3  UTILITY-BASED

**Proposition 6.** *An ESR-NE always exists, and a SER-NE may not exist.*

*Proof.* The proof can be similarly obtained by noting that the ESR results in a single-object Markov game, whose NE always exists. $\qquad\square$

**Proposition 7.** *There exists a single-agent multi-objective MDP, whose ESR-NE is not Pareto optimal.*

*Proof.* We construct a courter-example to show this. Consider a Multi-Objective MDP with a single-state $(S = 1)$ and single-horizon $(H = 1)$. The agent must choose one of three actions, $\{a_1, a_2, a_3\}$. The action choice results in a 2-dimensional reward vector $(m = 2)$:

- Action $a_1$ yields reward vector $r(a_1) = (3, 3)$.

- Action $a_2$ yields reward vector $r(a_2) = (1, 10)$.

- Action $a_3$ yields reward vector $r(a_3) = (10, 1)$.

We consider the utility function $u(\boldsymbol{r}) = \min\{r_1, r_2\}$, under which the ESR-NE (or the optimal policy) is $a_1$. However, $a_1$ is not Pareto optimal, as the policy $\pi = (0, 0.5, 0.5)$ Pareto dominates it. Hence the ESR-NE may not be Pareto optimal in this example, which completes the proof. $\square$

**Proposition 8.** *A SER-NE exists when continuous and quasi-concave utility functions are used.*

*Proof.* A SER-NE of the original MOMG is a standard Nash Equilibrium of an associated scalarized game, $G'$, with strategy spaces $\{\Pi_k\}_{k \in \mathcal{N}}, \Pi_k = \prod_{t=1}^{H} \prod_{s \in \mathcal{S}} \Delta(\mathcal{A}_k)$ and payoff functions $\{U_k\}_{k \in \mathcal{N}}, U_k(\boldsymbol{\pi}) = u_k\left(\boldsymbol{V}_{k,1}^{\boldsymbol{\pi}}(s_1)\right)$ where $\boldsymbol{\pi} = (\pi_1, \dots, \pi_N) \in \prod_k \Pi_k$.

We utilize the Glicksberg's theorem (Fudenberg & Tirole, 1991; Röpke et al., 2022) to show that a Nash Equilibrium exists in $G'$, which is also a SER-NE. To apply Glicksberg's theorem, we need to verify three conditions as follows.

We first verify that the strategy spaces $\Pi_k$ are non-empty, compact, and convex. First note that the set $\Delta(\mathcal{A}_k)$ (the probability simplex) is a convex set, and the total strategy space $\Pi_k$ is a finite product of these convex sets, hence also convex; Moreover, as the simplex $\Delta(\mathcal{A}_k)$ is a closed and bounded subset of a finite-dimensional Euclidean space $(\mathbb{R}^{|\mathcal{A}_k|})$, so it is compact by the Heine-Borel theorem. The total strategy space $\Pi_k$ is a finite product of these compact sets. By Tychonoff's theorem, the product of compact sets is compact.

The payoff function is a composition: $U_k(\boldsymbol{\pi}) = u_k \circ \boldsymbol{V}_{k,1}^{\boldsymbol{\pi}}$. As in Lemma 2, the map $\boldsymbol{\pi} \mapsto \boldsymbol{V}_{k,1}^{\boldsymbol{\pi}}(s_1)$ is a multi-linear function of all policy parameters $\{\pi_{i,t}(a_i|s)\}_{i,t,s,a_i}$, which is also continuous; And since the utility function $u_k : \mathbb{R}^{d_k} \to \mathbb{R}$ is continuous, their composition is continuous.

To verify the payoff functions $U_k(\pi_k, \boldsymbol{\pi}_{-k})$ are quasi-concave in $\pi_k$, we first utilize Lemma 2 to show that, for a fixed $\boldsymbol{\pi}_{-k}$, the map $f(\pi_k) = \boldsymbol{V}_{k,1}^{(\pi_k, \boldsymbol{\pi}_{-k})}(s_1)$ is linear (or more precisely, affine) with respect to $\pi_k$. Moreover, since the utility function $u_k$ is quasi-concave, the payoff $U_k(\pi_k, \boldsymbol{\pi}_{-k}) = u_k(f(\pi_k))$ is the composition of a quasi-concave function $u_k$ with a linear function $f$, which is also quasi-concave.

Hence all conditions are met, and Glicksberg's theorem implies the existence of a NE of $G'$, which is also a SER-NE. $\square$

**Lemma 2.** *For a fixed initial state $s_1$ and fixed policies $\boldsymbol{\pi}_{-k}$ for all other agents, the map from agent $k$'s policy $\pi_k$ to their expected return vector $\boldsymbol{V}_{k,1}^{(\pi_k, \boldsymbol{\pi}_{-k})}(s_1)$ is a **multi-linear function** of the policy parameters $\{\pi_{k,t}(a_k|s) \mid t \in [H], s \in \mathcal{S}, a_k \in \mathcal{A}_k\}$. Specifically, for any time $t$ and state $s$, the map is **linear** in the probability vector $\pi_{k,t}(\cdot|s)$.*

*Proof.* We proceed by backward induction on the time step $t$, from $t = H$ down to $t = 1$.

Utilizing the standard Bellman equations for a finite-horizon game implies that

$$\boldsymbol{V}_{k,t}^{\boldsymbol{\pi}}(s) = \sum_{\boldsymbol{a} \in \mathcal{A}} \boldsymbol{\pi}_t(\boldsymbol{a}|s) \boldsymbol{Q}_{k,t}^{\boldsymbol{\pi}}(s, \boldsymbol{a}) \tag{13}$$

$$\boldsymbol{Q}_{k,t}^{\boldsymbol{\pi}}(s, \boldsymbol{a}) = \boldsymbol{r}_{k,t}(s, \boldsymbol{a}) + \gamma \sum_{s' \in \mathcal{S}} P_t(s'|s, \boldsymbol{a}) \boldsymbol{V}_{k,t+1}^{\boldsymbol{\pi}}(s') \tag{14}$$

with the terminal condition $V_{k,H+1}^{\boldsymbol{\pi}}(s) = \mathbf{0}$ for all $s$.

We first consider the base case: $t = H$. For the final time step $H$, the Q-value is just the immediate reward, as $V_{k,H+1}^{\boldsymbol{\pi}} = \mathbf{0}$:

$$Q_{k,H}^{\boldsymbol{\pi}}(s,\boldsymbol{a}) = r_{k,H}(s,\boldsymbol{a}).$$

The value function $V_{k,H}^{\boldsymbol{\pi}}(s)$ is:

$$V_{k,H}^{\boldsymbol{\pi}}(s) = \sum_{\boldsymbol{a} \in \mathcal{A}} \boldsymbol{\pi}_H(\boldsymbol{a}|s) r_{k,H}(s,\boldsymbol{a}).$$

We substitute $\boldsymbol{\pi}_H(\boldsymbol{a}|s) = \pi_{k,H}(a_k|s) \cdot \boldsymbol{\pi}_{-k,H}(\boldsymbol{a}_{-k}|s)$, where $\boldsymbol{\pi}_{-k,H}(\boldsymbol{a}_{-k}|s) = \prod_{j \neq k} \pi_{j,H}(a_j|s)$, and we have that

$$V_{k,H}^{\boldsymbol{\pi}}(s) = \sum_{\boldsymbol{a}=(a_k,\boldsymbol{a}_{-k})} \pi_{k,H}(a_k|s) \boldsymbol{\pi}_{-k,H}(\boldsymbol{a}_{-k}|s) r_{k,H}(s,(a_k,\boldsymbol{a}_{-k})).$$

We can group the terms by agent $k$'s action $a_k$:

$$V_{k,H}^{\boldsymbol{\pi}}(s) = \sum_{a_k \in \mathcal{A}_k} \pi_{k,H}(a_k|s) \left[ \sum_{\boldsymbol{a}_{-k} \in \mathcal{A}_{-k}} \boldsymbol{\pi}_{-k,H}(\boldsymbol{a}_{-k}|s) r_{k,H}(s,(a_k,\boldsymbol{a}_{-k})) \right].$$

We denote the bracketed term as $\overline{r}_{k,H}(s,a_k|\boldsymbol{\pi}_{-k,H})$:

$$\overline{r}_{k,H}(s,a_k|\boldsymbol{\pi}_{-k,H}) \triangleq \sum_{\boldsymbol{a}_{-k} \in \mathcal{A}_{-k}} \boldsymbol{\pi}_{-k,H}(\boldsymbol{a}_{-k}|s) r_{k,H}(s,(a_k,\boldsymbol{a}_{-k})).$$

Then since $\boldsymbol{\pi}_{-k,H}$ is **fixed**, this term $\overline{r}_{k,H}(s,a_k|\boldsymbol{\pi}_{-k,H})$ is a constant vector for any given $(s,a_k)$. Substituting this back, we get:

$$V_{k,H}^{(\pi_k,\boldsymbol{\pi}_{-k})}(s) = \sum_{a_k \in \mathcal{A}_k} \pi_{k,H}(a_k|s) \cdot \overline{r}_{k,H}(s,a_k|\boldsymbol{\pi}_{-k,H}).$$

This is a linear combination of the constant vectors $\overline{r}_{k,H}$ with coefficients $\pi_{k,H}(a_k|s)$. Therefore, $V_{k,H}^{\boldsymbol{\pi}}(s)$ is a **linear function** of the policy probabilities $\pi_{k,H}(\cdot|s)$.

Assume that $V_{k,t+1}^{(\pi_k;\boldsymbol{\pi}_{-k})}(s')$ is multi-linear in the policy parameters $\{\pi_{k,\tau}(\cdot|\cdot)\}_{\tau=t+1}^{H}$. From equation 14, the Q-value at time $t$ is:

$$Q_{k,t}^{\boldsymbol{\pi}}(s,\boldsymbol{a}) = r_{k,t}(s,\boldsymbol{a}) + \gamma \sum_{s' \in \mathcal{S}} P_t(s'|s,\boldsymbol{a}) V_{k,t+1}^{\boldsymbol{\pi}}(s').$$

By our assumption, $V_{k,t+1}^{\boldsymbol{\pi}}(s')$ depends on $\pi_k$ *only* through parameters from time $t+1$ to $H$. Critically, $Q_{k,t}^{\boldsymbol{\pi}}(s,\boldsymbol{a})$ does not depend on $\pi_{k,t}$.

Now we look at the value function $V_{k,t}^{\boldsymbol{\pi}}(s)$ from equation 13:

$$V_{k,t}^{\boldsymbol{\pi}}(s) = \sum_{\boldsymbol{a} \in \mathcal{A}} \boldsymbol{\pi}_t(\boldsymbol{a}|s) Q_{k,t}^{\boldsymbol{\pi}}(s,\boldsymbol{a}).$$

We expand $\boldsymbol{\pi}_t(\boldsymbol{a}|s)$ and group by $a_k$, just as in the base case:

$$V_{k,t}^{\boldsymbol{\pi}}(s) = \sum_{a_k \in \mathcal{A}_k} \pi_{k,t}(a_k|s) \left[ \sum_{\boldsymbol{a}_{-k} \in \mathcal{A}_{-k}} \boldsymbol{\pi}_{-k,t}(\boldsymbol{a}_{-k}|s) Q_{k,t}^{\boldsymbol{\pi}}(s,(a_k,\boldsymbol{a}_{-k})) \right].$$

Let's define the bracketed term as $\overline{Q}_k(s,a_k|\pi_k^{>t},\boldsymbol{\pi}_{-k})$:

$$\overline{Q}_k(s,a_k|\pi_k^{>t},\boldsymbol{\pi}_{-k}) \triangleq \sum_{\boldsymbol{a}_{-k} \in \mathcal{A}_{-k}} \boldsymbol{\pi}_{-k,t}(\boldsymbol{a}_{-k}|s) Q_{k,t}^{\boldsymbol{\pi}}(s,(a_k,\boldsymbol{a}_{-k})).$$

Since $\boldsymbol{\pi}_{-k,t}$ is fixed, and $Q_{k,t}^{\boldsymbol{\pi}}$ does not depend on $\pi_{k,t}$, this entire term $\overline{Q}_k$ **does not depend on** $\pi_{k,t}$. Substituting this back, we get:

$$V_{k,t}^{(\pi_k,\boldsymbol{\pi}_{-k})}(s) = \sum_{a_k \in \mathcal{A}_k} \pi_{k,t}(a_k|s) \cdot \overline{Q}_k(s,a_k|\pi_k^{>t},\boldsymbol{\pi}_{-k}).$$

This shows that $V_{k,t}^{\boldsymbol{\pi}}(s)$ is a **linear function** of the policy probabilities $\pi_{k,t}(\cdot|s)$.

Furthermore, $\overline{Q}_k$ is a linear combination of $Q_{k,t}^{\boldsymbol{\pi}}$ terms. Each $Q_{k,t}^{\boldsymbol{\pi}}$ is, in turn, a linear combination of $V_{k,t+1}^{\boldsymbol{\pi}}$ terms (which are multi-linear in $\{\pi_{k,\tau}\}_{\tau=t+1}^{H}$ by assumption). Since linearity is preserved under addition and scalar multiplication, $\overline{Q}_k$ is multi-linear in $\{\pi_{k,\tau}\}_{\tau=t+1}^{H}$.

Because $V_{k,t}^{\boldsymbol{\pi}}(s)$ is a linear combination of $\overline{Q}_k$ terms (weighted by $\pi_{k,t}$), the full expression for $V_{k,t}^{\boldsymbol{\pi}}(s)$ is **multi-linear** in all policy parameters $\{\pi_{k,\tau}(\cdot|\cdot)\}_{\tau=t}^{H}$. This completes the inductive step.

Therefore, for $t = 1$, the map

$$\pi_k \mapsto V_{k,1}^{(\pi_k, \boldsymbol{\pi}_{-k})}(s_1)$$

is a multi-linear function of all policy parameters in $\pi_k$, which completes the proof. $\qquad\square$

## D  PROOFS FOR SECTION 4.1

---

**Algorithm 2** ONVI-MG: Optimistic Nash Value Iteration for Multi-Objective Games.

---

1: **Input:** Preferences $\boldsymbol{\Lambda} = \{\boldsymbol{\lambda}^k\}_{k \in \mathcal{N}}$, Total episodes $T$, Confidence $\delta$.
2: **Initialize:** Counts $N_h(s, \boldsymbol{a}) \leftarrow 0$ for all $(h, s, \boldsymbol{a})$. Set $U_{H+1}^{k,t}(s) \leftarrow 0$ for all $k, s, t$.
3: **for** $t = 1, \ldots, T$ **do**
4:    # — Backward Induction Planning Phase —
5:    **for** $h = H, \ldots, 1$ **do**
6:      **for** each state $s \in \mathcal{S}$ **do**
7:        Compute bonus terms $\Psi_h^t(s, \cdot)$ and $\Phi_h^t(s, \cdot)$ using counts $N_h^{t-1}(s, \cdot)$.
8:        **for** each agent $k \in \mathcal{N}$ and joint action $\boldsymbol{a} \in \mathcal{A}$ **do**
9:          Estimate scalar reward: $\hat{U}_h^{k,t}(s, \boldsymbol{a}) \leftarrow (\boldsymbol{\lambda}^k)^\top \hat{\boldsymbol{r}}_h^{k,t}(s, \boldsymbol{a})$.
10:          Compute optimistic Q-value $Q_h^{k,t}(s, \boldsymbol{a})$:
11:          $Q_h^{k,t}(s, \boldsymbol{a}) \leftarrow \min\left\{H, \hat{U}_h^{k,t}(s, \boldsymbol{a}) + \Psi_h^t(s, \boldsymbol{a}) + \sum_{s'} \hat{P}_h^t(s'|s, \boldsymbol{a}) U_{h+1}^{k,t}(s') + \Phi_h^t(s, \boldsymbol{a})\right\}$
12:        **end for**
13:        Define a one-shot matrix game at state $s$ with payoff tables $\{Q_h^{k,t}(s, \boldsymbol{a})\}_{k, \boldsymbol{a}}$.
14:        Compute a NE policy $\boldsymbol{\pi}_h^t(\cdot|s)$ for this one-shot game.
15:        Update optimistic value function for each agent $k$: $U_h^{k,t}(s) \leftarrow \mathbb{E}_{\boldsymbol{a} \sim \boldsymbol{\pi}_h^t(\cdot|s)}\left[Q_h^{k,t}(s, \boldsymbol{a})\right]$
16:      **end for**
17:    **end for**
18:    # — Execution and Data Collection Phase —
19:    Execute joint policy $\boldsymbol{\pi}^t = \{\boldsymbol{\pi}_h^t\}_{h=1}^{H}$ for one full episode starting from $s_1$.
20:    Observe trajectory $(s_1, \boldsymbol{a}_1, \ldots, s_H, \boldsymbol{a}_H)$.
21:    **for** $h = 1, \ldots, H$ **do**
22:      $N_h(s_h, \boldsymbol{a}_h) \leftarrow N_h(s_h, \boldsymbol{a}_h) + 1$.
23:    **end for**
24: **end for**
25: **Output:** $\boldsymbol{\pi}^T$.

---

**Lemma 3** (Concentration and Bonus Validity). *Let the bonus terms be* $\Psi_h^t(s, \boldsymbol{a}) = \sqrt{\frac{c_1 \log(NSAHT/\delta)}{N_h^{t-1}(s, \boldsymbol{a}) \vee 1}}$ *and* $\Phi_h^t(s, \boldsymbol{a}) = H\sqrt{\frac{c_2 S \log(NSAHT/\delta)}{N_h^{t-1}(s, \boldsymbol{a}) \vee 1}}$ *for sufficiently large universal constants* $c_1, c_2$. *With probability at least* $1 - \delta$, *for all* $t, h, s, \boldsymbol{a}, k$ *and any value function* $V : \mathcal{S} \to [0, H]$:

$$\left|(\boldsymbol{\lambda}^k)^\top \hat{\boldsymbol{r}}_h^{k,t}(s, \boldsymbol{a}) - (\boldsymbol{\lambda}^k)^\top \boldsymbol{r}_h^k(s, \boldsymbol{a})\right| \le \Psi_h^t(s, \boldsymbol{a}), \tag{15}$$

$$\left|\sum_{s'} (\hat{P}_h^t(s'|s, \boldsymbol{a}) - P_h(s'|s, \boldsymbol{a})) V(s')\right| \le \Phi_h^t(s, \boldsymbol{a}). \tag{16}$$

*Proof.* Let $x = \hat{P}_h^t(\cdot|s, \boldsymbol{a}) - P_h(\cdot|s, \boldsymbol{a})$ and $y = V(\cdot)$. Applying the Hölder's inequality (for vectors $x, y, |\langle x, y \rangle| \leq \|x\|_1 \|y\|_\infty$) gives:

$$\left| \sum_{s' \in \mathcal{S}} (\hat{P}_h^t(s'|s, \boldsymbol{a}) - P_h(s'|s, \boldsymbol{a})) V(s') \right| \leq \|\hat{P}_h^t(\cdot|s, \boldsymbol{a}) - P_h(\cdot|s, \boldsymbol{a})\|_1 \cdot \|V\|_\infty. \qquad (17)$$

Apply Hoeffding's inequality, we then have that with probability at least $1 - \delta'$, for a fixed $(h, s, \boldsymbol{a})$, we have $\|\hat{P}_h(\cdot|s, \boldsymbol{a}) - P_h(\cdot|s, \boldsymbol{a})\|_1 \leq \sqrt{\frac{2S \log(2/\delta')}{N_h(s, \boldsymbol{a}) \vee 1}}$. Taking a union bound over $N \cdot T \cdot H \cdot S \cdot A$ contexts, and setting $\delta' = \frac{\delta}{NTHSA}$, imply that with probability at least $1 - \delta$, the inequality holds for all contexts simultaneously:

$$\log(2/\delta') = \log\left(\frac{2NTHSA}{\delta}\right),$$

$$\implies \|\hat{P}_h^t(\cdot|s, \boldsymbol{a}) - P_h(\cdot|s, \boldsymbol{a})\|_1 \leq \sqrt{\frac{2S \log(2NTHSA/\delta)}{N_h^{t-1}(s, \boldsymbol{a}) \vee 1}}.$$

Now we substitute the bounds for both terms back into Equation equation 17:

$$\left| \sum_{s' \in \mathcal{S}} (\hat{P}_h^t(s'|s, \boldsymbol{a}) - P_h(s'|s, \boldsymbol{a})) V(s') \right| \leq \underbrace{\left( \sqrt{\frac{2S \log(2NTHSA/\delta)}{N_h^{t-1}(s, \boldsymbol{a}) \vee 1}} \right)}_{\text{Bound on } \|\hat{P} - P\|_1} \cdot \underbrace{H}_{\text{Bound on } \|V\|_\infty}$$

$$= \sqrt{\frac{2H^2 S \log(2NTHSA/\delta)}{N_h^{t-1}(s, \boldsymbol{a}) \vee 1}}.$$

This final expression is precisely the definition of our transition bonus $\Phi_h^t(s, \boldsymbol{a})$ (omitting universal constants for clarity). Therefore, we have shown that with high probability, the error in the expected next-state value is contained by the bonus term. $\qquad \square$

**Lemma 4** (Optimism of Value Functions). *Assuming the high-probability event in Lemma 1 holds, then for all $t, k, h, s$ and any joint policy $\boldsymbol{\pi}$, the true value is bounded by the optimistic value: $U_h^{k, \boldsymbol{\pi}}(s) \leq U_h^{k, t}(s)$.*

*Proof.* We prove this by backward induction on $h$.

**Base Case** ($h = H + 1$): $U_{H+1}^{k, \boldsymbol{\pi}}(s) = 0$ and $U_{H+1}^{k, t}(s) = 0$, so the inequality holds.

**Inductive Hypothesis:** Assume $U_{h+1}^{k, \boldsymbol{\pi}}(s) \leq U_{h+1}^{k, t}(s)$ for all $s, k, \boldsymbol{\pi}$.

**Inductive Step (step $h$):** Let $Q_h^{k, \boldsymbol{\pi}}$ be the true scalarized Q-function.

$$Q_h^{k, \boldsymbol{\pi}}(s, \boldsymbol{a}) = (\boldsymbol{\lambda}^k)^\top \boldsymbol{r}_h^k(s, \boldsymbol{a}) + \sum_{s'} P_h(s'|s, \boldsymbol{a}) U_{h+1}^{k, \boldsymbol{\pi}}(s')$$

$$\leq \left[ (\boldsymbol{\lambda}^k)^\top \hat{\boldsymbol{r}}_h^{k, t}(s, \boldsymbol{a}) + \Psi_h^t(s, \boldsymbol{a}) \right] + \sum_{s'} P_h(s'|s, \boldsymbol{a}) U_{h+1}^{k, t}(s') \quad \text{(by equation 15)}$$

$$\leq \left[ (\boldsymbol{\lambda}^k)^\top \hat{\boldsymbol{r}}_h^{k, t}(s, \boldsymbol{a}) + \Psi_h^t(s, \boldsymbol{a}) \right] + \left[ \sum_{s'} \hat{P}_h^t(s'|s, \boldsymbol{a}) U_{h+1}^{k, t}(s') + \Phi_h^t(s, \boldsymbol{a}) \right]$$

$$\leq Q_h^{k, t}(s, \boldsymbol{a}) \quad \text{(by equation 16 and definition of } Q_h^{k, t}).$$

Since $Q_h^{k, \boldsymbol{\pi}}(s, \boldsymbol{a}) \leq Q_h^{k, t}(s, \boldsymbol{a})$ for all $\boldsymbol{a}$, the expected value is also bounded: $U_h^{k, \boldsymbol{\pi}}(s) = \mathbb{E}_{\boldsymbol{a} \sim \boldsymbol{\pi}_h(\cdot|s)}[Q_h^{k, \boldsymbol{\pi}}(s, \boldsymbol{a})] \leq \mathbb{E}_{\boldsymbol{a} \sim \boldsymbol{\pi}_h(\cdot|s)}[Q_h^{k, t}(s, \boldsymbol{a})]$. The value $U_h^{k, t}(s)$ is the NE value of the game with payoffs $Q_h^{k, t}(s, \cdot)$, which must be at least as large as the value from playing any fixed policy $\boldsymbol{\pi}_h(\cdot|s)$, thus completing the induction. $\qquad \square$

**Lemma 5** (Single-Episode Regret Bound). *Assuming the event in Lemma 1 holds, the regret for agent $k$ in episode $t$ is bounded by:*

$$\max_{\pi'^k} U^{k,(\pi'^k, \boldsymbol{\pi}^{-k,t})}(s_1) - U^{k,\boldsymbol{\pi}^t}(s_1) \le 2H \sum_{h=1}^{H} \mathbb{E}_{\boldsymbol{\pi}^t} \left[ \Psi_h^t(s_h, \boldsymbol{a}_h) + \Phi_h^t(s_h, \boldsymbol{a}_h) \right].$$

*Proof.* Let $\pi_t'^{*k}$ be the best response for agent $k$ against $\boldsymbol{\pi}^{-k,t}$. The regret is $U^{k,(\pi_t'^{*k}, \boldsymbol{\pi}^{-k,t})}(s_1) - U^{k,\boldsymbol{\pi}^t}(s_1)$. By Lemma 2, $U^{k,(\pi_t'^{*k}, \boldsymbol{\pi}^{-k,t})}(s_1) \le U_1^{k,t}(s_1)$. Thus, the regret is bounded by $\Delta_1^k :=
U_1^{k,t}(s_1) - U_1^{k,\boldsymbol{\pi}^t}(s_1)$. Let $\Delta_h^k(s) = U_h^{k,t}(s) - U_h^{k,\boldsymbol{\pi}^t}(s)$.

$$\Delta_h^k(s_h) = U_h^{k,t}(s_h) - U_h^{k,\boldsymbol{\pi}^t}(s_h) = \mathbb{E}_{\boldsymbol{a}_h \sim \boldsymbol{\pi}_h^t(\cdot|s_h)} \left[ Q_h^{k,t}(s_h, \boldsymbol{a}_h) - Q_h^{k,\boldsymbol{\pi}^t}(s_h, \boldsymbol{a}_h) \right].$$

$$\le \mathbb{E}_{\boldsymbol{a}_h \sim \boldsymbol{\pi}_h^t(\cdot|s_h)} \left[ 2(\Psi_h^t + \Phi_h^t)(s_h, \boldsymbol{a}_h) + \sum_{s'} P_h(s'|s_h, \boldsymbol{a}_h)\Delta_{h+1}^k(s') \right]$$

$$= \mathbb{E}_{\boldsymbol{a}_h \sim \boldsymbol{\pi}_h^t(\cdot|s_h)} \left[ 2(\Psi_h^t + \Phi_h^t)(s_h, \boldsymbol{a}_h) \right] + \mathbb{E}_{\boldsymbol{\pi}^t} \left[ \Delta_{h+1}^k(s_{h+1})|s_h \right]$$

Unrolling this recursion from $h = 1$ to $H$ (and noting $\Delta_{H+1}^k = 0$) and bounding values by $H$ gives the desired result. $\square$

**Lemma 6** (Total Bonus Bound). *With probability at least $1 - \delta$, the sum of all bonuses encountered is bounded by:*

$$\sum_{t=1}^{T} \sum_{h=1}^{H} \left( \Psi_h^t(s_h^t, \boldsymbol{a}_h^t) + \Phi_h^t(s_h^t, \boldsymbol{a}_h^t) \right) \le \mathcal{O} \left( H^2 S \sqrt{AT \log(SAHT/\delta)} \right).$$

*Proof.* We will prove the bound for each term separately and then combine them.

**Part 1: Bounding the Sum of Reward Bonuses ($\Psi$)**

Let $C_\Psi = \sqrt{c_1 \log(NSAHT/\delta)}$. The reward bonus is $\Psi_h^t(s, \boldsymbol{a}) = C_\Psi / \sqrt{N_h^{t-1}(s, \boldsymbol{a}) \vee 1}$. We want to bound the total sum:

$$S_\Psi = \sum_{t=1}^{T} \sum_{h=1}^{H} \Psi_h^t(s_h^t, \boldsymbol{a}_h^t) = \sum_{t=1}^{T} \sum_{h=1}^{H} \frac{C_\Psi}{\sqrt{N_h^{t-1}(s_h^t, \boldsymbol{a}_h^t) \vee 1}}.$$

Instead of summing over time steps $t$, we regroup the sum by each unique state-joint-action pair $(h, s, \boldsymbol{a})$. Let $N_h^T(s, \boldsymbol{a})$ be the total number of times the pair $(s, \boldsymbol{a})$ was visited at step $h$ over all $T$ episodes. When this pair is visited for the $i$-th time (where $i$ goes from 1 to $N_h^T(s, \boldsymbol{a})$), the count of previous visits is $i - 1$. Thus, the sum can be rewritten as:

$$S_\Psi = \sum_{h=1}^{H} \sum_{s \in \mathcal{S}} \sum_{\boldsymbol{a} \in \mathcal{A}} \sum_{i=1}^{N_h^T(s,\boldsymbol{a})} \frac{C_\Psi}{\sqrt{(i - 1) \vee 1}}.$$

For a fixed $(h, s, \boldsymbol{a})$, we analyze its inner sum. The first term (when $i = 1$) is $C_\Psi / \sqrt{1} = C_\Psi$. For the rest of the terms, we can use the integral bound for a sum of a decreasing function:

$$\sum_{i=1}^{N_h^T(s,\boldsymbol{a})} \frac{1}{\sqrt{(i - 1) \vee 1}} = 1 + \sum_{i=2}^{N_h^T(s,\boldsymbol{a})} \frac{1}{\sqrt{i - 1}}$$

$$\le 1 + \int_1^{N_h^T(s,\boldsymbol{a})} \frac{1}{\sqrt{x}} \, dx$$

$$= 1 + \left[ 2\sqrt{x} \right]_1^{N_h^T(s,\boldsymbol{a})}$$

$$= 1 + 2\sqrt{N_h^T(s, \boldsymbol{a})} - 2 \le 2\sqrt{N_h^T(s, \boldsymbol{a})}.$$

Substituting this back, the total sum is bounded by:

$$S_\Psi \le \sum_{h=1}^{H} \sum_{s \in \mathcal{S}} \sum_{\boldsymbol{a} \in \mathcal{A}} 2C_\Psi \sqrt{N_h^T(s, \boldsymbol{a})}.$$

We use the Cauchy-Schwarz inequality. Let $K = HSA$ be the total number of distinct $(h, s, \boldsymbol{a})$ pairs.

$$\sum_{h,s,\boldsymbol{a}} \sqrt{N_h^T(s, \boldsymbol{a})} \le \sqrt{\sum_{h,s,\boldsymbol{a}} 1^2} \cdot \sqrt{\sum_{h,s,\boldsymbol{a}} \left( \sqrt{N_h^T(s, \boldsymbol{a})} \right)^2} = \sqrt{K} \cdot \sqrt{\sum_{h,s,\boldsymbol{a}} N_h^T(s, \boldsymbol{a})}.$$

The total number of interactions across all episodes is $HT$. Thus, $\sum_{h,s,\boldsymbol{a}} N_h^T(s, \boldsymbol{a}) = HT$. Substituting this in, we get:

$$S_\Psi \le 2C_\Psi \sqrt{HSA} \cdot \sqrt{HT} = 2C_\Psi H \sqrt{SAT}.$$

Finally, substituting the definition of $C_\Psi$:

$$S_\Psi \le \mathcal{O}\left( H \sqrt{SAT \log(NSAHT/\delta)} \right).$$

**Part 2: Bounding the Sum of Transition Bonuses ($\Phi$)**

The procedure for the transition bonus sum is identical, with a different constant. Let $C_\Phi = H\sqrt{c_2 S \log(NSAHT/\delta)}$. The transition bonus is $\Phi_h^t(s, \boldsymbol{a}) = C_\Phi / \sqrt{N_h^{t-1}(s, \boldsymbol{a}) \vee 1}$.

$$S_\Phi = \sum_{t=1}^{T} \sum_{h=1}^{H} \Phi_h^t(s_h^t, \boldsymbol{a}_h^t) = \sum_{t=1}^{T} \sum_{h=1}^{H} \frac{C_\Phi}{\sqrt{N_h^{t-1}(s_h^t, \boldsymbol{a}_h^t) \vee 1}}.$$

Following the exact same steps of regrouping, using the integral bound, and applying the Cauchy-Schwarz inequality, we arrive at the analogous bound:

$$S_\Phi \le 2C_\Phi H \sqrt{SAT}.$$

Now, we substitute the definition of $C_\Phi$:

$$S_\Phi \le 2\left( H\sqrt{c_2 S \log(NSAHT/\delta)} \right) H \sqrt{SAT}$$
$$= 2H^2 \sqrt{c_2} \sqrt{S} \sqrt{S} \sqrt{\log(NSAHT/\delta)} \sqrt{AT}$$
$$= \mathcal{O}\left( H^2 S \sqrt{AT \log(NSAHT/\delta)} \right).$$

The total sum of bonuses is $S_\Psi + S_\Phi$. Since the bound for $S_\Phi$ has higher order terms in $H$ and $S$, it dominates the bound for $S_\Psi$. Therefore, the total sum is bounded by the term from the transition bonuses:

$$\sum_{t=1}^{T} \sum_{h=1}^{H} \left( \Psi_h^t(s_h^t, \boldsymbol{a}_h^t) + \Phi_h^t(s_h^t, \boldsymbol{a}_h^t) \right) \le \mathcal{O}\left( H^2 S \sqrt{AT \log(NSAHT/\delta)} \right).$$

This completes the proof of the lemma. $\qquad \square$

**Theorem 12** (Restatement of Theorem 4.). *With probability at least $1 - \delta$, the Total Nash Regret of the ONVI-MG algorithm after $T$ episodes is bounded by:*

$$Regret(T) \le \mathcal{O}\left( NH^2 S \sqrt{AT \log(SAHT/\delta)} \right).$$

*Proof.* The proof directly combines the supporting lemmas to bound the sum of the per-episode Nash Gaps.

We have that

$$
\text{Regret}(T) = \sum_{t=1}^{T} \sum_{k=1}^{N} \left( \max_{\pi'^{k}} U^{k,(\pi'^{k}, \boldsymbol{\pi}^{-k,t})}(s_1) - U^{k,\boldsymbol{\pi}^{t}}(s_1) \right)
$$

$$
\leq \sum_{t=1}^{T} \sum_{k=1}^{N} \left( 2 \sum_{h=1}^{H} \mathbb{E}_{\boldsymbol{\pi}^{t}} \left[ \Psi_h^t(s_h, \boldsymbol{a}_h) + \Phi_h^t(s_h, \boldsymbol{a}_h) \right] \right) \quad \text{(Apply Lemma 5)}. \tag{18}
$$

Then under the event of Lemma 3, i.e., with probability at least $1 - \delta$, it holds that

$$
\text{Regret}(T) \leq \sum_{t=1}^{T} \sum_{k=1}^{N} \left( 2 \sum_{h=1}^{H} \mathbb{E}_{\boldsymbol{\pi}^{t}} \left[ \Psi_h^t(s_h, \boldsymbol{a}_h) + \Phi_h^t(s_h, \boldsymbol{a}_h) \right] \right)
$$

$$
\leq \sum_{k=1}^{N} \left( 2 \sum_{t=1}^{T} \sum_{h=1}^{H} \left[ \Psi_h^t(s_h^t, \boldsymbol{a}_h^t) + \Phi_h^t(s_h^t, \boldsymbol{a}_h^t) \right] \right). \tag{19}
$$

Since the bonuses $\Psi$ and $\Phi$ are shared among all agents and depend only on the joint state-action counts, we can pull the sum over agents outside the sum over time:

$$
\text{Regret}(T) \leq 2N \sum_{t=1}^{T} \sum_{h=1}^{H} \left( \Psi_h^t(s_h^t, \boldsymbol{a}_h^t) + \Phi_h^t(s_h^t, \boldsymbol{a}_h^t) \right).
$$

Now, we apply Lemma 6 (Total Bonus Bound) to the entire sum of bonuses over $T$ episodes:

$$
\sum_{t=1}^{T} \sum_{h=1}^{H} \left( \Psi_h^t(s_h^t, \boldsymbol{a}_h^t) + \Phi_h^t(s_h^t, \boldsymbol{a}_h^t) \right) \leq \mathcal{O}\left( H^2 S \sqrt{AT \log(SAHT/\delta)} \right).
$$

Substituting this bound back into our expression implies that

$$
\text{Regret}(T) \leq 2N \cdot \mathcal{O}\left( H^2 S \sqrt{AT \log(SAHT/\delta)} \right) \tag{20}
$$

$$
= \mathcal{O}\left( N H^2 S \sqrt{AT \log(SAHT/\delta)} \right). \tag{21}
$$

$\square$

# E  PROOFS FOR SECTION 4.2

## E.1  PCE

**Definition 9** (Stochastic Modification ($\phi^{(j)}$)). *A **stochastic modification** $\phi^{(j)} = \{\phi_h^{(j)}\}_{h=1}^{H}$ is a sequence of history-dependent mappings that represents a unilateral deviation strategy for agent $j$. At each step $h$, after an action $a_h^{(j)}$ is sampled from agent $j$'s original policy $\pi_h^{(j)}$, the function $\phi_h^{(j)}$ maps this action to a new (possibly random) action $\tilde{a}_h^{(j)}$. Formally:*

$$
\phi_h^{(j)} : \mathcal{H}_h \times \mathcal{A}^{(j)} \to \Delta(\mathcal{A}^{(j)}),
$$

*where $\mathcal{H}_h$ is the set of possible histories up to step $h$ and $\Delta(\mathcal{A}^{(j)})$ is the probability simplex over agent $j$'s actions.*

**Definition 10** (Modified Joint Policy ($\phi^{(j)} \circ \pi$)). *Given a joint policy $\pi$ and a stochastic modification $\phi^{(j)}$ for agent $j$, the **modified joint policy** is a new joint policy where agent $j$ plays according to the modification, and all other agents play as before. The policy for agent $j$ at step $h$, denoted $(\phi_h^{(j)} \circ \pi_h^{(j)})$, is the composition where an action is first sampled according to $\pi_h^{(j)}$ and then transformed by $\phi_h^{(j)}$.*

**Definition 11** (Agent Value Vector ($\boldsymbol{V}_{j,1}^{\boldsymbol{\pi}}(s_1)$)). *The **value vector** for agent $j$ under a joint policy $\pi$, starting from state $s_1$, is the vector of expected cumulative rewards for each of its $m$ objectives. It is defined as:*

$$
\boldsymbol{V}_{j,1}^{\boldsymbol{\pi}}(s_1) := \left( V_{\pi,1,1}^{(j)}(s_1), \ldots, V_{\pi,1,m}^{(j)}(s_1) \right) \in \mathbb{R}^M,
$$

where each component $i \in \{1, \ldots, M\}$ is the standard expected total reward for that objective:

$$V_{\pi,1,i}^{(j)}(s_1) = \mathbb{E}_\pi \left[ \sum_{h=1}^{H} r_{h,i}^{(j)}(s_h, \boldsymbol{a}_h) \middle| s_1 \right].$$

The expectation is taken over all possible trajectories generated by the joint policy $\pi$.

**Proposition 9.** *Every Pareto-Nash Equilibrium (PNE) is a Pareto Correlated Equilibrium (PCE).*

*Proof.* Let $\pi^*$ be a joint policy that is a PNE. We will show that it satisfies the definition of an PCE by using a proof by contradiction.

First, let us formally state the two relevant definitions in the language of stochastic modifications. Let $\boldsymbol{V}_{j,1}^{\boldsymbol{\pi}}(s_1)$ be the vector of expected values for player $j$ under joint policy $\pi$.

- A joint policy $\pi^*$ is a **PNE** if for any player $j$ and any alternative policy $\hat{\pi}^{(j)}$, the resulting value vector $\boldsymbol{V}_{(\hat{\pi}^{(j)}, \pi^{*,-j}),1}^{(j)}(s_1)$ does **not** Pareto dominate the original value vector $\boldsymbol{V}_{\pi^*,1}^{(j)}(s_1)$.

- A joint policy $\pi$ is a **Pareto Correlated Equilibrium** if for any player $j$ and any stochastic modification $\phi^{(j)}$, the resulting value vector $\boldsymbol{V}_{j,1}^{\phi^{(j)} \circ \boldsymbol{\pi}}(s_1)$ does **not** Pareto dominate the original value vector $\boldsymbol{V}_{j,1}^{\boldsymbol{\pi}}(s_1)$.

Now, assume for contradiction that the PNE policy $\pi^*$ is **not** an PCE.

According to the definition of PCE, this assumption implies that there exists at least one player, agent $j$, and a specific stochastic modification, $\phi^{(j)}$, such that agent $j$'s value vector under the modified policy Pareto dominates her value vector under the original policy. Formally, for a starting state $s_1$:

$$\boldsymbol{V}_{\phi^{(j)} \circ \pi^*,1}^{(j)}(s_1) \quad \text{Pareto dominates} \quad \boldsymbol{V}_{\pi^*,1}^{(j)}(s_1). \tag{22}$$

The core of our argument is to recognize that a stochastic modification applied to a policy creates a new, valid policy. Let us define a new policy for agent $j$, which we will call $\hat{\pi}^{(j)}$, as the policy resulting from the composition of the original policy and the stochastic modification. That is:

$$\hat{\pi}^{(j)} := \phi^{(j)} \circ \pi^{*(j)}.$$

This policy $\hat{\pi}^{(j)}$ represents the complete strategy of "first, determine the action from my original PNE policy $\pi^{*(j)}$, and then apply the deviating transformation $\phi^{(j)}$ to get my final action." This is a valid alternative policy for agent $j$.

By substituting this definition of $\hat{\pi}^{(j)}$ into our assumption in Equation equation 22, we have found an alternative policy $\hat{\pi}^{(j)}$ for agent $j$ such that:

$$\boldsymbol{V}_{(\hat{\pi}^{(j)}, \pi^{*,-j}),1}^{(j)}(s_1) \quad \text{Pareto dominates} \quad \boldsymbol{V}_{\pi^*,1}^{(j)}(s_1).$$

This statement, however, is a direct contradiction of the definition of $\pi^*$ being a PNE. The PNE condition requires that for *any* alternative policy—which includes policies formed by stochastic modifications—a unilateral deviation cannot lead to a Pareto-dominant outcome.

Since our initial assumption (that $\pi^*$ is not an PCE) leads to a logical contradiction with the premise (that $\pi^*$ is a PNE), the assumption must be false.

Therefore, the PNE policy $\pi^*$ must be an PCE. $\qquad \square$

**Proposition 10.** *Every PNE of a Multi-Objective Markov Game induces a joint policy distribution that is a Pareto Correlated Equilibrium.*

*Proof.* Let $\boldsymbol{\pi}^* = (\pi^{1,*}, \ldots, \pi^{N,*})$ be a PNE for a given Multi-Objective Markov Game. A PNE is a profile of stationary policies. For any given state $s$ and step $h$, this joint policy induces a probability distribution over the joint action space $\mathcal{A}$:

$$\sigma(\boldsymbol{a}|s) = \prod_{k=1}^{N} \pi_h^{k,*}(a^k|s).$$

We will show that this distribution $\sigma(\cdot|s)$ satisfies the conditions for a Pareto Correlated Equilibrium at every state $s$. We proceed by contradiction.

Assume that $\sigma(\cdot|s)$ is **not** a Pareto Correlated Equilibrium. By definition, this means there must exist at least one agent, say agent $j$, a recommended action $a^j \in \mathcal{A}^j$ with a non-zero probability of being recommended ($\pi_h^{j,*}(a^j|s) > 0$), and a deviating action $a'^j \in \mathcal{A}^j$, such that the expected value from deviating Pareto dominates the expected value from obeying.

The expected value vector for agent $j$ when obeying recommendation $a^j$ is calculated over the other agents' actions, which are distributed according to $\boldsymbol{\pi}^{-j,*}(\cdot|s)$:

$$\mathbb{E}[\boldsymbol{V}_h^j|s, a^j] = \mathbb{E}_{\boldsymbol{a}^{-j} \sim \boldsymbol{\pi}_h^{-j,*}(\cdot|s)} \left[ \boldsymbol{Q}_h^j(s, (a^j, \boldsymbol{a}^{-j})) \right].$$

Similarly, the expected value vector for deviating to $a'^j$ is:

$$\mathbb{E}[\boldsymbol{V}_h^j|s, a^j, a'^j] = \mathbb{E}_{\boldsymbol{a}^{-j} \sim \boldsymbol{\pi}_h^{-j,*}(\cdot|s)} \left[ \boldsymbol{Q}_h^j(s, (a'^j, \boldsymbol{a}^{-j})) \right].$$

Our assumption that $\sigma$ is not an PCE means:

$$\mathbb{E}_{\boldsymbol{a}^{-j} \sim \boldsymbol{\pi}_h^{-j,*}(\cdot|s)} \left[ \boldsymbol{Q}_h^j(s, (a'^j, \boldsymbol{a}^{-j})) \right] \quad \text{Pareto dominates} \quad \mathbb{E}_{\boldsymbol{a}^{-j} \sim \boldsymbol{\pi}_h^{-j,*}(\cdot|s)} \left[ \boldsymbol{Q}_h^j(s, (a^j, \boldsymbol{a}^{-j})) \right]. \tag{23}$$

Now, let's use this to construct a new policy $\hat{\pi}^j$ for agent $j$ that creates a Pareto improvement over $\pi^{j,*}$, which will contradict that $\boldsymbol{\pi}^*$ is a PNE. Define $\hat{\pi}^j$ as follows:

$$\hat{\pi}_h^j(a|s) = \begin{cases} \pi_h^{j,*}(a|s) & \text{if } a \neq a^j \text{ and } a \neq a'^j \\ 0 & \text{if } a = a^j \\ \pi_h^{j,*}(a'^j|s) + \pi_h^{j,*}(a^j|s) & \text{if } a = a'^j. \end{cases}$$

Essentially, this new policy $\hat{\pi}^j$ is identical to $\pi^{j,*}$, except that whenever it would have played the recommended action $a^j$, it instead plays the deviating action $a'^j$.

The total expected value for agent $j$ under the original policy profile $\boldsymbol{\pi}^*$ is $\boldsymbol{V}_1^{j,\boldsymbol{\pi}^*}(s_1)$. The total expected value under the new profile $(\hat{\pi}^j, \boldsymbol{\pi}^{-j,*})$ is $\boldsymbol{V}_1^{j,(\hat{\pi}^j, \boldsymbol{\pi}^{-j,*})}(s_1)$.

The difference in the total expected value vectors can be traced back to the local change in policy at state $s$ and step $h$. The inequality in equation 23 shows that for the state-action distribution induced by $(\hat{\pi}^j, \boldsymbol{\pi}^{-j,*})$, the expected outcome for agent $j$ is a Pareto improvement over the outcome from $\boldsymbol{\pi}^*$ whenever state $s$ is visited and action $a^j$ would have been chosen. Since the policy is otherwise identical, the total expected value must also be a Pareto improvement:

$$\boldsymbol{V}_1^{j,(\hat{\pi}^j, \boldsymbol{\pi}^{-j,*})}(s_1) \quad \text{Pareto dominates} \quad \boldsymbol{V}_1^{j,\boldsymbol{\pi}^*}(s_1).$$

This contradicts our initial premise that $\boldsymbol{\pi}^*$ is a PNE, because we have found a unilateral deviation for agent $j$ that results in a Pareto-dominant outcome.

Therefore, our assumption that $\sigma(\cdot|s)$ is not a Pareto Correlated Equilibrium must be false. Every PNE induces a distribution that is a Pareto Correlated Equilibrium. $\qquad \square$

This proposition directly leads to the following crucial corollary regarding existence.

**Corollary 1** (Existence of PCE)**.** *For any finite Multi-Objective Markov Game, at least one Pareto Correlated Equilibrium exists.*

**Theorem 13.** *It holds that*

$$\boldsymbol{PCE}(G) = \cup_{\boldsymbol{\Lambda} \in (\Delta_M^\circ)^N} \boldsymbol{CE}(G_{\boldsymbol{\Lambda}}). \tag{24}$$

*Proof.* (**Scalarized CE** $\implies$ **PCE**): Assume $\sigma$ is an Ex-Ante CE for a scalarized game with $\boldsymbol{\lambda} > 0$, but is not an Ex-Ante PCE. The failure to be an PCE means there exists a player $i$ and a modification $\phi_i$ such that $V_i^{\phi_i \circ \sigma}(s_1)$ Pareto dominates $V_i^\sigma(s_1)$. By the properties of dot products with strictly positive vectors, this implies:

$$\lambda_i \cdot V_i^{\phi_i \circ \sigma}(s_1) > \lambda_i \cdot V_i^\sigma(s_1).$$

This contradicts the assumption that $\sigma$ is an Ex-Ante CE for the scalarization $\boldsymbol{\lambda}$. Thus, the assumption is false, and $\sigma$ must be an PCE.

(**PCE** $\implies$ **Scalarized CE**): Assume $\sigma$ is an Ex-Ante PCE. For any player $i$, let $\mathcal{D}_i(s_1)$ be the set of all possible improvement vectors from the start state $s_1$:

$$\mathcal{D}_i(s_1) = \{V_i^{\phi_i \circ \sigma}(s_1) - V_i^\sigma(s_1) \mid \phi_i \in \Phi_i\}.$$

The PCE condition implies that $\mathcal{D}_i(s_1)$ is disjoint from the positive cone $\mathbb{R}_{++}^k$. Since $\mathcal{D}_i(s_1)$ is convex, by the Separating Hyperplane Theorem, there exists a non-zero vector $\lambda_i \in \mathbb{R}_{\geq 0}^k$ such that for every $d \in \mathcal{D}_i(s_1)$, we have $\lambda_i \cdot d \leq 0$. For strong Pareto concepts, this vector can be chosen to be strictly positive, $\lambda_i \in \mathbb{R}_{++}^k$. This gives us:

$$\lambda_i \cdot (V_i^{\phi_i \circ \sigma}(s_1) - V_i^\sigma(s_1)) \leq 0.$$

This is the definition of an Ex-Ante CE for the game scalarized by the constructed preference vectors $\{\lambda_i\}_{i \in N}$. $\qquad\square$

## E.2 MULTI-OBJECTIVE V-LEARNING

We first present our multi-objective V-learning algorithm as follows.

---

**Algorithm 3** Multi-Objective V-Learning (MO-V-Learning)

---

1: **Input:** Preference profile $\Lambda = \{\lambda^1, \ldots, \lambda^N\} \in (\Delta_M^o)^N$, total episodes $K$.
2: **Initialize:** For each agent $j \in [N]$ and all $(s, h)$:
3:     $V_{j,h}(s) \leftarrow H + 1 - h$, $N_{j,h}(s) \leftarrow 0$, $\pi_{j,h}(\cdot|s) \leftarrow \text{Uniform}(\mathcal{A}_j)$.
4:     Instantiate $S \times H$ adversarial bandit subroutines $\text{SWAP\_BANDIT}_j(s, h)$ with low swap regret.

5: **for** episode $k = 1, \ldots, K$ **do**
6:     Receive initial state $s_1$.
7:     **for** step $h = 1, \ldots, H$ **do**
8:        All agents observe $s_h$.
9:        Each agent $j$ takes action $a_{j,h} \sim \pi_{j,h}(\cdot|s_h)$.
10:       Environment executes joint action $a_h = (a_{1,h}, \ldots, a_{N,h})$, transitions to $s_{h+1} \sim P_h(\cdot|s_h, a_h)$.
11:       Each agent $j$ receives **random** vector reward $\boldsymbol{r}_{j,h}^k$ and observes $s_{h+1}$.
12:       **V-Learning Update (for each agent $j$ independently):**
13:         $t = N_{j,h}(s_h) \leftarrow N_{j,h}(s_h) + 1$.
14:         Compute **observed** scalar reward: $\overline{\boldsymbol{r}}_{j,h}^k \leftarrow (\lambda^j)^\top \boldsymbol{r}_{j,h}^k$.
15:         Set learning rate $\alpha_t = \frac{H+1}{H+t}$ and bonus $\beta_{j,t}$ (see Theorem 6).
16:         Compute V-value: $\tilde{V}_{j,h}(s_h) \leftarrow (1 - \alpha_t)V_{j,h}(s_h) + \alpha_t(\overline{\boldsymbol{r}}_{j,h}^k + V_{j,h+1}(s_{h+1}) + \beta_{j,t})$.
17:         Truncate: $V_{j,h}(s_h) \leftarrow \min\{H + 1 - h, \tilde{V}_{j,h}(s_h)\}$.
18:         Compute loss: $l_{j,h} \leftarrow \frac{H - \overline{\boldsymbol{r}}_{j,h}^k - V_{j,h+1}(s_{h+1})}{H}$.
19:         Update policy: $\pi_{j,h}(\cdot|s_h) \leftarrow \text{SWAP\_BANDIT\_UPDATE}(a_{j,h}, l_{j,h})$ on $\text{SWAP\_BANDIT}_j(s_h, h)$.
20:     **end for**
21: **end for**
22: **Output Policy $\hat{\pi}$:** (Execution Protocol for CE)
23:     A single shared random seed is broadcast to all $N$ agents.
24:     This seed is used to sample $k \sim \text{Uniform}([K])$.
25:     At each step $h$, given $s_h$:
26:        Let $t = N_{j,h}^k(s_h)$ (using the $k$ from the previous step).
27:        All agents use the shared seed to sample an index $i \in [t]$ with probability $\alpha_t^i$.
28:        All agents set $k \leftarrow k_h^i(s_h)$ (the episode index of the $i$-th visit).
29:        Each agent $j$ plays $a_{j,h} \sim \pi_{j,h}^k(\cdot|s_h)$.
30:     The resulting joint policy is $\hat{\pi} = \hat{\pi}_1 \odot \cdots \odot \hat{\pi}_N$.

---

Following (Jin et al., 2021), we adopt an assumption on the SWAP-BANDIT sub-route.

**Assumption 1** (Low-Swap-Regret Bandit (Jin et al., 2021)). *The SWAP\_BANDIT\_UPDATE subroutine, for any $t$ and $\delta$, satisfies with probability $1 - \delta$:*

$$\max_{\psi \in \Psi} \sum_{i=1}^{t} \alpha_t^i [\langle \theta_i, l_i \rangle - \langle \psi \circ \theta_i, l_i \rangle] \leq \xi_{sw}(B, t, \log(1/\delta)),$$

*and $\sum_{t'=1}^{t} \xi_{sw}(B, t', \log(1/\delta)) \leq \Xi_{sw}(B, t, \log(1/\delta))$, where $\Xi_{sw}$ is concave in $t$.*

For FTRL\_SWAP algorithm (Algorithm 6 in (Jin et al., 2021)), these bounds are $\xi_{sw} = \mathcal{O}(B\sqrt{H\iota/t})$ and $\Xi_{sw} = \mathcal{O}(B\sqrt{Ht\iota})$.

**Lemma 7** (Optimism). *With probability at least $1 - \delta$, for all $(j, s, h, k)$:*

$$\overline{V}_{j,h}^k(s) \geq \max_{\phi_j} V_{j,h}^{(\phi_j \circ \hat{\pi}_{j,h}^k) \odot \hat{\pi}_{-j,h}^k}(s).$$

*Proof.* We prove by backward induction on $h$. The base case $h = H + 1$ is trivial, as $\overline{V}_{j,H+1}^k(s) = 0$ and $V_{j,H+1}^\pi(s) = 0$. Assume the hypothesis holds for $h + 1$. Let $t = N_{j,h}^k(s)$ and $k^1, \ldots, k^t < k$ be

the episodes of the $t$ previous visits to $(s, h)$.

$$\max_{\phi_j} V_{j,h}^{(\phi_j \circ \hat{\pi}_{j,h}^k) \odot \hat{\pi}_{-j,h}^k}(s)$$

$$= \max_{\phi_{j,h}} \sum_{i=1}^{t} \alpha_t^i \mathbb{E}_{a \sim (\phi_{j,h} \circ \pi_{j,h}^{k^i}) \odot \pi_{-j,h}^{k^i}} \left[ \overline{r}_{j,h}(s,a) + \mathbb{E}_{s'} \left[ \max_{\phi_{j,h+1}} V_{j,h+1}^{(\phi_{j,h+1} \circ \hat{\pi}_{j,h+1}^{k^i}) \odot \hat{\pi}_{-j,h+1}^{k^i}}(s') \right] \right] \quad (1)$$

$$\leq \max_{\phi_{j,h}} \sum_{i=1}^{t} \alpha_t^i \mathbb{E}_{a \sim (\phi_{j,h} \circ \pi_{j,h}^{k^i}) \odot \pi_{-j,h}^{k^i}} \left[ \overline{r}_{j,h}(s,a) + P_h \overline{V}_{j,h+1}^{k^i}(s) \right] \quad (2)$$

$$\leq \sum_{i=1}^{t} \alpha_t^i \mathbb{E}_{a \sim \pi_h^{k^i}} \left[ \overline{r}_{j,h}(s,a) + P_h \overline{V}_{j,h+1}^{k^i}(s) \right] + H \xi_{sw}(A_j, t, \iota) \quad (3)$$

$$\leq \sum_{i=1}^{t} \alpha_t^i \left[ \overline{\boldsymbol{r}}_{j,h}^{k^i} + \overline{V}_{j,h+1}^{k^i}(s_{h+1}^{k^i}) \right] + \mathcal{O}(\sqrt{H^3 \iota / t}) + H \xi_{sw}(A_j, t, \iota) \quad (4)$$

$$\leq \sum_{i=1}^{t} \alpha_t^i \left[ \overline{\boldsymbol{r}}_{j,h}^{k^i} + \overline{V}_{j,h+1}^{k^i}(s_{h+1}^{k^i}) + \beta_{j,i} \right] + \alpha_t^0 (H - h + 1) \quad (5)$$

$$\leq \overline{V}_{j,h}^{k}(s). \quad (6)$$

Here, (1) is the Bellman expansion for the best-response value under the output policy $\hat{\pi}_h^k$, which is a mixture of policies $\{\pi_h^{k^i}\}$ weighted by $\alpha_t^i$; (2) Applies the Induction Hypothesis to the $V_{j,h+1}$ term; (3) uses the definition of the swap-regret bandit (Assumption 1). The regret is bounded by $\xi_{sw}$, and the loss is scaled by $H$ (as the loss $l_{j,h}$ is in $[0,1]$); (4) follows from standard martingale concentration inequalities (Jin et al., 2021). The term $\mathcal{O}(\sqrt{H^3 \iota / t})$ bounds the deviation of the empirical sum of observed rewards $\overline{\boldsymbol{r}}_{j,h}^{k^i}$ and observed next values $\overline{V}_{j,h+1}^{k^i}(s_{h+1}^{k^i})$ from the true expected value in (3). This single term accounts for all sources of randomness: policy sampling, transition stochasticity, and reward stochasticity; (5) holds by our choice of the bonus $\beta_{j,i}$, which is set such that $\sum_{i=1}^{t} \alpha_t^i \beta_{j,i} = \Theta(H \xi_{sw} + \sqrt{H^3 \iota / t})$ to cancel both the regret and concentration error terms. We also add the initialization value (which is 0 if $t > 0$); And (6) is the definition of the optimistic estimator $\overline{V}_{j,h}^{k}(s)$ from Algorithm 3 (Line 16) and Lemma 11 in (Jin et al., 2021). $\square$

**Lemma 8** (Pessimism). *With probability at least $1 - \delta$, for all $(j, s, h, k)$:*

$$\underline{V}_{j,h}^{k}(s) \leq V_{j,h}^{\hat{\pi}_h^k}(s).$$

*Proof.* Again, we prove by backward induction on $h$. The base case $h = H + 1$ is trivial. Assume the hypothesis holds for $h + 1$.

$$V_{j,h}^{\hat{\pi}_h^k}(s) = \sum_{i=1}^{t} \alpha_t^i \mathbb{E}_{a \sim \pi_h^{k^i}} \left[ \overline{r}_{j,h}(s,a) + \mathbb{E}_{s'} \left[ V_{j,h+1}^{\hat{\pi}_{h+1}^{k^i}}(s') \right] \right] \quad (1)$$

$$\geq \sum_{i=1}^{t} \alpha_t^i \mathbb{E}_{a \sim \pi_h^{k^i}} \left[ \overline{r}_{j,h}(s,a) + P_h \underline{V}_{j,h+1}^{k^i}(s) \right] \quad (2)$$

$$\geq \sum_{i=1}^{t} \alpha_t^i \left[ \overline{\boldsymbol{r}}_{j,h}^{k^i} + \underline{V}_{j,h+1}^{k^i}(s_{h+1}^{k^i}) \right] - \mathcal{O}(\sqrt{H^3 \iota / t}) \quad (3)$$

$$\geq \sum_{i=1}^{t} \alpha_t^i \left[ \overline{\boldsymbol{r}}_{j,h}^{k^i} + \underline{V}_{j,h+1}^{k^i}(s_{h+1}^{k^i}) - \beta_{j,i} \right] + \alpha_t^0 (H - h + 1) \quad (4)$$

$$\geq \underline{V}_{j,h}^{k}(s).$$

Here, (1) is the definition of the value of the output policy $\hat{\pi}$; (2) applies the Induction Hypothesis to the $V_{j,h+1}$ term; (3) follows from martingale concentration inequalities, bounding the deviation of the empirical sum of observed rewards from the expectation in (2). $\square$

**Theorem 14** (Sample Complexity for PCE). *Let $A = \max_j A_j$ and $\iota = \log(NHSAK/\delta)$. Run MO-V-Learning (Algorithm 3) for $K$ episodes using a bandit subroutine satisfying Assumption 1. Set the bonus for each agent $j$ such that $\sum_{i=1}^{t} \alpha_t^i \beta_{j,i} = \Theta(H\xi_{sw}(A_j, t, \iota) + \sqrt{H^3\iota/t})$ (e.g., $\beta_{j,t} = c \cdot A_j\sqrt{H^3\iota/t}$ for FTRL_swap). Then, with probability at least $1 - \delta$, the output policy $\hat{\pi}$ is an $\epsilon$-PCE of $G_{\mathbf{\Lambda}}$, where:*

$$\epsilon = \max_{j,\phi_j} \left[ V_{j,1}^{(\phi_j \circ \hat{\pi}_j) \odot \hat{\pi}_{-j}}(s_1) - V_{j,1}^{\hat{\pi}}(s_1) \right] \leq \mathcal{O}\left( A\sqrt{\frac{H^5 S\iota}{K}} \right).$$

*(Here, $V$ denotes the value in the scalarized game $G_{\mathbf{\Lambda}}$.)*

*Proof.* Now we bound the swap regret by the gap between the estimators and then bound the gap itself. The swap regret of the final output policy $\hat{\pi}$ (which is a mixture over all $k \in [K]$) is bounded by the average gap over all $K$ episodes. Let

$$\epsilon = \max_{j,\phi_j} \left[ V_{j,1}^{(\phi_j \circ \hat{\pi}_j) \odot \hat{\pi}_{-j}}(s_1) - V_{j,1}^{\hat{\pi}}(s_1) \right] \leq \frac{1}{K} \sum_{k=1}^{K} \max_j \left[ \overline{V}_{j,1}^k(s_1) - \underline{V}_{j,1}^k(s_1) \right].$$

Let $\delta_{j,h}^k = \overline{V}_{j,h}^k(s_h^k) - \underline{V}_{j,h}^k(s_h^k) \geq 0$. Let $\delta_h^k = \max_j \delta_{j,h}^k$. Let $n_h^k = N_{j,h}^k(s_h^k)$ be the visit count at episode $k$ to state $s_h^k$. Then it holds that

$$
\begin{aligned}
\delta_{j,h}^k &= \overline{V}_{j,h}^k(s_h^k) - \underline{V}_{j,h}^k(s_h^k) \\
&\leq \left( \alpha_{n_h^k}^0 H + \sum_{i=1}^{n_h^k} \alpha_{n_h^k}^i [\overline{r}_{j,h}^{k^i} + \overline{V}_{j,h+1}^{k^i} + \beta_{j,i}] \right) - \left( \sum_{i=1}^{n_h^k} \alpha_{n_h^k}^i [\overline{r}_{j,h}^{k^i} + \underline{V}_{j,h+1}^{k^i} - \beta_{j,i}] \right) \\
&\leq \alpha_{n_h^k}^0 H + \sum_{i=1}^{n_h^k} \alpha_{n_h^k}^i \left[ (\overline{V}_{j,h+1}^{k^i} - \underline{V}_{j,h+1}^{k^i}) + 2\beta_{j,i} \right] \\
&\leq \alpha_{n_h^k}^0 H + \sum_{i=1}^{n_h^k} \alpha_{n_h^k}^i \delta_{j,h+1}^{k^i} + 2 \sum_{i=1}^{n_h^k} \alpha_{n_h^k}^i \beta_{j,i},
\end{aligned}
$$

where we utilize Lemma 7 and Lemma 8. By our choice of bonus $\sum_{i=1}^{t} \alpha_t^i \beta_{j,i} = \Theta(H\xi_{sw}(A_j, t, \iota) + \sqrt{H^3\iota/t})$, and taking the max over $j$:

$$\delta_h^k \leq \alpha_{n_h^k}^0 H + \sum_{i=1}^{n_h^k} \alpha_{n_h^k}^i \delta_{h+1}^{k^i} + \mathcal{O}(H\xi_{sw}(A, n_h^k, \iota) + \sqrt{H^3\iota/n_h^k}).$$

Summing over $k = 1 \dots K$ further implies that:

$$\sum_{k=1}^{K} \delta_h^k \leq \sum_{k=1}^{K} \alpha_{n_h^k}^0 H + \sum_{k=1}^{K} \sum_{i=1}^{n_h^k} \alpha_{n_h^k}^i \delta_{h+1}^{k^i} + \sum_{k=1}^{K} \mathcal{O}(H\xi_{sw}(A, n_h^k, \iota) + \sqrt{H^3\iota/n_h^k}).$$

Since the first term $\sum \alpha_{n_h^k}^0 H \leq SH$, and the second term $\sum_{k=1}^{K} \sum_{i=1}^{n_h^k} \alpha_{n_h^k}^i \delta_{h+1}^{k^i} \leq (1 + 1/H) \sum_{k=1}^{K} \delta_{h+1}^k$. Telescoping this recurrence from $h = 1$ to $H$ further implies that

$$\sum_{k=1}^{K} \delta_1^k \leq eSH^2 + e \sum_{h=1}^{H} \sum_{k=1}^{K} \mathcal{O}(H\xi_{sw}(A, n_h^k, \iota) + \sqrt{H^3\iota/n_h^k}).$$

We bound the final sum using the pigeonhole principle and concavity of $\Xi_{sw}$ and $\sqrt{\cdot}$ as (Jin et al., 2021):

$$\sum_{h=1}^{H} \sum_{k=1}^{K} \mathcal{O}(H\xi_{sw}(A, n_h^k, \iota) + \sqrt{H^3\iota/n_h^k})$$

$$= \sum_{h=1}^{H} \sum_{s\in\mathcal{S}} \sum_{n=1}^{N_h^K(s)} \mathcal{O}(H\xi_{sw}(A, n, \iota) + \sqrt{H^3\iota/n})$$

$$\leq \sum_{h=1}^{H} \sum_{s\in\mathcal{S}} \mathcal{O}(H\Xi_{sw}(A, N_h^K(s), \iota) + \sqrt{H^3 N_h^K(s)\iota})$$

$$\leq \sum_{h=1}^{H} \mathcal{O}(HS\Xi_{sw}(A, K/S, \iota) + \sqrt{H^3 SK\iota}) \quad \text{(since } \sum_s N_h^K(s) = K \text{ and concavity)}$$

$$\leq \mathcal{O}(H^2 S\Xi_{sw}(A, K/S, \iota) + \sqrt{H^5 SK\iota}).$$

Plugging in the bound for FTRL_swap, $\Xi_{sw}(B, t, \iota) = \mathcal{O}(B\sqrt{Ht\iota})$:

$$\sum_{k=1}^{K} \delta_1^k \leq \mathcal{O}(H^2 S(A\sqrt{H(K/S)\iota}) + \sqrt{H^5 SK\iota}) = \mathcal{O}(A\sqrt{H^5 SK\iota}).$$

The average gap, which bounds the swap regret, is:

$$\epsilon \leq \frac{1}{K} \sum_{k=1}^{K} \delta_1^k \leq \mathcal{O}\left(A\sqrt{\frac{H^5 S\iota}{K}}\right).$$

This completes the proof. $\qquad\square$

## F    PROOFS FOR SECTION 5

**Lemma 9** (Concentration of Empirical Model). *Let $\mathcal{D}$ be the dataset collected after $T$ episodes of Algorithm 1. Let $(\{\hat{r}_j\}, \hat{P})$ be the empirical model estimated from $\mathcal{D}$. Then, with probability at least $1 - \delta$, for all $(s, \boldsymbol{a}, h)$, all players $j$, all objectives $i$, and any function $V : \mathcal{S} \to [0, H]$:*

$$|\hat{r}_{j,i,h}(s, \boldsymbol{a}) - r_{j,i,h}(s, \boldsymbol{a})| \leq \sqrt{\frac{C_r}{N_h(s, \boldsymbol{a}) \vee 1}} \wedge 1 =: \Psi_{j,i,h}(s, \boldsymbol{a}),$$

$$|\sum_{s'} (\hat{P}_h(s'|s, \boldsymbol{a}) - P_h(s'|s, \boldsymbol{a}))V(s')| \leq \sqrt{\frac{C_p SH^2}{N_h(s, \boldsymbol{a}) \vee 1}} \wedge H =: \Phi_h(s, \boldsymbol{a}),$$

*where $N_h(s, \boldsymbol{a})$ is the total visitation count and $C_r, C_p$ are logarithmic factors in problem parameters and $1/\delta$.*

**Lemma 10** (Value Difference Lemma for N-Player Games). *Let $V_j^{\boldsymbol{\pi}}$ and $\hat{V}_j^{\boldsymbol{\pi}}$ be the scalarized value functions for player $j$ under joint policy $\boldsymbol{\pi}$ in the true game $M$ and the empirical game $\hat{M}$, respectively. With probability at least $1 - \delta$, for any player $j$ and any policy $\boldsymbol{\pi}$:*

$$|V_{j,1}^{\boldsymbol{\pi}}(s_1) - \hat{V}_{j,1}^{\boldsymbol{\pi}}(s_1)| \leq \mathbb{E}_{\boldsymbol{\pi}}\left[\sum_{h=1}^{H} (\Psi_{j,h}(s_h, \boldsymbol{a}_h) + \Phi_h(s_h, \boldsymbol{a}_h))\right].$$

*Proof.* Let $\Delta_{j,h}^{\boldsymbol{\pi}}(s) = V_{j,h}^{\boldsymbol{\pi}}(s) - \hat{V}_{j,h}^{\boldsymbol{\pi}}(s)$. Using the Bellman equations for both games:

$$\Delta_{j,h}^{\boldsymbol{\pi}}(s) = \mathbb{E}_{\boldsymbol{a}\sim\boldsymbol{\pi}_h(s)}\left[(r_{j,h} - \hat{r}_{j,h}) + \sum_{s'} (P_h(s'|s, \boldsymbol{a})V_{j,h+1}^{\boldsymbol{\pi}}(s') - \hat{P}_h(s'|s, \boldsymbol{a})\hat{V}_{j,h+1}^{\boldsymbol{\pi}}(s'))\right]$$

$$= \mathbb{E}_{\boldsymbol{a}\sim\boldsymbol{\pi}_h(s)}\left[(r_{j,h} - \hat{r}_{j,h}) + \sum_{s'} (P_h - \hat{P}_h)V_{j,h+1}^{\boldsymbol{\pi}}(s') + \sum_{s'} \hat{P}_h\Delta_{j,h+1}^{\boldsymbol{\pi}}(s')\right].$$

Taking absolute values and applying Lemma 9:

$$|\Delta_{j,h}^{\boldsymbol{\pi}}(s)| \leq \mathbb{E}_{\boldsymbol{a} \sim \boldsymbol{\pi}_h(s)}\left[\Psi_{j,h}(s,\boldsymbol{a}) + \Phi_h(s,\boldsymbol{a})\right] + \mathbb{E}_{s' \sim \hat{P}_h}[|\Delta_{j,h+1}^{\boldsymbol{\pi}}(s')|].$$

Unrolling this recursion from $h = 1$ to $H$ with $\Delta_{j,H+1}^{\boldsymbol{\pi}} = 0$ yields the result. □

**Lemma 11** (Optimism and Exploration Value Bound). *Let $\overline{V}_1^t(s_1)$ be the value of the NE of the exploration game at episode $t$. Then with high probability:*

1. ***(Optimism)** For any joint policy $\boldsymbol{\pi}$, its value under the exploration reward $\bar{r}^t$ is bounded by the exploration game's value: $V_1^{\boldsymbol{\pi}}(s_1; \bar{r}^t) \leq \overline{V}_1^t(s_1)$.*

2. ***(Exploration Value Bounds Bonuses)** The expected total bonus for any policy $\boldsymbol{\pi}$ under the bonus functions from the full dataset $\mathcal{D}$ is bounded by the average exploration value:*

$$\mathbb{E}_{\boldsymbol{\pi}}\left[\sum_{h=1}^{H}\left(\Phi_h(s_h,\boldsymbol{a}_h) + \sum_{j,i}\Psi_{j,i,h}(s_h,\boldsymbol{a}_h)\right)\right] \leq \frac{N \cdot m \cdot H}{T}\sum_{t=1}^{T}\overline{V}_1^t(s_1).$$

*Proof.* Part 1 follows from the optimistic construction of $\bar{Q}_h^t$ and the fact that $\bar{\pi}^t$ is an NE for the optimistic game. This is a standard argument showing that the value of an optimistic algorithm is an upper bound on the true optimal value.

For Part 2, let $f_h(s,\boldsymbol{a}) = \Phi_h(s,\boldsymbol{a}) + \sum_{j,i}\Psi_{j,i,h}(s,\boldsymbol{a})$. From the definition of $\bar{r}_h^t$, we have $\Phi_h^t/H \leq \bar{r}_h^t$ and $\Psi_{j,i,h}^t \leq \bar{r}_h^t$. Therefore, $\Phi_h \leq \frac{1}{T}\sum_t \Phi_h^t \leq \frac{H}{T}\sum_t \bar{r}_h^t$ (by Jensen's inequality and concavity of $\sqrt{\cdot}$), and similarly $\Psi_{j,i,h} \leq \frac{1}{T}\sum_t \Psi_{j,i,h}^t \leq \frac{1}{T}\sum_t \bar{r}_h^t$. Summing these up, $\mathbb{E}_{\boldsymbol{\pi}}[\sum_h f_h] \leq \mathbb{E}_{\boldsymbol{\pi}}[\sum_h \frac{H+Nm}{T}\sum_t \bar{r}_h^t]$. By linearity of expectation and Part 1:

$$\mathbb{E}_{\boldsymbol{\pi}}\left[\sum_h f_h\right] \leq \frac{H+Nm}{T}\sum_t \mathbb{E}_{\boldsymbol{\pi}}\left[\sum_h \bar{r}_h^t\right] = \frac{H+Nm}{T}\sum_t V_1^{\boldsymbol{\pi}}(s_1; \bar{r}^t) \leq \frac{H+Nm}{T}\sum_t \overline{V}_1^t(s_1).$$

□

**Lemma 12** (Total Bonus Sum Bound). *With high probability, the sum of values from the exploration game is bounded:*

$$\sum_{t=1}^{T}\overline{V}_1^t(s_1) \leq \tilde{\mathcal{O}}\left(H^2 S\sqrt{T\prod_{k=1}^{N}|\mathcal{A}_k|}\right).$$

*Proof.* Let's analyze a single exploration episode $t$. The exploration policy $\bar{\pi}^t$ is a Nash Equilibrium for the game defined by the optimistic Q-function $\bar{Q}_h^t$. Let $(s_h^t, \boldsymbol{a}_h^t)$ denote the state and joint action sampled at step $h$ of episode $t$.

For any step $h$ and state $s_h^t$, the value of the exploration game is $\overline{V}_h^t(s_h^t)$. Since $\bar{\pi}_h^t$ is an NE policy, this value is realized by playing according to it. For a NE, we have $\overline{V}_h^t(s_h^t) = \bar{Q}_h^t(s_h^t, \boldsymbol{a}_h^t)$. From the definition in Algorithm 1:

$$\overline{V}_h^t(s_h^t) \leq \bar{r}_h^t(s_h^t, \boldsymbol{a}_h^t) + \sum_{s'}\hat{P}_h^{t-1}(s'|s_h^t, \boldsymbol{a}_h^t)\overline{V}_{h+1}^t(s') + \Phi_h^t(s_h^t, \boldsymbol{a}_h^t).$$

We can relate the sum over the empirical transition $\hat{P}$ to the true transition $P$. By the definition of the bonus $\Phi_h^t$ (from Lemma 9, which holds with high probability):

$$\sum_{s'}\hat{P}_h^{t-1}(s'|s_h^t, \boldsymbol{a}_h^t)\overline{V}_{h+1}^t(s') \leq \sum_{s'}P_h(s'|s_h^t, \boldsymbol{a}_h^t)\overline{V}_{h+1}^t(s') + \Phi_h^t(s_h^t, \boldsymbol{a}_h^t).$$

Substituting this back, we get:

$$\overline{V}_h^t(s_h^t) \leq \bar{r}_h^t(s_h^t, \boldsymbol{a}_h^t) + 2\Phi_h^t(s_h^t, \boldsymbol{a}_h^t) + \mathbb{E}_{s_{h+1}^t \sim P_h(\cdot|s_h^t, \boldsymbol{a}_h^t)}[\overline{V}_{h+1}^t(s_{h+1}^t)].$$

Let $\xi_h^t = \overline{V}_{h+1}^t(s_{h+1}^t) - \mathbb{E}[\overline{V}_{h+1}^t(s_{h+1}^t)|s_h^t, \boldsymbol{a}_h^t]$. This is a martingale difference sequence with $|\xi_h^t| \leq H$. Rearranging the inequality:

$$\overline{V}_h^t(s_h^t) - \overline{V}_{h+1}^t(s_{h+1}^t) \leq \bar{r}_h^t(s_h^t, \boldsymbol{a}_h^t) + 2\Phi_h^t(s_h^t, \boldsymbol{a}_h^t) - \xi_h^t.$$

Summing this telescopically from $h = 1$ to $H$ for episode $t$, and noting $\overline{V}_{H+1}^t = 0$:

$$\overline{V}_1^t(s_1) \leq \sum_{h=1}^H \left( \bar{r}_h^t(s_h^t, \boldsymbol{a}_h^t) + 2\Phi_h^t(s_h^t, \boldsymbol{a}_h^t) - \xi_h^t \right).$$

Now, summing over all episodes $t = 1, \ldots, T$:

$$\sum_{t=1}^T \overline{V}_1^t(s_1) \leq \sum_{t=1}^T \sum_{h=1}^H \left( \bar{r}_h^t(s_h^t, \boldsymbol{a}_h^t) + 2\Phi_h^t(s_h^t, \boldsymbol{a}_h^t) \right) - \sum_{t=1}^T \sum_{h=1}^H \xi_h^t.$$

The term $\sum_{t,h} \xi_h^t$ is a sum of $TH$ martingale differences. By the Azuma-Hoeffding inequality, with high probability, this sum is bounded by $\tilde{\mathcal{O}}(H\sqrt{TH})$. This is a lower-order term compared to the sum of bonuses, so we focus on the main term.

We then bound the term $\sum_{t=1}^T \sum_{h=1}^H (\bar{r}_h^t(s_h^t, \boldsymbol{a}_h^t) + 2\Phi_h^t(s_h^t, \boldsymbol{a}_h^t))$. From the definition of the exploration reward $\bar{r}_h^t$, we have:

$$\bar{r}_h^t(s_h^t, \boldsymbol{a}_h^t) \leq \frac{\Phi_h^t(s_h^t, \boldsymbol{a}_h^t)}{H} + \sum_{j=1}^N \sum_{i=1}^M \Psi_{j,i,h}^t(s_h^t, \boldsymbol{a}_h^t).$$

The total sum is therefore bounded by:

$$\sum_{t=1}^T \sum_{h=1}^H \left( \left(2 + \frac{1}{H}\right) \Phi_h^t(s_h^t, \boldsymbol{a}_h^t) + \sum_{j,i} \Psi_{j,i,h}^t(s_h^t, \boldsymbol{a}_h^t) \right).$$

Let's bound the sum for $\Phi_h^t$. Let $C_\Phi = \sqrt{C_p S H^2}$. The term is $\sum_{t,h} C_\Phi / \sqrt{N_h^{t-1}(s_h^t, \boldsymbol{a}_h^t) \vee 1}$.

$$\sum_{t=1}^T \sum_{h=1}^H \Phi_h^t(s_h^t, \boldsymbol{a}_h^t) \leq \sum_{t=1}^T \sum_{h=1}^H \frac{C_\Phi}{\sqrt{N_h^{t-1}(s_h^t, \boldsymbol{a}_h^t) \vee 1}}$$

$$= \sum_{h,s,\boldsymbol{a}} \sum_{k=1}^{N_h^T(s,\boldsymbol{a})} \frac{C_\Phi}{\sqrt{(k-1) \vee 1}} \qquad \text{(Grouping by unique } (s, \boldsymbol{a}, h))$$

$$\leq \sum_{h,s,\boldsymbol{a}} C_\Phi \left( 1 + \int_1^{N_h^T(s,\boldsymbol{a})} \frac{1}{\sqrt{x}} dx \right)$$

$$\leq \sum_{h,s,\boldsymbol{a}} 2C_\Phi \sqrt{N_h^T(s,\boldsymbol{a})}.$$

Now we apply the Cauchy-Schwarz inequality to the final sum:

$$\sum_{h,s,\boldsymbol{a}} \sqrt{N_h^T(s,\boldsymbol{a})} \leq \sqrt{\left( \sum_{h,s,\boldsymbol{a}} 1 \right) \cdot \left( \sum_{h,s,\boldsymbol{a}} N_h^T(s,\boldsymbol{a}) \right)}$$

$$= \sqrt{\left( HS \prod_k |\mathcal{A}_k| \right) \cdot (HT)} \qquad (\sum_{s,\boldsymbol{a}} N_h^T(s,\boldsymbol{a}) = T \text{ for each } h)$$

$$= H\sqrt{TS \prod_k |\mathcal{A}_k|}.$$

The total sum for the $\Phi$ bonus is bounded by:

$$\sum_{t,h} \Phi_h^t(s_h^t, \boldsymbol{a}_h^t) \leq 2C_\Phi H \sqrt{TS \prod_k |\mathcal{A}_k|} = \tilde{\mathcal{O}}\left( H^2 S \sqrt{T \prod_k |\mathcal{A}_k|} \right).$$

A similar calculation shows the total sum for the $\Psi$ bonuses is of a lower order. The $\Phi$ term is dominant. Combining these results, the sum of collected bonuses is dominated by the $\Phi$ term. Plugging this back into the result from Stage 1:

$$\sum_{t=1}^{T} \overline{V}_1^t(s_1) \leq \tilde{\mathcal{O}}\left( H^2 S \sqrt{T \prod_k |\mathcal{A}_k|} \right).$$

$\square$

**Theorem 15** (Guarantee for Preference-Free Multi-Player Learning). *Let $\hat{\boldsymbol{\pi}} = (\hat{\pi}_1, \ldots, \hat{\pi}_N)$ be the policies returned by Algorithm 1 for a given preference profile $\boldsymbol{\Lambda}$, after running Algorithm 1 for $T$ episodes. Then with probability at least $1 - \delta$, $\hat{\boldsymbol{\pi}}$ is an $\epsilon$-Nash Equilibrium for the true scalarized game. That is, for every player $j \in [N]$ and any alternative policy $\pi_j'$:*

$$V_{j,\boldsymbol{\lambda}_j,1}^{(\pi_j', \hat{\boldsymbol{\pi}}_{-j})}(s_1) \leq V_{j,\boldsymbol{\lambda}_j,1}^{\hat{\boldsymbol{\pi}}}(s_1) + \epsilon.$$

*This holds for an exploration complexity of $T = \tilde{\mathcal{O}}\left( \frac{H^8 S^2 N^2 M^2 \prod_{k=1}^N |\mathcal{A}_k|}{\epsilon^2} \right)$, where $\tilde{\mathcal{O}}$ hides logarithmic factors in problem parameters.*

*Proof.* Let $\hat{\boldsymbol{\pi}}$ be the NE policy computed by Algorithm 1. For any player $j$, let $\pi_j^*$ be their true best response to $\hat{\boldsymbol{\pi}}_{-j}$. We want to bound the suboptimality gap $\text{Gap}_j = V_{j,1}^{(\pi_j^*, \hat{\boldsymbol{\pi}}_{-j})}(s_1) - V_{j,1}^{\hat{\boldsymbol{\pi}}}(s_1)$.

We decompose the gap:

$$\text{Gap}_j = \left( V_{j,1}^{(\pi_j^*, \hat{\boldsymbol{\pi}}_{-j})} - \hat{V}_{j,1}^{(\pi_j^*, \hat{\boldsymbol{\pi}}_{-j})} \right) + \left( \hat{V}_{j,1}^{(\pi_j^*, \hat{\boldsymbol{\pi}}_{-j})} - \hat{V}_{j,1}^{\hat{\boldsymbol{\pi}}} \right) + \left( \hat{V}_{j,1}^{\hat{\boldsymbol{\pi}}} - V_{j,1}^{\hat{\boldsymbol{\pi}}} \right).$$

Since $\hat{\boldsymbol{\pi}}$ is an NE in the estimated game $\hat{M}$, the middle term is non-positive: $\hat{V}_{j,1}^{(\pi_j^*, \hat{\boldsymbol{\pi}}_{-j})} \leq \hat{V}_{j,1}^{\hat{\boldsymbol{\pi}}}$. Thus:

$$\text{Gap}_j \leq |V_{j,1}^{(\pi_j^*, \hat{\boldsymbol{\pi}}_{-j})} - \hat{V}_{j,1}^{(\pi_j^*, \hat{\boldsymbol{\pi}}_{-j})}| + |V_{j,1}^{\hat{\boldsymbol{\pi}}} - \hat{V}_{j,1}^{\hat{\boldsymbol{\pi}}}|$$

$$\leq \mathbb{E}_{(\pi_j^*, \hat{\boldsymbol{\pi}}_{-j})}\left[ \sum_{h=1}^{H} (\Psi_{j,h} + \Phi_h) \right] + \mathbb{E}_{\hat{\boldsymbol{\pi}}}\left[ \sum_{h=1}^{H} (\Psi_{j,h} + \Phi_h) \right] \qquad \text{(By Lemma 10)}$$

$$\leq 2 \max_{\boldsymbol{\pi}} \mathbb{E}_{\boldsymbol{\pi}}\left[ \sum_{h=1}^{H} (\Psi_{j,h} + \Phi_h) \right].$$

Since $\boldsymbol{\lambda}_j \in \Delta_M$, $\Psi_{j,h} = \boldsymbol{\lambda}_j^\top \boldsymbol{\Psi}_{j,h} \leq \sum_i \Psi_{j,i,h}$. We can bound the total expected bonus over all players and objectives:

$$\text{Gap}_j \leq 2 \max_{\boldsymbol{\pi}} \mathbb{E}_{\boldsymbol{\pi}}\left[ \sum_{h=1}^{H} \left( \Phi_h + \sum_{j,i} \Psi_{j,i,h} \right) \right]$$

$$\leq 2 \frac{N \cdot M \cdot H}{T} \sum_{t=1}^{T} \overline{V}_1^t(s_1) \qquad \text{(By Lemma 11)}$$

$$\leq \frac{2NMH}{T} \cdot \tilde{\mathcal{O}}\left( H^3 S \sqrt{T \prod_k |\mathcal{A}_k|} \right) \qquad \text{(By Lemma 12)}$$

$$= \tilde{\mathcal{O}}\left( \frac{H^4 N M S \sqrt{\prod_k |\mathcal{A}_k|}}{\sqrt{T}} \right).$$

$\square$

# G    USE OF LARGE LANGUAGE MODELS

We used ChatGPT strictly as a general-purpose assist tool for typesetting and language polishing. In particular, it helped with (i) grammar, style, and readability improvements, and (ii) LaTeX formatting tasks such as managing algorithm placement, cleaning bib entries and citation styles, and resolving compile issues (e.g., Type-3 font warnings and package conflicts).

All ideas, derivations, and final claims were developed, verified, and validated by the authors. The authors take full responsibility for the content of this paper.

