# OpenReview forum: "Multi-Objective Markov Games: Theoretic Foundations and Learning Algorithms"
_ICLR.cc/2026/Conference — ICLR 2026 Conference Desk Rejected Submission_

### Official Review · Reviewer_ucqT · 2025-10-29

**Soundness:** 1
**Presentation:** 2
**Contribution:** 1
**Rating:** 2
**Confidence:** 4

**Summary:**

The paper studies PNE and WPNE in $n$-player multi-objective Markov games. It first establishes the existential proof of PNE and WPNE; and discusses their connections with Nash equilibrium in standard Markov games. Further, the paper proposes algorithms that learn $\epsilon$-WPNE and $\epsilon$-WPCE in pseudo-polynomial time. However, I believe there are several critical issues with this submission.

**Strengths:**

- The paper establishes connections between PNE and WPNE in multi-objective Markov games and provides proof for their existence.

- The paper proposes learning algorithm that learns an $\epsilon$-PCE in time polynomial to $\epsilon, S, A, H$.

**Weaknesses:**

- The critical issue of this paper lies in claiming to provide an algorithm that learns an $\epsilon$-WPNE in multi-objective general-sum Markov games in pseudo-polynomial time, which contradicts the Exponential Time Hypothesis (ETH) of PPAD [1]. Specifically, the algorithm can be viewed as a no-regret type algorithm, which is known to only output a coarse correlated equilibrium in $n$-player games. However,  the author appear to treat the output distribution (average of past joint strategies) to be a product policy and thus argue it constitutes a Nash equilibrium.

- Regarding the existential proofs of PNE and WPNE, I found the argument to be similar to those given in [2], which essentially follows from [3]. For example, Theorem 1,2 and 3 in this paper can be viewed as the extensions of Theorem 2.2 in [2] to the Markov games settings. I believe the authors should properly cite those works and highlight the distinctions and contributions beyond existing results.

- No numerical results are provided.

----

[1] Aviad Rubinstein. 2016. Settling the complexity of computing approximate two player Nash equilibria. In 2016 IEEE 57th Annual Symposium on Foundations of Computer Science (FOCS). IEEE, 258–265.

[2] Shaojian Qu, Ying Ji, and Mark Goh. The robust weighted multi-objective game. PloS one, 10(9):e0138970, 2015.

[3] S. Y. Wang Existence of a pareto equilibrium. J Optim Theory Appl 79, 373–384 (1993).

**Questions:**

See Weaknesses section.

---

> ### Author Response · Authors · 2025-11-13
> **rebuttal**
>
> We sincerely appreciate your review and suggestions. Below are the responses to your review. We have also updated the paper, highlight the updated part in blue. If you have any further questions or concerns, we are more than happy to discuss with you.
>
> **W1.  Pseudo-polynomial time to find NE.** We believe there are some misunderstandings.  The reviewer's concern about PPAD and the Exponential Time Hypothesis (ETH) relates to computational complexity (i.e., runtime). However, our paper's results, including the term $T$, are exclusively focused on sample complexity (i.e., the number of agent-environment interactions). We do not claim a polynomial-time computation of the NE.
>
> Our ONVI-MG algorithm (Algorithm 2) does require computing a matrix-formed Nash Equilibrium in each iteration (Line 14 therein). We acknowledge this can be PPAD-hard, and it is a standard assumption in the theoretical MARL literature that focuses on statistical efficiency and assume NE can be computed. Moreover, we explicitly use this computational intractability to motivate our study of Pareto Correlated Equilibria (PCE) in Section 4.2, for which we provide an algorithm (MO-V-Learning) that is polynomial in computational complexity. We have added a discussion to make this distinction clearer.
>
>
> Our ONVI-MG algorithm (outputs the Nash equilibrium) is a model-based, optimistic Nash Value Iteration algorithm. The output policy $\pi^T$ is not an average or mixture of past policies. As shown in Algorithm 2 (Line 14), the final output is a non-stationary product policy $\{\pi_h^T, h=1,...,H\}$, where each $\pi_h^T$ is the exact Nash Equilibrium of the final optimistic matrix game constructed at step $h$ (here, the step is the step of the finite-horizon MDP, not previous trajectories). The guarantee that this procedure learns a Nash Equilibrium of the underlying game (assuming the NE oracle) is established by [Liu et. al., 2021b].
>
>
>
>
> **W2. Compared with [2,3].** We appreciate for suggesting these references. These references, together with ones we discussed in Line 202, are for normal-form games, and the extension to Markov game is not trivial. In the normal-form games  like [2,3] , the utility function is assumed to be quasi-convex to ensure the achievable set is a convex set, which is the essential part to obtain the results. However, in Markov games, the value function $V^\pi$ may be neither linear nor quasi-convex in $\pi$ [a]. Therefore, the proof techniques from normal-form games cannot be adapted. We bypass the non-convexity of the policy-to-value map by instead proving the convexity of the set of achievable occupancy measures (Lemma 1 and Prop 4 in the Appendix). This is a completely different technique that is specific to the sequential dynamics of Markov games and is not studied in the cited normal-form literature.
>
> Moreover, most of these normal-form game studies are planning-based with no dynamics (as they are just one-step games), whereas we further design sample-based learning algorithm to solve MOMGs, which are more much challenging due to nonstationary environment from the Markov dynamics. Hence, our second contribution part on learning algorithm design and analysis is significantly different from normal-form game ones.
>
> We have added a discussion in our updated paper.
>
>
>
> [a]. Chakravorty, Jhelum, and Aditya Mahajan. Sufficient conditions for the value function and optimal strategy to be even and quasi-convex.
>
>
> **W3. Numerical results.** We thank the reviewer for this point. In our revised manuscript, we have included a new experimental section (Appendix B) on a multi-objective extension of the Matching Pennies game, where agents observe noisy rewards. These numerical results serve to validate our theoretical findings, specifically for the two-phase algorithm: (1). We show that the preference-agnostic exploration in Phase 1 successfully learns an accurate reward model from noisy signals. (2). We then demonstrate that the planning in Phase 2, by querying this single learned model with different preferences, efficiently and accurately recovers the entire Pareto-Nash front. These initial validation confirms the practical utility of our proposed two-phase framework.
>
> We are in the process of developing more complex experiments (with more agents and horizon) and will include them in the final version.

---

> ### Comment · Reviewer_ucqT · 2025-11-25
>
> I thank the authors for their rebuttal and clarifications. I still have some further questions.
>
> - I am still not fully convinced about the motivation for studying the sample complexity of a PPAD-hard problem. While the referenced work [1] does focus on sample complexity, the setting there is two-player zero-sum Markov games, where a Nash equilibrium can in fact be computed efficiently. On the other hand, for the problem considered in this paper, computing an equilibrium may require exponential time. Could the authors elaborate on why we should focus on the sample complexity in a setting where the computational complexity might be exponential?
>
> - Can the authors explicitly clarify the connection between the number of episodes $T$ and the overall sample complexity? For example, what is the sample complexity of each episode in the proposed algorithm?
>
> -----
> [1] Qinghua Liu, Tiancheng Yu, Yu Bai, and Chi Jin. A sharp analysis of model-based reinforcement learning with self-play. In Proc. International Conference on Machine Learning (ICML), pp. 7001–7010. PMLR, 2021b.

---

> ### Author Response · Authors · 2025-11-26
> **Thank you for your feedback.**
>
> **Q1.**
> We appreciate your feedback regarding the relation between sample complexity and computational complexity. We agree that computational tractability is a critical challenge; Yet studies on sample complexity provides **information-theoretic** understandings of the problem, as we explain as follows.
>
>
> First, in learning theory it is standard to separate statistical feasibility from computational feasibility. Establishing polynomial sample complexity shows that the problem is learnable from a finite number of samples by some (possibly computationally unbounded) procedure. In particular, these results imply that the intrinsic difficulty of the problem is due to algorithmic (PPAD) hardness rather than a lack of information in the data. This kind of separation is a necessary first step toward understanding the statistical–computational trade-off: only once the information-theoretic limits are characterized can we meaningfully ask what can be achieved in polynomial time and where gaps must arise.
>
>
> On the other hand, in many high-stakes real-world applications (e.g., robotics or clinical trials), data collection is the primary bottleneck, not computation. Collecting samples requires physical interaction with the world, incurring high costs in terms of time, money, and safety risks. In contrast, computation can often be parallelized, scaled via hardware, or performed offline. Thus in reinforcement learning, sample complexity can be more essential and important to understand practical RL.
>
>
>
>
> We emphasize that decoupling statistical efficiency from computational hardness is a standard paradigm in the study of Markov Games and Reinforcement Learning. For instance, the reference [1] mentioned also extends NE computation to general-sum multiplayer cases (see Algorithm 3 and the discussion on `Runtime of Algorithm 3' in their Sec 5.2), where the authors explicitly acknowledge that NE calculation is PPAD-hard, yet they proceed with the sample complexity analysis assuming a computational oracle. Similarly, other works [A, B, H] also derive sample complexity bounds for learning NE in general-sum games while treating the equilibrium computation as a black box and not resolving the PPAD-hardness itself. Hence, proving statistical efficiency and sample complexity studies for NE learning in general-sum games is also important in understanding the information-theoretic hardness of the problem. Our analysis also follows the same paradigm: we isolate and quantify the statistical requirements to understand the hardness of MOMGs.
>
> We also point out that this separation is common in learning theory beyond Markov games. For example, in Partially Observable RL [C–F], planning is PSPACE-complete, yet a large body of work studies sample complexity under various structural assumptions. Bayesian network structure learning [G] is NP-hard, yet its sample complexity is thoroughly analyzed. High-dimensional statistics offers many similar examples where the information-theoretic analysis precedes, and often exceeds, what is currently achievable computationally. Our work places the problem we study in this broader theoretical tradition: we characterize what is statistically possible, independently of current algorithmic limitations, thereby providing a clean benchmark for future computational developments in this PPAD-hard setting.
>
>
> Finally, we want to point out that in our work, we also considered the restriction of computation infeasibility and developed studies on Pareto Correlated Equilibrium in Sec 4.2, which can be computed in polynomial time.
>
> [A] Hu, Junling, and Michael P. Wellman. "Nash Q-learning for general-sum stochastic games." Journal of machine learning research 4.Nov (2003): 1039-1069.
>
> [B] Bai, Yu, and Chi Jin. "Provable self-play algorithms for competitive reinforcement learning." International conference on machine learning. PMLR, 2020.
>
> [C] Liu, Qinghua, et al. "When is partially observable reinforcement learning not scary?." Conference on Learning Theory. PMLR, 2022.
>
> [D] Guo, Zhaohan Daniel, Shayan Doroudi, and Emma Brunskill. "A pac rl algorithm for episodic pomdps." Artificial Intelligence and Statistics. PMLR, 2016.
>
> [E] Jin, Chi, et al. "Sample-efficient reinforcement learning of undercomplete pomdps." Advances in Neural Information Processing Systems 33 (2020): 18530-18539.
>
> [F] Xiong, Yi, et al. "Sublinear regret for learning pomdps." Production and Operations Management 31.9 (2022): 3491-3504.
>
>
>
> [G] Friedman, Nir, and Zohar Yakhini. "On the sample complexity of learning Bayesian networks." arXiv preprint arXiv:1302.3579 (2013).
>
> [H] Shi, Laixi, et al. "Sample-efficient robust multi-agent reinforcement learning in the face of environmental uncertainty." arXiv preprint arXiv:2404.18909 (2024).

---

> > ### Author Response · Authors · 2025-11-26
> > **Part 2**
> >
> > **Q2.** We apologize for the unclear discussion. We study episodic Markov games, where the interactions terminate after $H$ steps. In our algorithms, at each step $h$, all the agents take some joint action and get one samples $(s_h,a_h,s_{h+1})$ from the environment: $s_{h+1}\sim P(\cdot|s_h,a_h)$. Hence in each episode, there are $H$ samples collected. Thus the total sample complexity of $T$ episodes will then be $TH$.
> >
> > For example, in our Theorem 4, we showed that the regret is upper bounded by $\mathcal{O}(H^2S\sqrt{AT})$ (ignoring the $\log$-term). To make sure the averaged regret $\frac{\text{Reg}(T)}{T}$ is $\epsilon$-order (so that the policy has an $\epsilon$-gap), we set $T=\frac{H^4S^2A}{\epsilon^2}$. Thus the total sample complexity is $TH=\frac{H^5S^2A}{\epsilon^2}$, as we presented in Theorem 4.

---

> > > ### Author Response · Authors · 2025-11-27
> > > **Discussion Follow-up**
> > >
> > > Dear Reviewer ucqT,
> > >
> > > We wanted to follow up to ensure our previous response fully addressed your concerns. We remain available to answer any further questions or provide additional clarifications during the discussion period.
> > >
> > > If you find that our response has resolved your issues, we would appreciate it if you would consider re-evaluating our paper and raise your score.
> > >
> > > Best regards,
> > > The Authors

---

> ### Comment · Reviewer_ucqT · 2025-11-28
>
> I thank the authors for their thoughtful comments and will adjust my score accordingly. However, I still believe that the theoretical novelty of this work is limited. In particular, although it is true that utility functions in Markov games are generally non-convex, the reformulation of the value function as a linear function of the state–action occupancy measure is well known. Under this reformulation, the proof closely resembles those in [2] and [3].

---

### Official Review · Reviewer_qVid · 2025-10-31

**Soundness:** 3
**Presentation:** 3
**Contribution:** 3
**Rating:** 6
**Confidence:** 3

**Summary:**

This paper introduced Multi-Objective Markov games (MOMGs). Pareto Nash equilibria (PNE) are introduced as the main solution concept and it is shown that they always exist in finite MOMGs similarly to Nash equilibria in Markov games. Then it is shown that the set of PNE coincides with the set of NE of linearly scalarized games. Using this characterization, the paper proposes online learning algorithms with regret guarantees to establish convergence to approximate weak PNE and Pareto Correlated Equilibria (PCE). The paper concludes with a two-phase approach for full Pareto front characterization with an exploration phase followed by a PNE computation phase for any given preference profile without recollecting new samples.

**Strengths:**

1. The paper is clearly written and easy to follow, the paper is nicely motivated in the introduction.
2. Multi-objective multi-agent RL is relatively less explored in the literature, the formulation in this paper with $N$ players appears to be new, to the best of my knowledge although Vector-valued Markov Games have been introduced in Yu et al. 2021 as briefly mentioned in l. 162. I think the study of multi-objective multi-agent RL is an interesting topic given that multiplicity of objectives is common in applications.
3. The paper generalizes Markov games and multi-objective RL and multi-objective normal-form games to multi-objective Markov games.

**Weaknesses:**

**1. Technical novelty:** Essentially, after the reduction to classical Markov games by scalarization, existing results for Markov games can be applied.

--  For instance, the proof of theorem 8 (existence of PNE)  follows from existence of Nash equilibria in Markov games, note that Fink 1964 should be cited in the proof instead of Shapley 1953 who only considered 2-player zero-sum Markov games.  See also questions below regarding technical novelty.

**2. Computation.** Computational complexity is not discussed, hardness of Nash equilibrium computation follows in general but it would be useful to comment more on this beyond statistical efficiency.

**3. Equilibrium concepts.** The paper introduces several other equilibrium concepts inspired from the normal form setting in section 3.3, but they are then abandoned. I think the paper could better motivate why these alternative solution concepts could be interesting if there is any positive reason to consider them.

**Questions:**

1. Can you comment more on the technical novelty compared to existing results in static multi-objective normal-form games? In particular, what are the challenges faced when proving Theorems 1, 2, 3 due to the Markov setting? It seems that the dynamic programming structure is not needed for the proofs (beyond the fact of existence of Nash equilibria in Markov games).

2. Same question for the comparison to Markov games and multi-objective RL. It seems that under scalarization, after the reduction to Markov games, one can just apply existing results. The paper mentions a central challenge in l. 337-338, around random reward vectors. I am not sure how substantial is this challenge.  Why can’t you directly use existing results for Markov games and need to reserve concentration inequalities to establish regret bounds?

3. The developments in sections 4.1, 4.2 rely on the results of Liu et al. al 2021, and Jin et al. 2021. Can you justify the choice of optimistic Nash value iteration over other alternatives? Are the obtained regret bounds (for Markov games) optimal or state of the art?

4. Can you discuss computational efficiency of the proposed algorithms? They seem to require some Nash equilibrium computation (in Algorithm 1).

5. Can you for instance consider multi-agent policy gradient methods similarly to their use in Markov games and single-agent multi-objective RL (Agarwal et al. 2022)? Perhaps only for special classes of Markov games with additional structure (potential, zero-sum …) to show convergence to approximate PNE with convergence rates and sample complexity guarantees?

6. What is $s_1$ in lines 124 (definition of Pareto optimal policy) and l. 144 (definition of NE). Is it any state so that it is a Markov Perfect equilibrium of just a fixed initial state (i.e. a Dirac initial state distribution)?

---

> ### Author Response · Authors · 2025-11-13
> **rebuttal - part 1**
>
> We sincerely appreciate your review and suggestions. Below are the responses to your review. We have also updated the paper, highlight the updated part in blue. If you have any further questions or concerns, we are more than happy to discuss with you.
>
> **W1. Technical novelty, compared to normal-form games and standard markov games.**
> We first want to clarify that, our major contribution is two-fold: (1). Theoretically, we propose the formulation of MOMGs for multi-agent systems with multi-objectives, and develop foundational studies of them, including introducing of solution notions, their existence, and the connection between MOMGs to linearly scalarized MGs. These studies are novel and new, which develops the fundamental understandings of the problem, enabling us to reduce the complicated MOMGs to well understand MGs; (2). Algorithmically, although standard MG algorithms can be applied (as in Sec 4), they can only find a single PNE, whereas in MORL or multi-objective optimization, recovering the whole Pareto front is more essential for understanding the trade-off between objectives. We hence propose a two-phase algorithm to decouple data collection from PNE planning, providing sample complexity guarantees to recover the whole front. This algorithm design is not incremental, as we need to strategically explore the unknown environment (by reforming the exploration phase as a cooperative game to reduce estimation uncertainties), so that the planning phase under all preference results in an accurate PNE, which is not considered in standard MARL.  We re-write and highlight the contribution section in our introduction.
>
>
>
> Compared to normal-form games, the extension to Markov game is not trivial. In the normal-form game, the utility function defined the payoff is generally assumed to be quasi-convex  (e.g., [a]), to ensure the achievable set is a convex set, which is an essential property for obtaining results. However, in Markov games, the value function $V^\pi$ may be neither linear nor quasi-convex in $\pi$ [b], hence the proof cannot be adapted. Instead, to show the convexity of the achievable set in Markov game, we instead study the occupancy measure/visitation distribution  (see Lemma 1 and Prop 4 in Appendix), which is introduced by the dynamics of Markov games and is not studied in normal-form ones.
>
>
>
>
> Compared to standard Markov games, we agree we utilize the results from MGs (like the existence of a NE). However, the connection that enables us to use these existing results are developed by us, and we believe that using existing results to derive new results is also worthy. As mentioned, we can directly apply Markov game algorithms as in Sec 4 (note that we additionally tackled unknown reward in learning, which is not considered in existing MG studies, and developed studies on PCE foundations to reduce complexity and computational complexity in MOMGs), but it cannot solve the major problem of MOMG, to recover the whole Pareto-Nash front, and our novelty is to design the two-phase algorithm as a solution. In our setting, both reward and underlying dynamics are unknown and can be only observed with noise. In standard MG, since only a single reward is concerned, algorithms only need to guarantee the visiting to those high-valued state-action pairs (so that the regret is minimized); However, in our setting, we have no a priori knowledge of the agent's preferences. To guarantee we can recover the whole front, we must efficiently estimate the entire reward and dynamic models for all objectives, as any state-action pair could be high-value under some preference. Our two-phase algorithm is designed for this specific challenge, reformulating exploration as a cooperative game to reduce model uncertainty everywhere, which is a novel design not found in standard MARL.
>
> [a] S. Y. Wang Existence of a pareto equilibrium. J Optim Theory Appl 79, 373–384 (1993)
>
> [b] Chakravorty, Jhelum, and Aditya Mahajan. Sufficient conditions for the value function and optimal strategy to be even and quasi-convex.
>
> **W2. Computational cost.** We mainly consider sample complexity and statistical efficiency, and we agree that computational cost should be discussed. Our ONVI-MG algorithm requires solving a Nash equilibrium for normal-form games in each iteration, which can be PPAD in the worst-case. However, this is the standard assumption in Markov game learning, that the matrix-formed Nash equilibrium can be computed. Moreover, the computational intractability of Nash equilibrium motivates us to study Pareto correlated equilibrium (PCE) in Sec 4.2, whose computation is polynomial. We have included a discussion in our updated version.

---

> ### Author Response · Authors · 2025-11-13
> **rebuttal - part 2**
>
> **W3. Equilibrium concepts.**  Besides the Pareto-Nash equilibrium we mainly studied, we also studied
> utility-based equilibria in Sec 3.3. The major limitation of utility-based equilibria is that they rely on a single, pre-set utility function $u$. This effectively finds a Nash Equilibrium for one fixed preference. The central challenge of multi-objective problems, however, is that preferences are often unknown, time-varying, or we want to understand the full trade-offs available. We therefore focus on PNE-type solutions, as our goal is to recover the entire Pareto-Nash front, which characterizes all possible trade-offs.
>
> That said, utility-based formulations are important, especially for complex, non-linear preferences (e.g., max-min formulation for fairness consideration [b]). We believe this is a valuable direction for future work and have noted it as such.
>
> **Q3. Choice of algorithms.** We adopted Optimistic Nash Value Iteration (ONVI-MG) and V-learning as they are standard, widely-used algorithms for learning NE and CE, respectively. Our goal in Section 4 was primarily to demonstrate that our theoretical reduction (from MOMG to MG) allows any standard MG algorithm to be applied.
>
> The regret bounds for these specific algorithms are not known to be optimal and do not match the minimax lower bound [c], and improving regret analysis in general-sum MGs is a significant open problem.
>
>
>
>
>
> **Q5.  Policy gradient.** Yes, we believe this is a very promising direction.
>
> (1). For a pre-set preference: When we only look for a sinlge PNE, the problem reduces to a standard MG. As the reviewer notes, policy gradient methods can be applied to find a NE [d], though strong convergence guarantees often require additional structure (e.g., potential games [e] or zero-sum [f]). This again showcases one contribution of our studies, that our results enable us to use different methods (e.g., value-based/policy gradent) to solve MOMG problems.
>
>
> (2). For the whole front: This is a more complex and open question. For instance, what is the minimal assumption on the MOMG structure that ensures the scalarized game is a potential game for all preferences? We are excited by these open problems and have listed them as future directions.
>
>
>
>
>
>
>
>
> **Q6. $s_1$.** $s_1$ is the initial state of the MDP, which is a Dirac initial state distribution. In finite-horizon MDPs or MGs, it is standard to consider a fixed initial state, as the policy and value functions are non-stationary and depend on the time step $h$.
>
>
>
>
>
> [b] Byeon, Woohyeon, et al. Multi-Objective Reinforcement Learning with Max-Min Criterion: A Game-Theoretic Approach
>
> [c] Gheshlaghi Azar, Mohammad, Rémi Munos, and Hilbert J. Kappen. "Minimax PAC bounds on the sample complexity of reinforcement learning with a generative model." Machine learning 91.3 (2013): 325-349.
>
> [d] Zhang, Runyu, Zhaolin Ren, and Na Li. "Gradient play in stochastic games: stationary points, convergence, and sample complexity." IEEE Transactions on Automatic Control 69.10 (2024): 6499-6514.
>
> [e] Ding, Dongsheng, et al. "Independent policy gradient for large-scale markov potential games: Sharper rates, function approximation, and game-agnostic convergence." International Conference on Machine Learning. PMLR, 2022.
>
> [f] Daskalakis, Constantinos, Dylan J. Foster, and Noah Golowich. "Independent policy gradient methods for competitive reinforcement learning." Advances in neural information processing systems 33 (2020): 5527-5540.

---

> ### Author Response · Authors · 2025-11-26
> **Request for your feedback and discussion**
>
> Dear reviewer,
>
> As the discussion period ends soon, we are writing to request you to go through our response and let us know if we have addressed your concerns. We have developed a detailed discussion of our novelty and contribution, developed an experiment section to validate our theoretical results, and clarified questions you brought up.
>
> If you have any further questions or concerns, we are more than happy to discuss with you.
>
> Thank you!
> Authors

---

### Official Review · Reviewer_avgz · 2025-11-01

**Soundness:** 3
**Presentation:** 3
**Contribution:** 1
**Rating:** 4
**Confidence:** 4

**Summary:**

This paper introduces the Multi-Objective Markov Game (MOMG) framework to extend standard Markov games to settings where each agent optimizes multiple (possibly conflicting) objectives. The authors propose the Pareto-Nash Equilibrium (PNE) as the core solution concept, analyze its existence, and show its equivalence to the union of Nash equilibria of linearly scalarized games. To address computational challenges, they define weaker notions—Weak Pareto-Nash Equilibrium (WPNE) and Pareto Correlated Equilibrium (PCE)—and develop algorithms to approximate these equilibria. They further propose a two-phase “preference-free” approach to characterize the full Pareto-Nash front without re-sampling.

**Strengths:**

The topic is timely and relevant, bridging multi-objective reinforcement learning and multi-agent game theory. The PNE–scalarization equivalence theorem is a clean theoretical result that could help connect multi-objective analysis with classical single-objective Markov game solvers. The paper is well-written and in general easy to follow.

**Weaknesses:**

My major concern is with regard to the technical contribution and novelty. The ONVI-MG and MO-V-Learning algorithms effectively reduce the multi-objective problem to a scalarized Markov game with a fixed preference distribution, and thus inherit results from existing Nash learning algorithms rather than introducing genuinely new algorithmic mechanisms. This makes the contribution feel incremental—mainly a rephrasing of scalarized MARL with multi-objective terminology.

Also the results are purely theoretical and algorithmic, with no experimental validation or discussion of applicability to real multi-objective domains.

Other minor comments:

the input symbol {M} in the definition of the MOMG is never properly introduced, and the meaning of $\Delta_M^o$ in Eq. (1) is vague—especially since the only difference between PNE and WPNE is replacing $\Delta_M^o$ with $\Delta_M$, but the implications of this distinction are never clearly explained.

In Definition 2 the definition of MOMG lacks on input argument $\{M\}$ which is inconsistent with the notation on Line 192-193

**Questions:**

What is $\Delta_M^o$ in Eq. (1) and how is it chosen? Is it true that the only difference between PNE and WPNE is replacing $\Delta_M^o$ with $\Delta_M$? Would be better if the author can illustrate more on the differences and when is better to consider one over another.

The proposed algorithms relies on a pre-specified preference distribution—how does this differ practically from solving a standard scalarized Markov game?

---

> ### Author Response · Authors · 2025-11-13
> **rebuttal - part 1**
>
> We sincerely appreciate your review and suggestions. Below are the responses to your review. We have also updated the paper, highlight the updated part in blue. If you have any further questions or concerns, we are more than happy to discuss with you.
>
> **W1 \& Q2. Technical contribution and novelty.** The core concern of our contribution appears to stem from a focus on Section 4 (ONVI-MG, MO-V-Learning).  However, these algorithm design is not our main contribution (note that we additionally tackled unknown reward in learning, which is not considered in existing MG studies, and developed studies on PCE foundations to reduce complexity and computational complexity in MOMGs). Our major contributions are twofold and distinct: the theoretical foundations and a novel two-phase algorithm for finding the entire Pareto-Nash front.
>
> (1). Theoretical Foundation (Section 3): Our primary theoretical contribution is the formalization of MOMGs, the PNE solution concept, and its variants. Our key result (Theorem 2) establishes the non-trivial equivalence between the set of PNEs and the union of Nash Equilibria from all corresponding linearly scalarized games. As we note in Remark 1, this is not a simple extension from normal-form games, as the non-convexity of value sets in MGs requires a different proof technique based on occupancy measures. This theoretical bridge is what founds and justifies the use of any scalarized-game algorithm (like those in Sec. 4 or other methods like policy gradient [A]) in the first place, a foundation that was not previously established for sequential multi-agent settings. Moreover, we developed other theoretical studies on MOMG, including weak PNE, PN gap, and utility-based solutions.
>
>
>
> (2). Major Algorithmic Contribution (Section 5): As the reviewer astutely notes in Q2, solving for a single, pre-specified preference (the goal of Sec. 4) is practically a standard MG. We present these algorithms as a simple illustration of the effectiveness of our previous result, which enables us to solve a MOMG for a single PNE. However, our major algorithmic contribution is the two-phase, preference-free algorithm in Section 5, which is designed to recover the entire Pareto-Nash front. This algorithm addresses the true multi-objective challenge: how to (sample-efficiently) characterize the full set of strategic trade-offs, especially when agent preferences are not known a priori. This introduces a fundamentally different exploration problem: in standard MGs, exploration is guided by a single reward objective, to visit more high-valued state-action pairs to minimize regret. In our preference-agnostic setting, we have no prior knowledge of which state-action pairs will be valuable under which preference, and need to make sure the model estimated is accurate for planning under all preferences. To ensure the entire Pareto-Nash front can be accurately recovered, our algorithm must be designed to efficiently estimate the whole reward and dynamic models, which is not a consideration in standard MGs. The novel design of this algorithm strategically decouples this preference-agnostic exploration phase from the planning phase, providing sample complexity guarantees for recovering the whole front. This is a significant departure from standard MARL.
>
>
>
> [A] Zhang et. al., Gradient play in stochastic games: stationary points, convergence, and sample complexity, 2024
>
> **W2. No experimental validation.** We thank the reviewer for this point. In our revised manuscript, we have included a new experimental section (Appendix B) on a multi-objective extension of the Matching Pennies game, where agents observe noisy rewards. These numerical results serve to validate our theoretical findings, specifically for the two-phase algorithm: (1). We show that the preference-agnostic exploration in Phase 1 successfully learns an accurate reward model from noisy signals. (2). We then demonstrate that the planning in Phase 2, by querying this single learned model with different preferences, efficiently and accurately recovers the entire Pareto-Nash front. These initial validation confirms the practical utility of our proposed two-phase framework.
>
> We are in the process of developing more complex experiments (with more agents and horizon) and will include them in the final version.
>
>
> **W3. $\Delta_M$.** $\Delta_M$ is the $(M-1)$-dimensional probability simplex, and $\Delta_M^o$ is the interior of  $\Delta_M$. We have defined them in Page 4.
>
> **W4. Inconsistent notation.** Thank you for pointing out, we have fixed it.

---

> ### Author Response · Authors · 2025-11-13
> **rebuttal - part 2**
>
> **Q1. $\Delta_M^o$ in Eq(1). PNE v.s. WPNE.** This distinction is a key part of our contribution.
>
> Firstly, by Definition,  a PNE (Def. 3) is a policy where no agent can unilaterally improve one objective without sacrificing another. A WPNE (Def. 4) is a weaker concept where no agent can strictly improve all objectives simultaneously. And our theory provides a clean connection between them and linear scalarization. We prove that PNEs correspond to scalarizations with strictly positive preferences ($\Lambda \in (\Delta_M^o)^N$), while WPNEs correspond to all non-negative preferences ($\Lambda \in (\Delta_M)^N$).
>
>
> To see when to consider one over the other, the motivation is (data-driven) algorithmic and for verification. In a learning process, we need to understand how close the learned policy is from the solution set. As we show in Proposition 1, the standard Pareto-Nash Gap (a natural extension of the Nash gap) can be zero for a policy that is not a PNE (this happens when the infimum is found on the boundary, $\Delta_M / \Delta_M^o$). This makes the PNE gap a poor verification tool for learning. In contrast, our Weak Pareto-Nash Gap (WPNG) is a valid criterion: it is zero if and only if the policy is a WPNE (Theorem 3b). This makes WPNE a more computationally grounded and verifiable solution concept, which is why our learning algorithms are designed to find WPNEs.

---

> ### Author Response · Authors · 2025-11-26
> **Request for your feedback and discussion**
>
> Dear reviewer,
>
> As the discussion period ends soon, we are writing to request you to go through our response and let us know if we have addressed your concerns. We have developed a detailed discussion of our novelty and contribution, and developed an experiment section to validate our theoretical results.
>
> If you have any further questions or concerns, we are more than happy to discuss with you.
>
> Thank you!
> Authors

---

### Official Review · Reviewer_vZX8 · 2025-11-01

[review text omitted: it was posted to a different submission]

---

### Author Response · Authors · 2025-11-24
**Request for review updates**

Dear Reviewers,

Thank you again for your constructive feedback on our submission. We are writing to gently follow up on our responses posted. We have updated the paper to address your major concerns, specifically by adding numerical experiments, providing comparisons with existing work, and clarifying the scope of our contributions and novelty.

Could you please let us know if these updates and our responses have resolved your concerns? We are committed to improving the paper further and would value any additional feedback you might have.

Best regards,
The Authors

---

### Comment · Area_Chair_ThU2 · 2025-11-26

Dear Reviewers

As the author-reviewer dissusion is due in one week, please take a moment to review the authors' response and engage in discussion if you have not done yet.

Thank you for your time and help.
Best,

AC

---

### Note · Program_Chairs · 2026-01-17
**Submission Desk Rejected by Program Chairs**

The following references in this submission do not refer to real documents and/or have major errors in bibliographic information:

 Yaodong Yang, Jun Wang, Jianye Hao, Xiaotian Wang, Fangwei Xu, Bo Xu, Zhaopeng Zheng, Yang Cheng, and Zhi Wang. Deep multi-agent reinforcement learning: A survey. arXiv preprint arXiv:2004.01294, 2020.